

# Estimating the effects of aerosol, cloud, and water vapor on the recent brightening in India during the monsoon season

Feiyue Mao[1, 2], Zengxin Pan[1, *], Wei Wang [1], Xin Lu [1], Wei Gong[1, 3]

[1]State Key Laboratory of Information Engineering in Surveying, Mapping and Remote Sensing, Wuhan University, Wuhan
430079, China
[2]School of Remote Sensing and Information Engineering, Wuhan University, Wuhan 430079, China
[3]Collaborative Innovation Center for Geospatial Technology, Wuhan 430079, China

*Correspondence to*: Zengxin Pan (pzx@whu.edu.cn)

**Abstract.** India is experiencing a leveling-off trend of solar radiation, even a transition from dimming to brightening. This process is significantly complicated because of the active atmospheric action during monsoon season. In this study, we use observations from multiple sensors in the A-Train satellite constellation to evaluate the effects of aerosol, cloud, and water vapor variations on recent changes of solar radiation during monsoon season (June–September) in India from 2006 to 2015. The results show that the increase in aerosol optical depth is paradoxical with the variation of surface shortwave radiation in India. Instead, the decreases in water vapor amount and clouds significantly contribute to the brightening, further affecting the surface warming in India. In general, clouds are reduced and thinned by approximately 9.4% and 182 m when cloud water path (by 53.4 g m$^{-2}$) and particle number concentration in cloud-sky condition decrease. The corresponding change of clouds weakens the shortwave cloud radiative effect in surface by approximately 45.5 W m$^{-2}$. Moreover, the precipitable water in clear-sky decreases by approximately 3.0 mm over the brightening period. Consequently, solar brightening increases by roughly 2.8 W m$^{-2}$ owing to its weakened absorption. Overall, the decreases in water vapor and clouds result in the increased absorption of direct solar radiation in surface and subsequent surface brightening.

## 1 Transition from dimming to brightening in India

Solar radiation incidence on the earth's surface plays a fundamental role in the surface energy balance (IPCC, 2013). Negative trends in the downwelling surface solar radiation are collectively called "dimming," whereas positive trends are called "brightening" (Wild et al., 2005). Changes in the amount of solar radiation affect the temperature field, fluxes of sensible and latent heat, atmospheric and oceanic general circulation, and the hydrological cycle.

Widespread reduction in surface solar radiation from the 1960s to the 1980s has been reported by many researchers (Stanhill and Moreshet, 1994; Liepert, 2002; Wild et al., 2005). Subsequently, the term "brightening" was coined to emphasize that the decline in global solar radiation no longer continues at many sites after the late 1980s (Wild et al., 2005). Long et al. (2009) found that solar dimming reverses to an increasing trend of 6 W m$^{-2}$ per decade in the continental United States between 1995 and 2007. Wild (2009) presented that the trends in the 1980s typically reverse from a dimming to a



brightening; they reported trends of between 2.2 and 6.6 W m$^{-2}$ per decade in the period from 1980s to 2000s. However, recent studies indicate that the developments of dimming and brightening after 2000 show mixed tendencies. Wild et al. (2009) reported a continuation of brightening at sites in Europe, United States, and parts of Asia, a leveling-off at sites in Japan and Antarctica, and indications for a renewed dimming in China. Conversely, the most current study shows that

continuing brightening remains in China after 2000 (Wang and Wild, 2016).

India is endowed with abundant solar energy because of its geographic position in the tropical belt. Dimming or brightening in India is more evident and complicated than in other regions, and its effects on regional and global climate and ecosystem are amplified by the monsoon circulation in the country (Padma et al., 2007; Wild, 2012). Contrary to the variable trend of surface solar radiation in other regions globally, Padma et al. (2007) found that sites in India show continued

dimming of −8.6 W m$^{-2}$ per decade for the period 1981–2004 based on 12 stations over the Indian region. Wild et al. (2009) proposed to focus on the slight tendency toward a stabilization of surface solar radiation since the late 1990s. Furthermore, Soni et al. (2016) found a trend reversal and partial recovery from dimming to brightening around 2001 in India.

Various mechanisms can potentially contribute to dimming and brightening. Kvalevåg and Myhre (2007) considered that the major contributor to dimming is aerosols (−2.4 W m$^{-2}$), and that the secondary effect is the changes of gas

concentrations (−0.64 W m$^{-2}$) since pre-industrial times, including tropospheric ozone, $NO_2$, $CH_4$, and $CO_2$. Many studies proposed that the changes of atmospheric aerosol load, cloud cover, and cloud properties are the main factors determining solar dimming and brightening (Wild, 2009; Kumari and Goswami, 2010; Soni et al., 2016). Liepert (2002) shown that the decrease in the global radiation from 1961 to 1990 is attributed to the increases in cloud optical thickness and direct effect of aerosols, which reduced 18 and 8 W m$^{-2}$ of solar radiation, respectively. Kambezidis et al. (2012) argued that the decline in

surface solar radiation in India is attributed to the increase in anthropogenic aerosols during the last 30 years of the 20th century. This deduction is supported by the fact that observed decadal changes of anthropogenic aerosol emissions are in line with the trends in global solar radiation (Wild, 2009; Folini and Wild, 2011). Moreover, solar dimming and brightening may be of local or regional nature, and is unavoidably contributed by regional sources and meteorology (Soni et al., 2016). Here, we focus mainly on the dimming and brightening caused by changes of aerosols, water vapor, and clouds.

In this study, we use observations from multiple sensors in the A-Train satellite constellation to evaluate the effect of aerosol, cloud, and water vapor variations on recent change of solar radiation in surface during monsoon season (June–September) in India from 2006 to 2015. This study mainly aims to find the possible reasons of the change of dimming and brightening and assess the connection between the variations of aerosol, cloud, and water vapor, and the recent brightening in India.



## 2 Data and Methods

### 2.1 CloudSat data

Cloud Profile Radar (CPR) is an active millimeter-wave radar and the only instrument on CloudSat, which were launched into the A-Train constellation in April 2006. CPR has a 1.3 km cross-track and a 1.7 km along-track footprint resolution, and its effective vertical resolution at nadir is 240 m (Sassen et al., 2008). CloudSat CPR is well suited for sensing a wide variety of cloud systems from cirrus and stratus to deep convective systems, and shows slight sensitivity to the time of day or season (Rajeevan et al., 2012). However, the estimated operational sensitivity of CloudSat CPR (−32 dBZ to −30 dBZ) is insufficient to observe thin cirrus with small ice water content (IWC) (Mace et al., 2009). CloudSat can infer the cloud microphysical characteristics on the basis of the backscatter return signal measured by CloudSat CPR, including cloud particle number concentration, size, shape, and phase (Heymsfield et al., 2010). Therefore, by combining a broadband radiative flux model known as BugsRad, CloudSat can be used to quantify the three-dimensional (3D) information of cloud macrophysical and microphysical characteristics, as well as the corresponding cloud radiative effect (CRE) (L'Ecuyer et al., 2008).

The CloudSat Radar-Only Cloud Water Content (2B-CWC-RO) product contains retrieved estimates of cloud liquid water content (LWC) and IWC, cloud liquid and ice particle number concentration (LNC and INC), cloud liquid and ice particle effective radius (LER and IER), and related quantities for each radar profile measured by the CPR on CloudSat (Woods et al., 2008). Most previous studies verify the CloudSat 2B-CWC-RO product, thereby allowing us to resolve the 3D cloud microphysical characteristics in detail (Austin et al., 2009; Protat et al., 2010; Rajeevan et al., 2012). Furthermore, retrieved profiles of cloud microphysical properties form the basis of algorithm of another data product (2B-FLXHR) that consists of high vertical resolution profiles of radiative fluxes and atmospheric heating rates. L'Ecuyer et al. (2008) detected biases between the radiative data of 2B-FLXHR from CloudSat and CERES with the monthly and 5° means in the global scale. The biases of outgoing shortwave radiation (OSR), outgoing longwave radiation, surface shortwave radiation (SSR), and surface longwave radiation are less than 0.1, 5.5, 13, and 16 W m$^{-2}$, respectively. Fortunately, the uncertainties in 2B-FLXHR fluxes decrease significantly for longer-time-scale averages (L'Ecuyer et al., 2008). Therefore, CRE derived from CloudSat can credibly describe the radiative effect of clouds, especially in large space- and time-scales.

### 2.2 CALIPSO data

Cloud-Aerosol Lidar and Infrared Pathfinder Satellite Observations (CALIPSO) were also launched in April 2006 (Winker et al., 2007). CALIPSO mainly loads a Cloud Aerosol Lidar with Orthogonal Polarization (CALIOP), which is an excellent active two-wavelength (532 and 1064 nm) polarization lidar (Winker et al., 2007). CALIOP is highly sensitive to thin cirrus clouds but does not penetrate cloud layers of optical depths exceeding three to five (Mace et al., 2009). The layers detected by CALIOP are correctly identified with a high degree of confidence (> 90%) by cloud aerosol mask analysis (Liu et al., 2009). Moreover, aerosols and clouds detected by CALIOP can be classified into multi sub-types on the basis of



classification algorithms of CALIOP scene. Therefore, CALIPSO provides 3D scientific materials for and perspective on studying the change, interactions, and transportation of aerosols and clouds at a global scale (Winker et al., 2010; Pan et al., 2015; Li et al., 2017).

## 2.3 CERES/MODIS data

Aqua became a member of the A-Train constellation in May 2002, with the load of Clouds and Earth's Radiant Energy System (CERES), Moderate Resolution Imaging Spectroradiometer (MODIS), and four other sensors. CERES/Aqua Edition 4A Single Scanner Footprint (SSF) and MODIS/Aqua Edition 6 Atmosphere Level 3 Joint Products (MYD08) data are used in this study. CERES SSF data sets combine CERES radiation measurements, cloud and aerosol microphysical property retrievals based on observations of MODIS, and ancillary meteorology fields to form a comprehensive, high-quality

compilation of satellite-derived cloud, aerosol, and radiation budget information for radiation and climate studies (Loeb and Manalosmith, 2005). Moreover, this study uses land-ocean combined aerosol optical depth (AOD) at 0.55 μm from MYD08. The AOD data have been widely used and verified by many studies because of its excellent capability to derive AOD from dark to bright surfaces (Remer et al., 2005; Huang et al., 2010; Munchak et al., 2013).

## 2.4 Radiative transfer model

BUGSrad is the official radiative transfer model used by the product of CloudSat 2B-FLXHR (Fu and Liou, 1992; L'Ecuyer et al., 2008). It is based on the two-stream, doubling-adding solution to the radiative transfer equation with the assumption of a plane-parallel atmosphere (Ritter and Geleyn, 1992). BUGSrad compute the Molecular absorption and scattering properties based on the correlated-k formulation (Fu and Liou, 1992). The calculation of BUGSrad is parallelly applied over six shortwave (SW) bands, and a constant hemisphere formulation is applied to 12 longwave (LW) bands. These bands are

appropriately weighted and combined into the two broadband flux estimates that are ultimately reported, one covering the SW from 0 to 4 μm and the other over the LW above 4 μm. Based on the comparison experiment with the observations of Atmospheric Radiation Measurement, the bias of SW and LW in clear-sky are 1.2 and 2.2 W m$^{-2}$ in TOA, respectively (Larson et al., 2007).

## 2.5 Principle and methods

In this study, the multiple sensors from A-Train are used to investigate the possible reasons of the dimming and brightening variation during monsoon season (June–September) in India from 2006 to 2015. The MYD08 and Level 2 aerosol layer products from MODIS and CALIPSO are used to describe the change of aerosol loading in India recently. The SSF product from CERES/Aqua is considered to evaluate the radiation energy variations. The CloudSat 2B-CLDCLASS, 2B-CWC-RO, and 2B-FLXHR products are used to quantify the 3D changes of cloud macrophysical and microphysical characteristics, and

the corresponding variable radiative effect of cloud. Moreover, environmental conditions are obtained from the European Center for Medium range Weather Forecast-AUXiliary analysis (ECMWF-AUX) product. The CRE inferred based on



CERES is always disturbed by water vapor, aerosol, and limited capability for MODIS to detect thin cirrus; furthermore, the CERES considers the impact of multi-layer clouds insufficiently (Sohn et al., 2010; Masters, 2012). Therefore, this study uses the CloudSat 2B-FLXHR to evaluate the effect of cloud variations on radiation.

Strict selection procedures are implemented to control the quality of data products to ensure credible conclusions. The CPR cloud mask and the radar reflectivity from the 2B-GEOPROF are set to be more than 20 and −28 dbZ, respectively, and the quality flag from the 2B-CLDCLASS being confident to CloudSat (Mace and Zhang, 2014). Note that we define the cloud-sky and clear-sky condition for all products as the scenes with cloud mask more or less than 20, which indicates that the probability of a false detection is less than 5% in comparison with CALIPSO. Data quality for CALIPSO is maintained by screening the cloud layer with cloud and aerosol discrimination (i.e., more than 20) (Hu et al., 2009). We only use the radiative data from CERES with sensor viewing zenith angle in surface less than 60 to restrict the uncertainty from the satellite non-nadir point (Christopher and Zhang, 2002). In this study, the shortwave (SW) solar radiation (< 5 μm) is assumed as the total incoming solar radiation in top of atmosphere (TOA) and surface. Surface solar radiation is analyzed in a relative sense with variable aerosols, clouds, and water vapor; thus, this study focuses mostly on the change of solar radiation but less on the absolute values of solar radiation.

CRE, also called cloud radiative forcing, has been widely used to quantify the degree of cloud–radiation interactions. CRE is defined as the net (down minus up) flux difference between the all-sky and clear-sky conditions on the atmosphere, surface, or TOA as following Eq. (1):

$$CRE = (F^{\downarrow} - F^{\uparrow})_{All-Sky} - (F^{\downarrow} - F^{\uparrow})_{Clear-Sky}, \qquad (1)$$

where $F^{\downarrow}$ and $F^{\uparrow}$ are the downwelling and upwelling radiative fluxes in TOA or surface, respectively.

One aim of this study is to quantify the impact of cloud variations on TOA and surface fluxes. Therefore, the radiative data from CloudSat 2B-FLXHR products in surface and TOA are used. Moreover, this study uses the precipitable water (PW) variation to describe the column change of water vapor in the atmosphere. PW is defined as the total water vapor of atmospheric column and is calculated on the basis of the data of environmental parameters from ECMWF-AUX product as following Eq. (2):

$$PW = \frac{1}{g_0} \int_{P_{TOA}}^{P_{Surface}} q(P)dP, \qquad (2)$$

where $g_0 = 9.80665$ m s$^{-2}$ is the standard acceleration due to gravity at mean sea level, $P$ is atmospheric pressure, and $q(P)$ is the specific humidity of air as a function of atmospheric pressure. We accurately evaluated the effect of water vapor variations on solar radiation based on a radiative transfer model (BUGSrad). For the input of the BUGSrad, we only change the annual average water vapor amount, and others enviromental parameters were set as the average value to exclude the effect of others factors. Due to the difference of averaged methods which were used by CloudSat and BUGSrad to evaluate



the radiative effect of water vapor, there possibly are some difference between the radiative fluxes calculated based on CloudSat and BUGSrad.

## 3 Results and Discussion

### 3.1 Changes of aerosols and solar radiation

Figure 1 shows the recent change of aerosol loading and solar radiation in surface and TOA. This study calculates the average AOD from CALIPSO with the scene that the lidar penetrates the entire atmosphere to exclude the effect of unpenetrated profile for lidar. Notably, the aerosol loading increases, and the change of average AOD from CALIPSO (approximately 0.15) is more intense than that of MODIS (approximately 0.05) during monsoon season in India from 2006 to 2015 (Figure 1a). These phenomena are mainly due to the effect of aerosols under clouds, which can be detected by CALIPSO but missed by MODIS (Winker et al., 2010). The low significant level occur in the line trend of AOD detected by MODIS, and mainly result from the disturbance of widespread cloud during monsoon season. Additionally, the quick increase of aerosol in India have been verified by many previous studies (Suresh et al., 2013; Srivastava, 2016). Moreover, aerosols mostly consist of dust, smoke, and polluted dust. These aerosols all present distinct absorbing effects on solar radiation, thereby attenuating the incoming solar radiation more intensively than do other types of aerosols in surface, especially smoke (Logan et al., 2013).

The average SSR in all-sky condition from CERES increases with an increment of 17.0 W $m^{-2}$ during monsoon season in India from 2006 to 2015, and with the partial fluctuation. Moreover, the average incoming solar radiation in TOA was confirmed as stable, whereas the change of OSR shows a negative correlation with the SSR during the study period. These phenomena indicate that the attenuating effect of atmosphere on solar radiation is gradually weakened, thereby the incident solar radiation in surface is increased (brightening). However, there is the fact that the reducing SSR is always occur with increasing aerosol levels (Folini and Wild, 2011). The increase of AOD is paradoxical with the SSR variation, and does not contribute to the recent brightening during monsoon season in India from 2006 to 2015.

### 3.2 Effect of clouds

Figure 2 shows the interannual variations of average cloud vertical frequency distribution in all-sky condition, and cloud microphysical parameters in cloud-sky condition during monsoon season in India from 2006 to 2015. The total distribution of cloud vertical frequency sequentially decreases during monsoon season in India from 2006 to 2015, with a maximum decrement of more than 7.1% (Figure 2a). The cloud water content and number concentration of liquid and ice in cloud-sky condition show a consistently decreasing trend, with maximum decrements of 18.0 mg $m^{-3}$ LWC and 1.4 mg $m^{-3}$ IWC, as well as approximately 6.0 $cm^{-3}$ LNC and 2.0 $L^{-3}$ INC (Figures 2b and 2g). These changes are consistent with the change of cloud vertical frequency during the study period. However, the LER and IER variations are not distinct and are not highly



consistent with the change of cloud vertical frequency (Figures 2c and 2f). Therefore, the clouds decrease with less cloud water content and number concentration, but no clear change of particle effective radius during monsoon season in India from 2006 to 2015.

As shown in Figure 3a, the total cloud fraction detected by CloudSat declines by 9.4% during monsoon season in India from 2006 to 2015, and the change of cloud water path (CWP) in cloud-sky condition is consistent with that of total cloud fraction with a decrement of 53.4 g m$^{-2}$. Moreover, the uppermost cloud top height (CTH) and cloud geometrical depth (CGD) decline by 426 and 182 m during the study period, whereas the fluctuation is extremely small for the lowermost cloud base height (CBH). These phenomena are mainly caused by the rapid decrease in high clouds detected by CloudSat.

We further analyze the spatial variations of cloud fraction, cloud height (CH), CWP, and CGD during monsoon season in India from 2006 to 2015. Figures 3c–3f illustrate the consistent decreases in cloud fraction, CH, CWP, and CGD, especially for the cloud fraction and CWP with the significant consistent change in space. By contrast, increases in CH and CGD are observed in certain parts of the western coastal area in India. In general, we can conclude that the clouds decrease and thin with the decreases in water content and particle number concentration during monsoon season in India from 2006 to 2015, thereby weakening the regulation of cloud on radiation.

Consequently, we investigate the radiative force of cloud vertical variation to discover its possible contribution to the recent rapid brightening (Figure 4). We use the vertically heating rates in all-sky to indicate the change of vertically radiative effect of cloud (Fu and Liou, 1992). Given the change of cloud macrophysical and microphysical characteristics, the SW heat rating decreases by 0.3–0.4 K d$^{-1}$ at an altitude of 5–15 km from 2006 to 2015, whereas a few increase in SW heat rating is located at low cloud area (below 5 km). These phenomena are mainly due to the steadily weakened cloud reflection and enhanced cloud transmission, thereby generating increased SW heat rating in the near surface (Henderson et al., 2013). However, the change of LW vertical heating rate is insignificant and similar to that in SW. For LW, there are radiative heating by less than 0.2 K d$^{-1}$ above 10 km, but cooling by 0.1–0.2 K d$^{-1}$ within the cloud, especially below 5 km. The phenomenon is mainly due to the weakened LW emission in cloud top above 10 km and the decrease in LW absorption emitted from surface (L'Ecuyer et al., 2008). In general, the net vertical heating rate weakens consistently, indicating that the changes of clouds physical characteristics shift the vertical heating and cooling, and weakening the vertical radiative effect of clouds.

In Figure 5a, the SW CRE in surface and TOA weaken by approximately 45.5 and 42.6 W m$^{-2}$, respectively, thereby increasing solar radiation to reach in surface during monsoon season in India from 2006 to 2015. The decrease of SW CRE in surface and TOA is highly consistent with that of total cloud faction (red line in Figure 5a). Although the CRE derived from CloudSat largely ignores the contribution of high thin clouds, this contribution is much smaller than that of low clouds (L'Ecuyer et al., 2008; Henderson et al., 2013). Moreover, it is distinct that SW CRE in surface is negatively correlated with SSR (Figure 5c). The reason is attributed to the spatial variations of the radiative effect in surface caused by the cloud




variables. In general, the cloud is reduced and thinned with the decreased water content and particle number concentration, thereby weakening the SW radiative effect of the clouds, and thus increasing absorption of direct solar radiation in surface and subsequent surface brightening.

### 3.3 Effect of water vapor

In this section, we evaluate the effect of water vapor on the recent brightening in India. We focus mainly on the effect of water vapor on solar radiation in clear-sky condition to eliminate the disturbance of cloud on the evaluation of water vapor in all-sky condition. The vertical relative humidity (RH) in all-sky condition gradually reduces with a maximum decrement of approximately 10% during monsoon season in India from 2006 to 2015 (Figure 6a). Moreover, the change of RH in all-sky is highly consistent with that of clear-sky (Figure 6a and 6b). Figure 6c shows the spatial variation of PW in clear-sky condition during monsoon season in India from 2006 to 2015. PW consistently reduces in space with a maximum decrement of 6.6 mm in India, especially in western and central India. Although a small increase in water vapor (less than 2 mm) occurs in part of the Indian region, its increment is much lesser than the decrease in water vapor in most regions of India.

Figure 7a shows that radiation is highly and negatively correlated with PW from ECMWF-AUX, thereby indicating that water vapor is a substantial controlling factor to the solar radiation variability. This control lies in a direct effect and an indirect effect. The former is its absorption to solar radiation, in which the PW decreases by nearly 3.0 mm over the brightening period. Accordingly, based on the evaluation of the BUGSrad, solar brightening is nearly 2.8 W m$^{-2}$ because of the weakened absorption caused by the decrease of water vapor; the value is much lower than the observed value of cloud variations (45.5 W m$^{-2}$). Note that the SSR in clear-sky from CloudSat have increased about 5.2 W m$^{-2}$. Except for the contribution of water vapor variations this radiative change is possibly caused by the change of atmospheric conditions and incoming solar radiation provided by ECMWF and CloudSat (Yang et al., 2012). The spatial variation of SSR in clear-sky condition is consistent with that of PW, especially in western and central India (Figure 7b). Moreover, in all-sky condition, there are the indirect effects of water vapor, which is water vapor-cloud interaction. Notably, high concentration of water vapor is favorable to generate more or stronger clouds and thus less radiation in surface, and vice versa (Yang et al., 2012). Therefore, the impact of water vapor in all-sky condition is complicated than that in clear-sky condition owing to both the direct and indirect effects of water vapor on solar radiation.

### 4 Conclusions

India is experiencing a leveling-off trend of solar radiation, even a transition from dimming to brightening. This process is significantly complicated because of the active atmospheric action during monsoon season. In this study, we use observations from multiple sensors on A-Train satellite to evaluate the effect of aerosol, cloud, and water vapor variations on recent change of solar radiation in surface during monsoon season (June–September) in India from 2006 to 2015.



We found that the increase in AOD is paradoxical with the SSR variation, and does not contribute to the recent brightening in India. On the contrary, the decreases in water vapor amount and clouds significantly contribute to solar brightening, further contribute to the surface warming in India. In general, the cloud is reduced and thinned by approximately 9.4% and 182 m when water path (by 53.4 g m$^{-2}$) and particle number concentration in cloud-sky condition decrease. The corresponding change of clouds in all-sky condition reduces the SW CRE in surface weakened by approximately 45.5 W m$^{-2}$. Moreover, the PW in clear-sky condition decreases by nearly 3.0 mm over the brightening period. As a result, solar brightening increases by around 2.8 W m$^{-2}$ because of its weakened absorption. The decreases in water vapor amount and clouds weaken the effect of water vapor and clouds on solar radiation, thereby resulting in the increased absorption of direct solar radiation in surface and subsequent surface brightening.

Notably, CloudSat is insufficient to observe thin cirrus with small IWC, thereby resulting in an uncertainty for quantifying the change of clouds. In the globally averaged experiment of radiative fluxes, L'Ecuyer et al. (2008) proposed that the impact of thin cirrus not detected by CloudSat on SW radiation in surface is −1.2 W m$^{-2}$, which is much smaller than the impact of low clouds. Moreover, the data in this study are from the instantaneous observation of multiple sensors; thus, the magnitudes of radiative impact are overestimated for the water vapor and cloud variations. However, the conclusion of this study is significant even when overestimated factors are considered. Given that the variations of aerosols, water vapor, and clouds highly interact with the change of monsoon circulation, it would be interesting to investigate the change mechanisms of aerosols, clouds, and water vapor in India. The chicken-and-egg relationship between monsoon and aerosols, water vapor, and clouds can also be explored.

**Acknowledgments**

This study was supported by the National Science Foundation of China (41627804), National Key Research and Development Program of China (2016YFC0200900), Program for Innovative Research Team in University of Ministry of Education of China (IRT1278), and China Postdoctoral Science Foundation (2016T90731). We would like to thank the CloudSat, CALIPSO, CERES and MODIS science teams for providing excellent and accessible data products that made the study possible. The CloudSat data in this study are acquired from CloudSat Data Processing Center (DPC, http://www.cloudsat.cira.colostate.edu/). The CERES SSF and CALIPSO data are obtained from NASA Langley Research Center Atmospheric Sciences Data Center (ASDC, https://eosweb.larc.nasa.gov/). The MODIS AOD data are obtained from the NASA Earth Observing System Data and Information System, Distributed Active Archive Center (DAAC, https://ladsweb.modaps.eosdis.nasa.gov/).



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





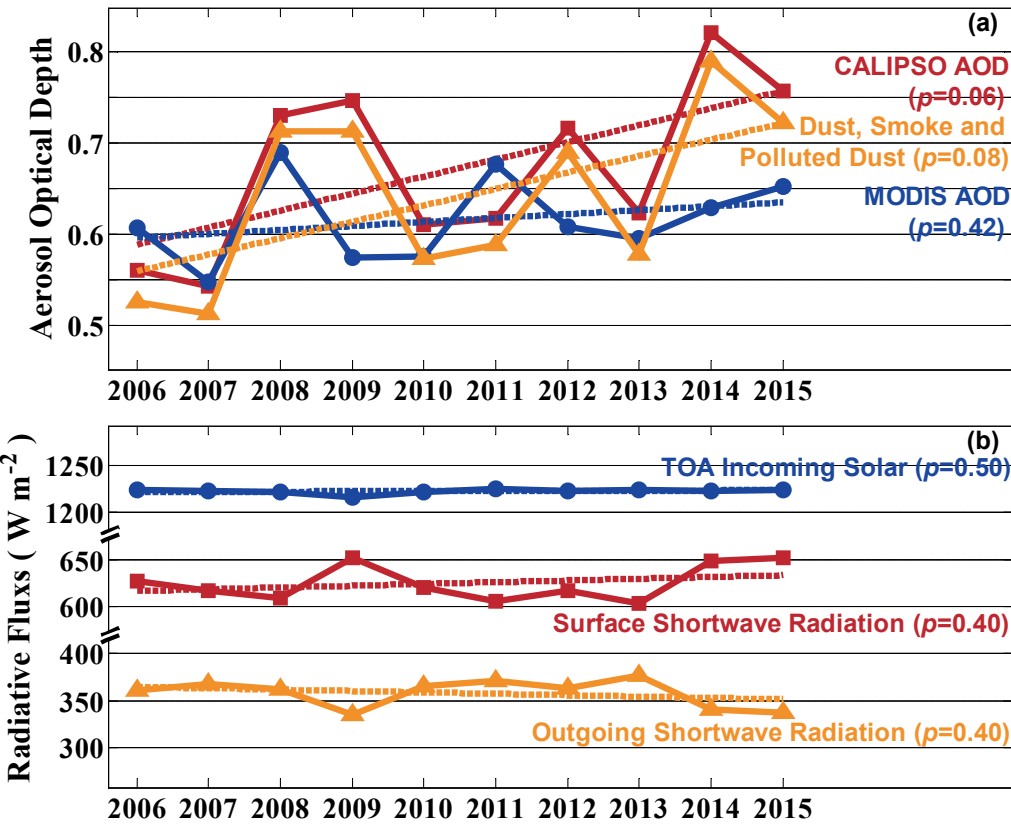

**Figure 1. Interannual variations of average (a) AOD from CALIPSO and MODIS, as well as (b) incoming solar in TOA, SSR, and OSR from CERES during monsoon season in India from 2006 to 2015. The dashed lines indicate the linear fit line of the solid line with same color; *p* is the significance level.**





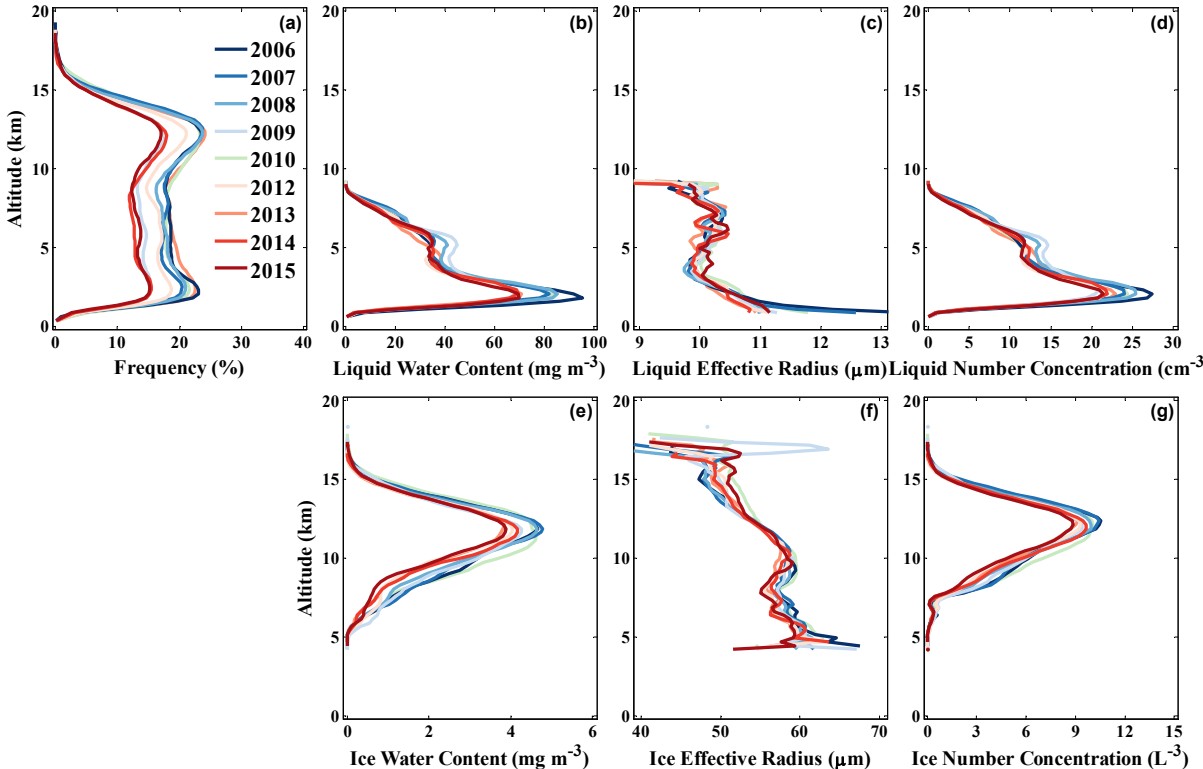

**Figure 2. Interannual variations of average vertical cloud physical parameters during monsoon season in India from 2006 to 2015: (a) cloud vertical frequency distribution, (b) LWC and (e) IWC, (c) LER and (f) IER, (d) LNC and (g) INC.**





**Figure 3.** Interannual variations of spatial average (a) cloud fraction and CWP, and (b) uppermost CTH, lowermost CBH, and CGD; the spatial distribution of the temporal changes of annual average (c) cloud fraction, (d) CH, (e) CWP, and (f) CGD during monsoon season in India from 2006 to 2015. The dashed lines indicate the linear fit line of the solid line with same color; $p$ is the significance level.





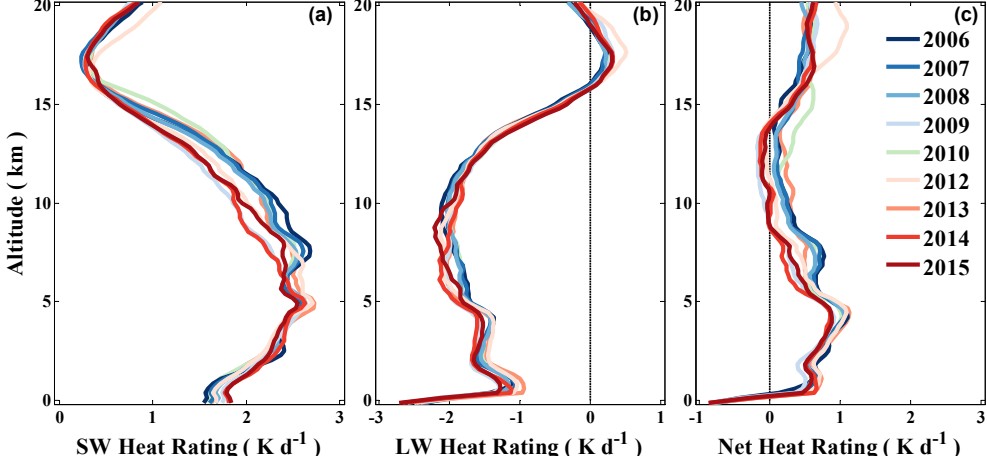

**Figure 4. Interannual variations of average vertical (a) SW, (b) LW, and (c) net heat rating in all-sky during monsoon season in India from 2006 to 2015.**







**Figure 5. Interannual variations of spatial average (a) cloud fraction and CRE, and the spatial distribution of the temporal changes of annual average (b) SW CRE in surface and (c) SSR during monsoon season in India from 2006 to 2015. The dashed lines indicate the linear fit line of the solid line with same color; $p$ is the significance level.**





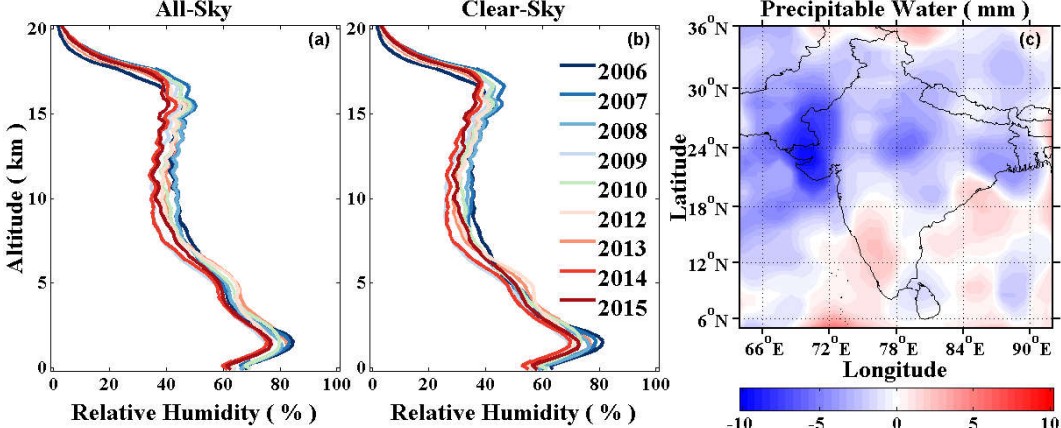

**Figure 6. Interannual variations of average vertical RH in (a) all-sky and (b) clear-sky, and the spatial distribution of the temporal changes of annual average (c) PW in clear-sky during monsoon season in India from 2006 to 2015.**





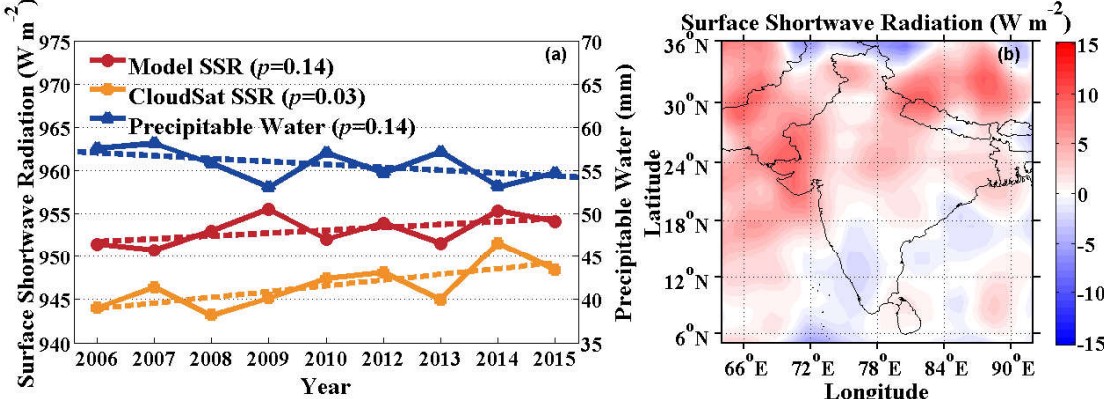

**Figure 7.** Interannual variations of average (a) PW (red line), as well as SSR from BUGSrad (blue line) and CloudSat (orange line), and the spatial distribution of the temporal changes of annual average (b) SSR from BUGSrad in clear-sky during monsoon season in India from 2006 to 2015. The dashed lines indicate the linear fit line of the solid line with same color; *p* is the significance
5  level.