# Peer review of "Estimating the effects of aerosol, cloud, and water vapor on the recent brightening in India during the monsoon season"

_Atmospheric Chemistry and Physics, 2017_

## Referee Comment (RC1) · B. G. Liepert (Referee) · 3 Aug 2017

General Comments:

This manuscript summarizes an interesting study of the intra-decadal variability of surface solar irradiance over India during the monsoon. With observations from the A Train satellite instruments, the authors shown that cloud properties and water vapor and not aerosols play main roles in modifying the shortwave energy budget at the surface from 2006 to 2015. I recommend publication of the manuscript after minor revisions that aim to clarify some of the content and extend the discussion. The issues I would ask the authors to address are as follows: 1) The authors discuss the dimming/brightening

[Figure]

in India during the monsoon period. Since the monsoon is commonly referred to the onset of rain over the Indian subcontinent, I would suggest restricting the data analysis to land only pixels, preferably Indian pixels. As shown in the figures, some of the cloud effects are different over the Bay of Bengal, the Indian Ocean compared to the subcontinent. 2) Albeit not subject of this study, I recommend putting the results into a broader context. This can be done by showing the surface solar radiation trend for the entire year and contrast the monsoon months, or by showing changes in temperatures/rainfall, or similar climate effects that put these interesting results into a broader context.

Specific Comments:

Abstract: The abstract does not reflect the entirety of the study and mentions temperature rises which are not discussed in the text. Some of the interesting results not mentioned are the TOA radiative changes and the atmospheric heating rates. Temperature and rainfall data could be added in the conclusions section to round up the study and be consistent with the abstract. "...increase in aerosol optical depth is paradoxical with the variation of surface shortwave radiation in India...". "Paradoxical" may not be the best choice. How about rephrasing it, for example, "inconsistent". As mentioned above, the figures span a much wider spatial range than "India". Please use a land mask and focus on Indian monsoon region or change the title and text accordingly.

p.2, 26: In the literature, there are discussions about the shifting monsoon onset and end times with global warming. It would be good to explicitly explain the choice of June-Sept time-period as monsoon period. This is also important with respect to the monsoon region as discussed above.

p.5, 11: The SW band is defined from 0 – 4um in the radiative transfer model and <5um in CERES. Can you comment on it?

p.5, 29: Why was annual mean water vapor content used and not monsoon months in BUGSrad?

Fig.1: Are these results for India or the region shown in the consecutive maps?

p.7, 1-15: With these cloud property changes the reader wants to know if these changes are reflected in rainfall data as well. A short discussion would be helpful.

p.7, 15-25: The changes in heating rates should also be mentioned in the abstract.

Technical Corrections:

In the abstract and throughout the text, please replace "in surface" with "at the surface".

p.3, 3: was instead of were.

p.4, 22: Is the bias a monthly bias, and which ARM sites do they refer to?

p.6,20: "However, there is the fact that the reducing SSR is always occur with increasing aerosol levels (Folini and Wild, 2011)." This sentence is not clear and should be rephrased.

---

## Referee Comment (RC2) · Anonymous Referee #2 · 4 Aug 2017

ACP-2017-429
*ESTIMATING THE EFFECTS OF AEROSOL, CLOUD, AND WATER VAPOR ON THE RECENT BRIGHTENING IN INDIA DURING THE MONSOON SEASON*
by Feiyue Mao, Zengxin Pan, Wei Wang, Xin Lu, and Wei Gong

Recommendation: Reject.

This manuscript examines the effect of aerosol, cloud, and water vapor variations on recent change of surface solar radiation and the associated dimming and brightening during monsoon season in India. Multi-year observations from A-Train satellite constellation have been used. The topic is quite interesting and important for the studied region. However, I have quit a few concerns that the authors need to address before it can be published.

My biggest concern is that the authors failed to accurately present and connect the observational findings and tended to draw conclusions without providing further evidences. I listed some of these in the specific comments, e.g., # 22, 26, 27, 30, and so on.

The second one is about the *Abstract*. It should basically give the readers a clear and concise overview of the manuscript in terms of the aim, main points, and conclusions. However, the *Abstract* of this study is just repeating what's been stated in the *Conclusion*. For example, the first few sentences in Lines 12-17 are exactly the same as those in the beginning of *Conclusion*.

My third concern is that the authors need to provide a more thorough literature review about the global and regional change of surface solar radiation and the underlying mechanisms so that readers can have a better context to understand all the discussions followed.

Another one is related to the uncertainty and limitation of data sources. I suggest the authors to rewrite the section of *Data and Methods*. It would be easier to follow if the authors could specify what variables from each dataset have been used after a general description is provided. And please also provide what are the main uncertainties of such variables, either from instrument/model itself or from the retrieval methods. Instead of just providing a reference, the authors need to expand it more including, but not limited to, how those uncertainties affect the current study.

Last but not least, a terrain map showing the study domain should be provided before any spatial-average results are shown. Also figure and panel numbers are missing a lot when the relevant results are discussed, which had made it difficult for me to follow.

Specific comments:
1. Abstract, 13: Add a sentence illustrating why this transition needs to be concerned. In other words, why is this study important?

2. Abstract, 16: Add '*surface*' before '*solar radiation*'.

3. Abstract, 18: Specify this is the spatial- and temporal-average '*increase*'.

4. P2, 37: Add reference to support this statement.

5. P2, 38: Be more specific: was this reduction global or regional and in what season did it occur?

6. P2, 44: What region was this trend observed over?

7. P2, 53: Add reference.

8. P3, 78-79: Specify why only the changes of aerosols, clouds and water vapor are considered. This is the basis of the current study.

9. P4, 90: Add reference.

10. P4, 114: Write out what CERES stands for.

11. P5, 118: Please be more specific and describe how it's relevant to this study.

12. P5, 131-133: This is a good place to state how CALIPSO data is used in the current study and how the authors combine it with other datasets.

13. P5, 138: Add reference.

14. P6, Sec. 2.4: The authors described a radiative transfer model here. But what is more important is how this model is relevant to this study.

15. P7, 173: Add reference for the ECMWF-AUX product.

16. P7, 195: Add reference.

17. P7, 196: Define what the '*all-sky*' condition is.

18. P8, 207: What is the reference for this equation? Are there any uncertainties and limitations to apply it? If the answer is yes, how does it affect this study?

19. P8, 214-216: Be more specific.

20. P8, 219: Figure 1 shows the time series of the spatial-average AOD. A terrain map with the studied domain should be provided first. Also explain in detail how the spatial and temporal average were done. Provide statistical analysis and show how significant the results are. Please do this for the other relevant results too. Also, the orange lines in Figure 1 are not mentioned in the text at all.

21. P9, 226: Explain why aerosols under clouds are missed by MODIS.

22. P9, 235: Add figure panel number from which the observation is made. The significance of the trend needs to be discussed before this statement can be made.

23. P9, 240-243: I don't follow these two sentences. Please rewrite them.

24. P9, 247-249: The '*distribution*' cannot '*decrease*'! What level is this maximum decrease is associated and what does it indicate?

25. P10, 256-258: Is this consistent with previous studies?

26. P10, 265-266: Provide evidence for this statement.

27. P10, 267-274: This paragraph needs to be expanded. What has caused such spatial variation for different cloud properties? Do the spatial patterns change year by year? How significant are the trends shown?

28. P11, 277: Again, how the '*all-sky*' condition is defined?

29. P11, 278-283: How about the trends above 15 km?

30. P10-P11, 281-283 and 286-288: Please provide evidence for these statements.

31. P11, 284: I cannot see the similarity here.

32. P11, 299-300: Expand this with more details.

33. P12, 311-312: This would be better seen if the difference between the two had been plotted.

34. P12, 314-317: What has caused the spatial variation of PW?

35. P12, 330-332: Describe and compare in more details the direct and indirect effects of water vapor.

36. P12, 333: What do you mean by '*stronger clouds*'?

37. Conclusion: This section needs to be expanded. Can the methodology used in this study be applied for those in other regions? How will the findings benefit the representation of surface solar heating in climate models? Are there any potential implications for climate variations?

---

## Referee Comment (RC3) · Anonymous Referee #1 · 8 Aug 2017

The authors examine changes in surface solar radiation (SSR) in the region of India during the monsoon period (June to September) from 2006 to 2015. They relate the SSR changes to changes in aerosol optical depth (AOD), in clouds, and in water vapor. The study relies mostly on satellite products, notably CloudSat, CALIPSO, CERES, and MODIS. The authors find SSR to increase with time, which they ascribe to a decrease in clouds that overcompensate the effect of increasing AOD.

The overall topic of the study - changing SSR in India and its causes - is of interest and suitable for ACP. However, in its present form the study suffers from various short-comings and a general lack of precision, as detailed below under major (and minor)

[Figure]

comments. In particular, the data presented looks, at least in parts, as if it were not a June-September average but rather representative for a specific time of the day. This casts doubt on the entire analysis. Therefore, I recommend rejection of the manuscript in its present form.

**Major comments:**

**1)** SSR magnitude / averaging period. The study deals (see its title) with recent SSR brightening in India. Looking at figure 1b, which shows corresponding key quantities (short-wave radiation at TOA and surface), two thoughts / questions come to mind. First, the radiative fluxes shown are very high, e.g. "TOA Incoming Solar" is around 1230 Wm-2 whereas one would expect for a June mean for India something more on the order of 400 to 500 Wm-2. Why? See also the next two major comments. Second, the SSR trend shown in the figure has a p-value of 0.42, thus the SSR brightening the authors aim to explain is not statistically significant.

**2)** A-train data. A-train has a small footprint, what then is the overpass statistics over the domain of interest? What is the statistical uncertainty of an individual June to September data point, e.g. CloudSat vertical cloud frequency? Is overpass time (A-train equator crossing around 1:30 am and 1:30pm local time) an issue for the presented analysis? The "TOA Incoming Solar" in Figure 1b may suggest so, as its value is around 1230 Wm-2, whereas a June mean for India is more on the order of 400 to 500 Wm-2. If the data is instantaneous, how representative is it for the entire monsoon season?

**3)** Products used. It would be helpful if the authors stated more clearly, in Sections 2 and 3 as well as in the figure captions, which variable they take from which satellite / model product and what the intrinsic uncertainty is of the product and why. Also, it should be discussed to what degree different products (satellite and model) can be inter-compared / combined.

**4)** Season chosen. The authors focus on the monsoon season, June to September, when AOD tends to be lowest over India and cloud cover tends to be particularly pronounced (e.g. Raja et al., Atmos.Env. 142, (2016), p. 238-250; or Nair et al., Clim.Dyn. 49, (2017), p.1411-1428). Why this choice? And how do results for this season compare with, for example, results based on annual means?

**5)** Averaging region. I assume that line plots, as e.g. in Figure 2 or in Figure 3a, are averages taken over the region shown in the maps, e.g. in Figure 3c. If so, the average includes a substantial amount of sea and of the Himalaya. So the averages shown are not actually averages over India. The authors should comment on how this affects their results.

**6)** Statistical significance. While the authors give p-values in some of the figures, they do not address them in the text. Given that many p-values exceed 0.05 or even 0.1 the authors should comment on them. Also, there is no statistical significance given for any of the maps.

**Minor comments:**

**a)** p.6, l.6: What do you mean by "... with the scene that the lidar penetrates the entire atmosphere to exclude the effect of un-penetrated profile for lidar. "?

**b)** p.6, l.10: What do you mean by "The low significant level occur in the line trend of AOD detected by MODIS, and mainly result from the disturbance of widespread cloud during monsoon season. "?

**c)** p.6, l.12: What do you mean by "Moreover, aerosols mostly consist of dust, smoke, and polluted dust. These aerosols all present distinct absorbing effects on solar radiation, thereby attenuating the incoming solar radiation more intensively than do other types of aerosols in the surface. " What other types of aerosols? And why should aerosol mostly consist of dust, smoke (black carbon?), and polluted dust?

**d)** p.7, l.21: "...the change of LW vertical heating rate is insignificant and similar to that in SW." However, for SW you claimed that it is significant. How then can it be similar?

**e)** p.7, l.23: "...but cooling by 0.1 - 0.2 Kd-1 within the cloud..." Looking at Figure 4, cooling seems rather on the order of 1 - 2 Kd-1.

**f)** p.7, l.24: What do you mean by "In general, the net vertical heating rate weakens consistently, indicating that the changes of clouds physical characteristics shift the vertical heating and cooling, and weakening the vertical radiative effect of clouds. "?

**g)** p.8, l.1: "particle number concentration", do you mean "cloud droplet number concentration"?

**h)** Section 3.3: How is clear sky defined? I assume via 'few clouds in the pixel', but a) what is few and b) how many pixels remain / what is the statistics?

**i)** p.8, l.22: "Notably, high concentration of water vapor is favorable to generate more or stronger clouds and thus less radiation in surface and vice versa." One may debate this statement as clouds depend on two parameters: water vapor and temperature. The latter changes as well in your data, I assume, as SSR and heating rates change.

**j)** p.9, l.14: "However, the conclusion of this study is significant even when overestimated factors are considered." Why should this be so?

**k)** Figure 1a: Where does the line for "Dust, Smoke, Polluted Dust" come from? Which satellite / model data?

**l)** Figure 1b: y-axes should read "Radiative Fluxes" (not Fluxs).

**m)** Figure 7a: As for Figure 1b, the values are way too high to be monthly means. Colors in the figure and in the figure caption do not match.

---

## Author Comment (AC1) · 23 Sep 2017

Dr. Zengxin PAN

State Key Laboratory of Information Engineering in Surveying, Mapping and Remote Sensing, Wuhan University, Wuhan 430079, China. Tel: (+86) 15107154425,

Email: pzx@whu.edu.cn

**Manuscript ID:** acp-2017-429 to ACP

**Title:** Estimating the effects of aerosol, cloud, and water vapor on the recent brightening during the South Asian monsoon season

**Author:** Feiyue Mao, Zengxin Pan*, Wei Wang, Xin Lu, Wei Gong.

Dear Anonymous Reviewer 1:

We greatly appreciate for your insightful comments and suggestions. We have revised our paper thoroughly in light of your comments. The changes have been highlighted in color in the manuscript, and also they are summarized below:

1.  We have added the explanation of instantaneous observations at Paragraph 2 Section 2.5;

2.  We have added the comments about significance level of aerosol, cloud, water vapor and related radiative fluxes variations in related explanation of figures;

3.  The details of products used in this study have been added as the Table 1. Moreover, the analysis of uncertainty about the products have added at the introduction of every products in Section 2;

4.  We defined the study domain within the South Asian subcontinent to restrict the effect of the Tibet Plateau and sea pixels on the results of this study (Figure 1);

5.  We have added the reason about focusing on the monsoon season at Paragraph 2 of Section 2.5. Moreover, we have compared the results between monsoon and non-monsoon seasons, and added the related descriptions in Section 3.1.

Please refer to the following point-to-point responses for more details.

Thank you for your time.

Sincerely,

Dr. Zengxin PAN

**General Comments:**

The authors examine changes in surface solar radiation (SSR) in the region of India during the monsoon period (June to September) from 2006 to 2015. They relate the SSR changes to changes in aerosol optical depth (AOD), in clouds, and in water vapor. The study relies mostly on satellite products, notably CloudSat, CALIPSO, CERES, and MODIS. The authors find SSR to increase with time, which they ascribe to a decrease in clouds that overcompensate the effect of increasing AOD.

The overall topic of the study - changing SSR in India and its causes - is of interest and suitable for ACP. However, in its present form the study suffers from various shortcomings and a general lack of precision, as detailed below under major (and minor) comments. In particular, the data presented looks, at least in parts, as if it were not a June-September average but rather representative for a specific time of the day. This casts doubt on the entire analysis. Therefore, I recommend rejection of the manuscript in its present form.

**Major Comments:**

1.  SSR magnitude / averaging period. The study deals (see its title) with recent SSR brightening in India. Looking at figure 1b, which shows corresponding key quantities (short-wave radiation at TOA and surface), two thoughts / questions come to mind. First, the radiative fluxes shown are very high, e.g. "TOA Incoming Solar" is around 1230 W m-2 whereas one would expect for a June mean for India something more on the order of 400 to 500 W m-2. Why? See also the next two major comments. Second, the SSR trend shown in the figure has a p-value of 0.42, thus the SSR brightening the authors aim to explain is not statistically significant.

    **RE:** Thank you for your suggestion.

    (1) Why the radiative fluxes shown are very high?

    In this study, all data indicate observed instantaneous information. We use the data of CloudSat, CALIPSO, MODIS and CERES in daytime. This satellites/sensors cross around equator in daytime at 13:30 local time, at which the incoming solar radiation is almost maximum within the day. Therefore, the instantaneous observation of radiative fluxes provided by CERES are much more than the daily average.

    We have added the corresponding explanation at Paragraph 2 Section 2.5 as

the following:

Additionally, the information about aerosol, cloud, and water vapor is calculated on the basis of instantaneous observations, which we then use to ensure consistent conditions for the observations of multiple satellites. The satellites or sensors of the A-Train cross the equator in the daytime at 13:30 local time, at which point the incoming solar radiation is maximum within the day (Liou, 2002). Therefore, the instantaneous observations of radiative fluxes used in this study are greater than the daily average. The results are more representative of the atmospheric condition variations at nominally 13:30 local time than the daily average.

(2) About the statistically significance.

In this study, the statistically significance ($p$-value) of radiative fluxes can be contributed by many factors: (1) the time series of data in this study is just ten years. The confidence of result in showing the temporal variation is highly and positively correlated with the data span; (2) the observations of polar orbit satellite are significantly affected by the widespread and various cloud in monsoon season.

Actually, we can get the aerosol, cloud and water vapor variations with high significance. The aerosol loading quickly increased in recent decades (Suresh et al., 2013; Srivastava, 2016). Moreover, the recent weakened rainfall and dried monsoon during summer monsoon season have been verified by many previous studies (Bollasina et al., 2011; Turner and Annamalai, 2012; Annamalai et al., 2013). The decreases in water vapor amount and clouds significantly contributed to the brightening, further affecting the surface warming in the South Asia. However, the resultant radiative impact of aerosol, cloud and water vapor variations are complicated and fluctuated during the monsoon season based on the observation of CERES. This phenomenon just confirm the fluctuated and unstable changes in the transition period, which is consistent with the study of Soni et al. (2016).

Furthermore, because the sun is the only significant energy source for the global ecosystem, any change in this precious energy source affects our habitats profoundly (Wild, 2012). The findings of this study is considered suggestive to focus on the recent change of solar radiation and its cause. In addition, this study is a new research perspective for understanding the

brightening during the South Asian monsoon season, and more data will be acquainted and used to enhance the statistically significance in our future study.

2. A-train data. A-train has a small footprint, what then is the overpass statistics over the domain of interest? What is the statistical uncertainty of an individual June to September data point, e.g. CloudSat vertical cloud frequency? Is overpass time (A-train equator crossing around 1:30 am and 1:30 pm local time) an issue for the presented analysis? The "TOA Incoming Solar" in Figure 1b may suggest so, as its value is around 1230 W m-2, whereas a June mean for India is more on the order of 400 to 500 W m-2. If the data is instantaneous, how representative is it for the entire monsoon season?

**RE:** Thank you for your suggestion. We have added the explanation about how to calculate the spatial and temporal average parameter of aerosol, cloud and water vapor and their radiative effect at Paragraph 2 of Section 2.5 as the following:

For the method of spatial and temporal averages, we firstly obtain the spatial distribution of temporal average within a $1° × 1°$ grid during the monsoon season. We then determine the spatial average on the basis of the above temporal average in the entire South Asian subcontinent. For the spatial patterns of temporal changes, we identify the results of the linear temporal changes in atmospheric conditions during the monsoon season from 2006 to 2015.

We have added the corresponding explanation about the uncertainty of CloudSat at Paragraph 2 and 3 in Section 2.1 and 2.5, respectively, as the following:

 **Paragraph 2 in Section 2.1:**
Considering the insensitivity of CloudSat to thin cirrus, Austin et al. (2009) showed that 2B-CWC-RO produces typical biases in IWC of −40% to +25% versus the observationally derived synthetic data. However, L'Ecuyer et al. (2008) proposed that the impact of the thin cirrus that is not detected by CloudSat on shortwave (SW) radiation at the surface is $−1.2$ W m$^{-2}$ in a globally averaged experiment on radiative fluxes; this value is much smaller than the impact value of low clouds.

**Paragraph 3 in Section 2.5:**

We define the cloudy-sk condition for all products as the scenes in which the cloud mask from the CloudSat footprint exceeds 20, and the others are defined as

clear-sky condition. Under this screened method of cloud, the probability of a false detection to CloudSat is less than 5% in comparison with that by CALIPSO (Mace and Zhang, 2014).

Moreover, A-Train satellites cross around equator in daytime at 13:30 local time, at which the incoming solar radiation is maximum within the day. The results in this study indicate the atmospheric variations in nominally constant solar zenith angle. The FIG 1 of the Response show the spatially average solar zenith angle in the moment of satellite observation during monsoon season in the South Asian subcontinent. The standard deviation of average solar zenith angle is mostly less than 0.4, indicating the interannual robustness of solar zenith angle in the moment of satellite observation (FIG 1b). Therefore, the results of this study were insignificantly affected by the solar zenith angle, and are representative for the atmospheric condition variations at nominally constant solar zenith angle.

[Figure]

**FIG 1 of the Response. The spatial distribution of the average (a) solar zenith angle and its standard deviation at the surface in the moment of satellite observation detected by CERES during monsoon season in the South Asian subcontinent from 2006 to 2015.**

We have added the corresponding explanation about the representativeness of the results at Paragraph 3 in Section 4 as the following:

The data in this study were from the instantaneous observations of multiple sensors. Thus, the magnitudes of radiative impact may were overestimated for the water vapor and cloud variations. The results are more representative of the atmospheric condition variations at nominally constant solar zenith angle than the daily average. In sum, the conclusions of this study are beneficial in the

evaluation of the recent variations in aerosol, cloud, and water vapor and their resultant influences on solar radiation.

3. Products used. It would be helpful if the authors stated more clearly, in Sections 2 and 3 as well as in the figure captions, which variable they take from which satellite / model product and what the intrinsic uncertainty is of the product and why. Also, it should be discussed to what degree different products (satellite and model) can be inter-compared / combined.

**RE:** Thank you for your suggestion. We have added the data source of every figure in the figure captions. Also, we have added the corresponding table of products used in this study in Table 1 as the following and highlighted in color:

**Table 1. Observations used in analysis along with their sources and spatial resolutions.**

| Sensor | Spatial Resolution | Products | Parameter |
|--------|--------------------|----------|-----------|
| CloudSat | 1.3 × 1.7 km | 2B-CLDCLASS | Cloud Fraction; cloud top/base height; |
| | | 2B-CWC-RO | Cloud water content, particle number concentration, and particle effective radius |
| | | 2B-FLXHR | Vertical heating rate; SW/LW radiative fluxes |
| CALIPSO | 5 × 5 km | Level 2 Aerosol Layer | Aerosol optical depth; vertical feature mask |
| MODIS | 1° × 1° | Level 3 MYD08 | Aerosol optical depth |
| ECMWF | 2.5° × 2.5° | ECMWF-AUX | Pressure; temperature; relative humidity |
| CERES | 20 × 20 km | Level 2 Single Scanner Footprint | SW radiation in TOA and Surface |

The analysis of intrinsic uncertainty about the products used in this study have added at the introduction of every products in Section 2. Moreover, the combined method of multiple satellite products have been added at Paragraph 1 Section 2.5 as the following:

CloudSat, CALIPSO, and Aqua in A-Train are maintained in a tight orbital coordination within 176 s (Stephens et al., 2017). This setting allows near-simultaneous observations of a wide variety of atmospheric and surface parameters, facilitating the comparison of satellite between each other, allowing for even more comprehensive studies of climate.

4. Season chosen. The authors focus on the monsoon season, June to September, when AOD tends to be lowest over India and cloud cover tends to be particularly pronounced (e.g. Raja et al., Atmos.Env. 142, (2016), p. 238-250; or Nair et al.,

Clim.Dyn. 49, (2017), p.1411-1428). Why this choice? And how do results for this season compare with, for example, results based on annual means?

**RE:** Thank you for your suggestion. We have read the above two paper, and cite them at Paragraph 2 and 4 of Section 3.1, respectively. We add the reason about focusing on the monsoon season in this study at Paragraph 2 of Section 2.5 as the following:

This study focuses on the effect of aerosol, cloud, and water vapor variations during the monsoon season. Due to the increased convective activity, the changes in aerosol, cloud, and water vapor and their impact on solar radiation possibly occur more obviously during the monsoon season than during non-monsoon seasons (Turner and Annamalai, 2012).

Moreover, we have compared the results between monsoon season and non-monsoon season, and added the description about these results in Section 3.1 as the following:

Figure 2 shows the recent changes in aerosol loading during different seasons, namely, pre-monsoon, monsoon, and dry seasons, respectively. These three seasons occur in the periods of March–May, June–September, and October–February, respectively. The aerosol loading increased, and the change in the average AOD detected by CALIPSO (approximately 0.13) was more intense than that detected by MODIS (approximately 0.05) during the monsoon season in the South Asian subcontinent from 2006 to 2015 (Figure 2a). These phenomena are mainly due to the effect of aerosols under clouds, which can be detected by CALIPSO but missed by MODIS because of MODIS only inferring the aerosol loading under clear-sky condition (Winker et al., 2010). Moreover, CALIPSO can identify aerosol and cloud layer with ~90% accuracy. The sum of dust, smoke, and polluted dust aerosol loadings is close to the total aerosol loading (orange line in Figure 2a). This phenomenon indicates that aerosols mostly consist of dust, smoke, and polluted dust during the monsoon season in the South Asian subcontinent, which is consistent with the previous study (Das et al., 2015). These aerosols all present distinct absorbing effects on solar radiation, thereby attenuating the incoming solar radiation more intensively than other types of aerosols (Logan et al., 2013).

Figure 2b shows the consistent increase in AOD during the pre-monsoon, monsoon, and dry seasons, respectively. The increment of AOD during the

pre-monsoon and dry seasons (by 0.08 and 0.11, respectively) were greater than those during the monsoon season according to the MODIS observations from 2006 to 2015. The significant level of AOD variation detected by MODIS was higher during non-monsoon season ($p<0.01$) than during the monsoon season, which mainly due to the disturbance of widespread clouds during the monsoon season (Remer et al., 2005). The change in the average AOD detected by CALIPSO showed a higher significance level ($p=0.07$) than that detected by MODIS due to the minimal disturbance of thin clouds and surface albedos to the aerosols detected by CALIPSO during the monsoon season (Redemann et al., 2012). Additionally, the robust and quick increase in aerosols in the South Asia has been verified by many previous studies (Suresh et al., 2013; Reddy et al., 2016; Srivastava, 2016).

[Figure]

**Figure 2. Temporal variations in spatial average AOD from (a) CALIPSO and MODIS during the monsoon season, as well as (b) MODIS during pre-monsoon, monsoon, and dry seasons in the South Asian subcontinent from 2006 to 2015, respectively. The dashed lines indicate the linear fit line of the solid line with the same color; p is the significance level; k is the slope of the line trend of each time series.**

As shown in Figure 3a, the average SSR (red line) in all-sky condition from CERES increased by 16.2 W m$^{-2}$ increments during the monsoon season in the South Asian subcontinent from 2006 to 2015. Moreover, the change in OSR (by

13.0 W m$^{-2}$) showed a highly negative correlation with that in the SSR during the study period (Figure 3b). These phenomena indicate that the attenuating effect of the atmosphere on solar radiation is gradually weakened, thereby increasing the incident solar radiation at the surface (brightening). There is the fact that the SSR is highly negatively correlated with aerosol loading (Folini and Wild, 2011). Therefore, the increase in AOD did not contribute to the recent brightening during the monsoon season in the South Asian subcontinent from 2006 to 2015.

Further, Figure 3a shows no distinct change in the SSR during the pre-monsoon season, even the reverse trend of SSR during the dry season (dimming) comparing to that during the monsoon season. The seasonal differences in OSR variations are similar to those of SSR (Figure 3b). Moreover, the trend of SSR during the dry season presented a higher significance level ($p<0.01$) than that during the monsoon season ($p=0.40$). Many reasons can contribute to the significance levels during different seasons. The average cloud fractions during the pre-monsoon and dry seasons are both approximately 20%, and much lower than that during the monsoon season (53.4%) according to CloudSat observations. Moreover, under clear-sky condition, aerosols control the change in the solar radiation that reaches the surface (Nair et al., 2016; Soni et al., 2016). Therefore, aerosols have greater weighted impact on solar radiation during the dry season than during the monsoon season. In addition, the resultant effects of aerosol, cloud, and water vapor are considerably complicated and varied due to the intensive atmospheric activities during the monsoon season. The pre-monsoon season is the transition season from the dry season to the monsoon season, and associated with the establishment of the southwestern wind regime over the South Asian subcontinent (Das et al., 2015). The atmospheric condition and radiation variations are not distinct comparing to the trend of SSR between the dry and monsoon seasons.

[Figure]

**Figure 3. Temporal variations in spatial average (a) SSR and (b) OSR from CERES during pre-monsoon, monsoon, and dry seasons in the South Asian subcontinent from 2006 to 2015, respectively. The dashed lines indicate the linear fit line of the solid line with the same color; p is the significance level; k is the slope of the line trend of each time series.**

5.  Averaging region. I assume that line plots, as e.g. in Figure 2 or in Figure 3a, are averages taken over the region shown in the maps, e.g. in Figure 3c. If so, the average includes a substantial amount of sea and of the Himalaya. So the averages shown are not actually averages over India. The authors should comment on how this affects their results.

    **RE:** Thank you for your suggestion. We defined the study domain within the South Asian subcontinent to restrict the effect of the Tibet Plateau and sea pixels on the results of this study, as shown in Figure 1. We have added the terrain map and showed the study region of this study in Figure 1 as the following:

[Figure]

**Figure 1. Terrain map during the monsoon season in the South Asia and surrounding regions. The blue line indicates rivers; the study region is within the red line.**

6. Statistical significance. While the authors give p-values in some of the figures, they do not address them in the text. Given that many p-values exceed 0.05 or even 0.1 the authors should comment on them. Also, there is no statistical significance given for any of the maps.

**RE:** Thank you for your suggestion. We have address the p-values in corresponding location in the text, and added comments about the significant level of relevant results. For example, the comment about shortwave surface radiation have been added at Paragraph 3 of Section 3.1 as the following:

Figure 2b shows the consistent increase in AOD during the pre-monsoon, monsoon, and dry seasons, respectively. The increment of AOD during the pre-monsoon and dry seasons (by 0.08 and 0.11, respectively) were greater than those during the monsoon season according to the MODIS observations from 2006 to 2015. The significant level of AOD variation detected by MODIS was higher during non-monsoon season ($p<0.01$) than during the monsoon season, which mainly due to the disturbance of widespread clouds during the monsoon season (Remer et al., 2005). The change in the average AOD detected by

CALIPSO showed a higher significance level (*p*=0.07) than that detected by MODIS due to the minimal disturbance of thin clouds and surface albedos to the aerosols detected by CALIPSO during the monsoon season (Redemann et al., 2012). Additionally, the robust and quick increase in aerosols in the South Asia has been verified by many previous studies (Suresh et al., 2013; Reddy et al., 2016; Srivastava, 2016).

The aerosol, cloud and water vapor variations is correlated with regional sources and meteorology. Consequently, solar dimming and brightening may be of local or regional nature, and is unavoidably inhomogeneous in space (Soni et al., 2016). Moreover, this study mainly focus on the total change of solar radiation at the surface and its significant reasons, not the relevant regional distribution. The recent transition from dimming to brightening is unstable and fluctuant in the South Asian subcontinent. However, due the crucial role of incident solar radiation on surface energy balance, any change with regional and global scale is worthy of our deep concern. We have added the comment of the solar radiation variations at the surface in space at Paragraph 3 of Section 4 as the following:

Additionally, the aerosol, cloud, and water vapor variations are correlated with regional sources and meteorology. Consequently, solar dimming and brightening may be of local or regional nature, and they are unavoidably inhomogeneous and unstable in space (Soni et al., 2016).

**Minor comments:**

1. p.6, l.6: What do you mean by "... with the scene that the lidar penetrates the entire atmosphere to exclude the effect of un-penetrated profile for lidar. "?

   **RE:** Thank you for your caution. Aerosol is mostly located at near-surface and the base of cloud. However, it is difficulty for CALIPSO lidar to penetrate the optically thick layer of optical depth exceeding 3-5 (Hu et al., 2009). In this case, CALIPSO many miss aerosol at near-surface due to the entirely attenuation of lidar signal. These case possible generate that the calculated AOD from CALISPO is less than the actual value. Therefore, we remove the unpenetrated profile for lidar before the calculation of AOD. We have added the corresponding explanation at Paragraph 1 Section 2.2 as the following:

   CALIPSO is highly sensitive to thin cirrus clouds, but cannot easily penetrate optically thick layers with optical depths exceeding 3–5 (Hu et al., 2009). In this case, CALIPSO determines no or low aerosol optical depth (AOD) at the

near-surface due to the complete attenuation of lidar signals. To exclude the effects of unpenetrated profiles for lidar, we calculate the average AOD from CALIPSO only on the basis of the scene in which the lidar penetrates the entire atmosphere.

2. p.6, l.10: What do you mean by "The low significant level occur in the line trend of AOD detected by MODIS, and mainly result from the disturbance of widespread cloud during monsoon season. "?

**RE:** Thank you for your comment. The quick increase of aerosol in the South Asian subcontinent have been verified by many previous studies (Suresh et al., 2013; Reddy et al., 2016; Srivastava, 2016). The annual average cloud fraction during monsoon season is 53.4%, is more than twice of that during non-monsoon (about 20%) based no CloudSat observations. However, due to the impact of widespread cloud during monsoon season, the AOD variations show the lower significance level than that during non-monsoon, especially low broken cloud and thin cirrus (Mao et al., 2015). We have revised the corresponding description at Paragraph 1 of Section 3.1 as the following:

The significant level of AOD variation detected by MODIS was higher during non-monsoon season ($p<0.01$) than during the monsoon season, which mainly due to the disturbance of widespread clouds during the monsoon season (Remer et al., 2005).

3. p.6, l.12: What do you mean by "Moreover, aerosols mostly consist of dust, smoke, and polluted dust? These aerosols all present distinct absorbing effects on solar radiation, thereby attenuating the incoming solar radiation more intensively than do other types of aerosols in the surface. "What other types of aerosols? And why should aerosol mostly consist of dust, smoke (black carbon?), and polluted dust?

**RE:** Thank you for your suggestion. CALIPSO classified the aerosols into six sub-types with high accurate, including clean marine, dust, polluted continental, clean continental, polluted dust and smoke. We have revised the corresponding description at Paragraph 1 of Section 3.1 as the following:

Moreover, CALIPSO can identify aerosol and cloud layer with ~90% accuracy. The sum of dust, smoke, and polluted dust aerosol loadings is close to the total aerosol loading (orange line in Figure 2a). This phenomenon indicates that aerosols mostly consist of dust, smoke, and polluted dust during the monsoon

season in the South Asian subcontinent, which is consistent with the previous study (Das et al., 2015).

4. p.7, l.21: "...the change of LW vertical heating rate is insignificant and similar to that in SW." However, for SW you claimed that it is significant. How then can it be similar?

   **RE:** Thank you for your suggestion. I am sorry for the ambiguity. We have revised the corresponding description at Paragraph 4 of Section 3.2 as the following:

   However, the change in LW vertical heating rate was not distinct compared with that of SW.

5. p.7, l.23: "...but cooling by 0.1 - 0.2 Kd-1 within the cloud..." Looking at Figure 4, cooling seems rather on the order of 1 - 2 Kd-1.

   **RE:** Thank you for your suggestion. We have revised the corresponding description at Paragraph 4 of Section 3.2 as the following:

   Correspondingly, the LW radiative heating increased by less than $0.2$ K $d^{-1}$ above 10 km, whereas that of heating decreased by $0.1$–$0.2$ K $d^{-1}$ within the cloud, especially below 4 km, from 2006 to 2015.

6. p.7, l.24: What do you mean by "In general, the net vertical heating rate weakens consistently, indicating that the changes of clouds physical characteristics shift the vertical heating and cooling, and weakening the vertical radiative effect of clouds. "?

   **RE:** Thank you for your suggestion. We have revised the corresponding description at Paragraph 4 of Section 3.2 as the following:

   In general, the net vertical heating rate weakened consistently, indicating that the changes in the clouds physical characteristics weakened their vertical radiative effect of clouds.

7. p.8, l.1: "particle number concentration", do you mean "cloud droplet number concentration"?

   **RE:** Thank you for your suggestion. You are right. The "particle number concentration" have been revised to "cloud droplet number concentration"

8. Section 3.3: How is clear sky defined? I assume via 'few clouds in the pixel', but a) what is few and b) how many pixels remain / what is the statistics?

**RE:** Thank you for your suggestion. We have defined the clear-sky condition at Paragraph 2 of Section 2.5 as the following:

We define the cloudy-sk condition for all products as the scenes in which the cloud mask from the CloudSat footprint exceeds 20, and the others are defined as clear-sky condition. Under this screened method of cloud, the probability of a false detection to CloudSat is less than 5% in comparison with that by CALIPSO (Mace and Zhang, 2014).

9. p.8, l.22: "Notably, high concentration of water vapor is favorable to generate more or stronger clouds and thus less radiation in surface and vice versa." One may debate this statement as clouds depend on two parameters: water vapor and temperature. The latter changes as well in your data, I assume, as SSR and heating rates change.

**RE:** Thank you for your suggestion. With the increase of SSR, the atmospheric temperature generally increased consistently. Similarly, high concentration of water vapor is favorable to generate more or stronger clouds comparing to the low concentration of water vapor. However, the resultant influence of atmospheric condition variations on solar radiation is complicated. In this study, we mainly focus on the impact of water vapor variation on solar radiation. We have revised the corresponding description at Paragraph 2 of Section 3.3 as the following:

Generally, aside from the effect of temperature, a high concentration of water vapor favors the generation of more or stronger convective clouds and results in decreased radiation at the surface, and vice versa (Yang et al., 2012).

10. p.9, l.14: "However, the conclusion of this study is significant even when overestimated factors are considered." Why should this be so?

**RE:** Thank you for your suggestion. We have revised the corresponding description at Paragraph 3 of Section 4 as the following:

In sum, the conclusions of this study are beneficial in the evaluation of the recent variations in aerosol, cloud, and water vapor and their resultant influences on solar radiation.

11. Figure 1a: Where does the line for "Dust, Smoke, Polluted Dust" come from? Which satellite / model data?

**RE:** Thank you for your suggestion. CALIPSO can identify the sub-types of aerosols, including clean marine, dust, polluted continental, clean continental, polluted dust and smoke. The sum of dust, smoke and polluted dust aerosol loading is calculated by CALIPSO data. We have revised the corresponding description at Paragraph 1 of Section 2.2 as the following:

Moreover, aerosols detected by CALIOP can be classified into the following six sub-types with high accuracy: clean marine, dust, polluted continental, clean continental, polluted dust, and smoke (Omar et al., 2009).

12. Figure 1b: y-axes should read "Radiative Fluxes" (not Fluxs).

    **RE:** Thank you for your suggestion. We have revised "Fluxs" to "Fluxes".

13. Figure 7a: As for Figure 1b, the values are way too high to be monthly means. Colors in the figure and in the figure caption do not match.

    **RE:** Thank you for your suggestion. The explanation about instantaneous and daily average radiation fluxes is provided in Question 1. Moreover, we have matched the colors in the figure and in the figure caption.

**References**

Annamalai, H., Hafner, J., Sooraj, K. P., and Pillai, P.: Global Warming Shifts the Monsoon Circulation, Drying South Asia, Journal of Climate, 26, 2701-2718, 2013.

Austin, R. T., Heymsfield, A. J., and Stephens, G. L.: Retrieval of ice cloud microphysical parameters using the CloudSat millimeter-wave radar and temperature, Journal of Geophysical Research Atmospheres, 114, 1065-1066, 2009.

Bollasina, M. A., Ming, Y., and Ramaswamy, V.: Anthropogenic Aerosols and the Weakening of the South Asian Summer Monsoon, Science, 334, 502-505, 2011.

Das, S., Dey, S., Dash, S. K., Giuliani, G., and Solmon, F.: Dust aerosol feedback on the Indian summer monsoon: Sensitivity to absorption property, Journal of Geophysical Research Atmospheres, 120, 9642-9652, 2015.

Folini, D., and Wild, M.: Aerosol emissions and dimming/brightening in Europe: Sensitivity studies with ECHAM5-HAM, Journal of Geophysical Research Atmospheres, 116, 21104, 2011.

Hu, Y., Winker, D., Vaughan, M., Lin, B., Omar, A., Trepte, C., Flittner, D., Yang, P., Nasiri, S. L., and Baum, B.: CALIPSO/CALIOP Cloud Phase Discrimination Algorithm, Journal of Atmospheric & Oceanic Technology, 26, 2293-2309, 2009.

L'Ecuyer, T. S., Wood, N. B., Haladay, T., Stephens, G. L., and Stackhouse, P. W.: Impact of clouds on atmospheric heating based on the R04 CloudSat fluxes and heating rates data set, Journal of Geophysical Research Atmospheres, 113, 2013-2018, 2008.

Liou, K.-N.: An Introduction to Atmospheric Radiation, Academic press, San Diego, USA, 2002.

Logan, T., Xi, B., Dong, X., Li, Z., and Cribb, M.: Classification and investigation of Asian aerosol absorptive properties, Atmospheric Chemistry and Physics, 13, 2253-2265, 2013.

Mace, G. G., and Zhang, Q.: The CloudSat radar-lidar geometrical profile product (RL-GeoProf): Updates, improvements, and selected results, Journal of Geophysical Research: Atmospheres, 119, 9441-9462, 2014.

Mao, F., Duan, M., Min, Q., Gong, W., Pan, Z., and Liu, G.: Investigating the impact of haze on MODIS cloud detection, Journal of Geophysical Research: Atmospheres, 120, 12237-12247, 10.1002/2015JD023555, 2015.

Nair, V. S., Babu, S. S., Manoj, M. R., Moorthy, K. K., and Chin, M.: Direct radiative effects of aerosols over South Asia from observations and modeling, Climate Dynamics, 1-18, 2016.

Omar, A. H., Winker, D. M., Kittaka, C., Vaughan, M. A., Liu, Z., Hu, Y., Trepte, C. R., Rogers, R. R., Ferrare, R. A., and Lee, K.-P.: The CALIPSO Automated Aerosol Classification and Lidar Ratio Selection Algorithm, Journal of Atmospheric and Oceanic Technology, 26, 1994, 2009.

Reddy, K. R. O., Balakrishnaiah, G., Gopal, K. R., Reddy, N. S. K., Rao, T. C., Reddy, T. L., Hussain, S. N., Reddy, M. V., Reddy, R. R., and Boreddy, S. K. R.: Long term (2007–2013) observations of columnar aerosol optical properties and retrieved size distributions over Anantapur, India using multi wavelength solar radiometer, Atmospheric Environment, 142, 238-250, 2016.

Redemann, J., Vaughan, M. A., Zhang, Q., Shinozuka, Y., Russell, P. B., Livingston, J. M., Kacenelenbogen, M., and Remer, L. A.: The comparison of MODIS-Aqua (C5) and CALIOP (V2 & V3) aerosol optical depth, Atmospheric Chemistry and Physics, 12, 3025-3043, 2012.

Remer, L. A., Kaufman, Y. J., Tanré, D., Mattoo, S., Chu, D. A., Martins, J. V., Li, R. R., Ichoku, C., Levy, R. C., and Kleidman, R. G.: The MODIS Aerosol Algorithm, Products, and Validation, Journal of Atmospheric Sciences, 62, 947-973, 2005.

Soni, V. K., Pandithurai, G., and Pai, D. S.: Is there a transition of solar radiation from dimming to

brightening over India?, Atmospheric Research, 169, 209-224, 2016.

Srivastava, R.: Trends in aerosol optical properties over South Asia, International Journal of Climatology, 37, n/a-n/a, 2016.

Stephens, G., Winker, D., Pelon, J., Trepte, C., Vane, D., Yuhas, C., L'Ecuyer, T., and Lebsock, M.: CloudSat and CALIPSO within the A-Train: Ten years of actively observing the Earth system, Bulletin of the American Meteorological Society, 0, null, 10.1175/bams-d-16-0324.1, 2017.

Suresh, B. S., Manoj, M. R., Krishna, M. K., Gogoi, M. M., Nair, V. S., Kumar, K. S., Satheesh, S. K., Niranjan, K., Ramagopal, K., and Bhuyan, P. K.: Trends in aerosol optical depth over Indian region: Potential causes and impact indicators, Journal of Geophysical Research Atmospheres, 118, 11,794–711,806, 2013.

Turner, A. G., and Annamalai, H.: Climate change and the South Asian summer monsoon, Nature Climate Change, 2, 587-595, 2012.

Wild, M.: Enlightening Global Dimming and Brightening, Bulletin of the American Meteorological Society, 93, 27-37, 2012.

Winker, D. M., Pelon, J., Coakley, J. A., Ackerman, S. A., Charlson, R. J., Colarco, P. R., Flamant, P., Fu, Q., Hoff, R. M., and Kittaka, C.: The CALIPSO Mission: A Global 3D View of Aerosols and Clouds, Bulletin of the American Meteorological Society, 91, 1211-1229, 2010.

Yang, K., Ding, B., Qin, J., Tang, W., Ning, L., and Lin, C.: Can aerosol loading explain the solar dimming over the Tibetan Plateau, Geophysical Research Letters, 39, L20710, 2012.

**Estimating the effects of aerosol, cloud, and water vapor on the recent brightening during the South Asian monsoon season**

Feiyue Mao [1, 2, 3], Zengxin Pan [2, *], Wei Wang [2], Xin Lu [2], Wei Gong [2, 3]

[1] School of Remote Sensing and Information Engineering, Wuhan University, Wuhan 430079, China
[2] State Key Laboratory of Information Engineering in Surveying, Mapping and Remote Sensing, Wuhan University, Wuhan 430079, China
[3] Collaborative Innovation Center for Geospatial Technology, Wuhan 430079, China

*Correspondence to*: Zengxin Pan (pzx@whu.edu.cn)

**Abstract.** South Asia is experiencing a leveling-off trend of solar radiation and even a transition from dimming to brightening. As the only significant energy source for the global ecosystem, any change in incident solar radiation affects our habitats profoundly. This process is significantly complicated because of the active atmospheric action during the monsoon season. Here, we use observations from multiple sensors in the A-Train satellite constellation to evaluate the effects of aerosol, cloud, and water vapor variations on the recent changes in surface solar radiation during the monsoon season (June–September) in the South Asian subcontinent from 2006 to 2015. Results show that during this period, the surface shortwave radiation (SSR) and outgoing shortwave (SW) radiation increased by 16.2 W m$^{-2}$ and decreased by 13.0 W m$^{-2}$, respectively. However, the increase in spatial average aerosol optical depth is inconsistent with the variation in SSR in the South Asian subcontinent. Instead, the decreases in the amount of water vapor and clouds significantly contributed to brightening, thus further affecting the surface warming in the South Asian subcontinent. Clouds generally reduced and thinned by approximately 9.4% and 182 m, respectively, with the decrease in cloud water path (by 53.4 g m$^{-2}$) and particle number concentration under cloud-sky condition. Given the change in the clouds, the atmospheric vertical SW heat rating decreased by 0.3–0.4 K d$^{-1}$ at an altitude of 5–15 km, whereas the SW heat rating slightly increased at the low-cloud and near-surface areas (below 4 km) from 2006 to 2015. The SW cloud radiative effect decreased by approximately 45.5 W m$^{-2}$ at the surface. Moreover, the precipitable water under clear-sky condition decreased by approximately 2.8 mm over the brightening period. Correspondingly, solar brightening increased by roughly 2.5 W m$^{-2}$ owing to the weakened absorption. Overall, the decreases in water vapor and clouds resulted in the increased absorption of direct solar radiation at the surface and subsequent surface brightening. Hence, brightening may play a prominent role in modulating the warming and rainfall variations in the South Asian subcontinent.

**1 Transition from Dimming to Brightening in South Asia**

Solar radiation incidence on the Earth's surface plays a fundamental determinant role in climate and life on our planet (IPCC, 2013). Surface solar radiation is a major component of the surface energy balance and governs many diverse surface processes, such as evaporation and associated hydrological components, snow and glacier melting, plant photosynthesis and related terrestrial carbon uptake, as well as the diurnal and seasonal courses of surface temperatures. Negative trends in the downwelling surface solar radiation are collectively called "dimming," whereas positive trends are called "brightening" (Wild et al., 2005). Any change in the amount of solar radiation profoundly affects the temperature field, atmospheric and oceanic general circulation, and hydrological cycle (Haywood et al., 2011).

Widespread reduction in the annual average surface solar radiation from the 1960s to the 1980s has been reported by many researchers at the global and regional scales, including those from America, Europe, and China (Liepert, 2002; Wild et al., 2005). Subsequently, the term "brightening" was coined to emphasize that global solar radiation no longer declines at many sites after the late 1980s (Wild et al., 2005). Long et al. (2009) found that solar dimming reversed at an increasing trend of 6 W m$^{-2}$ per decade in the continental United States in 1995–2007. Wild (2009) presented that the globally averaged trends in the 1980s typically reversed from dimming to brightening; the study reported trends of 2.2–6.6 W m$^{-2}$ per decade from the 1980s to the 2000s. However, recent studies indicate that the developments in dimming and brightening after 2000 show mixed tendencies. Wild et al. (2009) reported a continuation of brightening at sites in Europe, United States, and parts of Asia, a leveling-off at sites in Japan and Antarctica, and indications of renewed dimming in China. Conversely, the most recent related study shows that brightening has been continuing in China after 2000 (Wang and Wild, 2016).

South Asia is endowed with abundant solar energy because of its geographic position in the tropical belt (Soni et al., 2016). Dimming or brightening in South Asia is more evident and complicated than that in other regions, and its effects on regional and global climate and ecosystem are amplified by the monsoon circulation in the country (Padma et al., 2007; Wild, 2012). The South Asian monsoon occurs between June and September, which correspond to the monsoon onset and end times, respectively (Turner and Annamalai, 2012). Contrary to the variable trend in surface solar radiation in other regions globally, Padma et al. (2007) found that continued dimming of −8.6 W m$^{-2}$ per decade occurred in India from 1981 to 2004 according to 12 stations over the Indian region. Wild et al. (2009) proposed to focus on the slight tendency toward the stabilization of surface solar radiation since the late 1990s. Furthermore, Soni et al. (2016) observed a trend reversal and partial recovery from dimming to brightening in India around 2001.

Various mechanisms can potentially contribute to dimming and brightening. Changes in surface solar radiation can be caused by either external changes in the amount of solar radiation incident on the planet at the top of atmosphere (TOA) or internal changes (within the climate system) in the transparency of the atmosphere, which modifies the solar beam on its way to the Earth's surface. However, Willson and Mordvinov (2003) verified that the decadal dimming or brightening cannot be explained by changes in the luminosity of the sun because these changes are at least an order of magnitude smaller than

those in climate system. Therefore, the observed dimming or brightening has to originate from alterations in the transparency of the atmosphere, which depends on the presence of clouds, aerosols, and radiatively active gases, particularly water vapor, which is a strong absorber of solar radiation (Kim and Ramanathan, 2008).

Kvalevåg and Myhre (2007) concluded that the major contributor to dimming is aerosols ($-2.4$ W m$^{-2}$) and that the secondary effect is the increase in gas concentrations ($-0.64$ W m$^{-2}$), including those of tropospheric ozone and water vapor, since pre-industrial times; they also identified $NO_2$, $CH_4$, and $CO_2$ as minor contributors. Many studies have proposed that the changes in atmospheric aerosol loading, cloud cover, and cloud properties are the main factors that determine solar dimming and brightening (Wild, 2009; Soni et al., 2016). Liepert (2002) showed that the decrease in global radiation in 1961–1990 is attributable to the increases in cloud optical thickness and the direct effects of aerosols, which reduced solar radiation by 18 and 8 W m$^{-2}$, respectively. Kambezidis et al. (2012) argued that the decline in surface solar radiation in South Asia is attributable to the increase in the amount of anthropogenic aerosols during the last 30 years of the 20th century. This deduction is supported by the agreement between the observed decadal changes in anthropogenic aerosol emissions and trends in global solar radiation (Wild, 2009; Folini and Wild, 2011). Moreover, solar dimming and brightening may be of local or regional nature and is unavoidably influenced by regional sources and meteorology (Soni et al., 2016). Here, we focus mainly on the dimming and brightening caused by changes in aerosols, water vapor, and clouds, which are the dominant factors that alter atmospheric transparency, and thereby regulate the solar radiation incident on the Earth's surface (Wild, 2009).

In this study, we use observations from multiple sensors in the A-Train satellite constellation to evaluate the effects of aerosol, cloud, and water vapor variations on the recent changes in solar radiation at the surface during the monsoon season (June–September) in the South Asian subcontinent from 2006 to 2015. This study mainly aims to identify the possible reasons for the changes in dimming and brightening and assess the connection between the variations in aerosols, clouds, and water vapor and the recent brightening in the South Asian subcontinent.

**2 Data and Methods**

**2.1 CloudSat data**

Cloud Profile Radar (CPR) is an active millimeter-wave radar and is the only instrument on CloudSat, which was launched into the A-Train constellation in April 2006 (Mace et al., 2009). CPR has a 1.3 km cross-track and a 1.7 km along-track footprint resolution, and its effective vertical resolution at nadir is 240 m (Pan et al., 2017). CloudSat CPR is well suited for sensing a wide variety of cloud systems, from cirrus and stratus to deep convective systems, and shows slight sensitivity to the time of day or season (Rajeevan et al., 2012). However, the estimated operational sensitivity of CloudSat CPR ($-32$ dBZ to $-30$ dBZ) is insufficient for observing thin cirrus clouds with low ice water content (IWC) (Mace et al., 2009). CloudSat can infer the cloud microphysical characteristics on the basis of the backscatter return signals measured by

CloudSat CPR, including cloud particle number concentration, size, shape, and phase (Heymsfield et al., 2010). Further, by combining with a broadband radiative flux model known as BUGSRad, CloudSat can be used to quantify the three-dimensional (3D) information of cloud macrophysical and microphysical characteristics, as well as the corresponding cloud radiative effect (CRE) (L'Ecuyer et al., 2008).

5    The CloudSat Radar-Only Cloud Water Content (2B-CWC-RO) product contains retrieved estimates of cloud liquid water content (LWC) and IWC, cloud liquid number concentration (LNC) and ice particle number concentration (INC), cloud liquid effective radius (LER) and ice particle effective radius (IER), and related quantities for each radar profile measured by CPR on CloudSat (Woods et al., 2008). Considering the insensitivity of CloudSat to thin cirrus, Austin et al. (2009) showed that 2B-CWC-RO produces typical biases in IWC of −40% to +25% versus the observationally derived

10   synthetic data. However, L'Ecuyer et al. (2008) proposed that the impact of the thin cirrus that is not detected by CloudSat on shortwave (SW) radiation at the surface is −1.2 W m$^{-2}$ in a globally averaged experiment on radiative fluxes; this value is much smaller than the impact value of low clouds. Moreover, Christensen et al. (2013) found that CloudSat-derived cloud liquid water path (LWP) is generally too high and systematically exceed those from Moderate Resolution Imaging Spectroradiometer (MODIS) by approximately 50%. Most previous studies have verified the CloudSat 2B-CWC-RO product,

15   thereby allowing us to discuss the 3D microphysical characteristics of clouds in detail (Austin et al., 2009; Rajeevan et al., 2012). Therefore, the products containing 3D cloud characteristics from CloudSat are significant in evaluating cloud characteristics and their variations.

Furthermore, retrieved profiles of cloud microphysical properties form the basis of the algorithm of another data product (2B-FLXHR), which consists of high vertical resolution profiles of radiative fluxes and atmospheric heating rates. L'Ecuyer

20   et al. (2008) detected biases between the radiative data of 2B-FLXHR from CloudSat and those from Clouds and Earth's Radiant Energy System (CERES) with monthly 5° means in the global scale. The biases of outgoing shortwave radiation (OSR), outgoing longwave (LW) radiation, surface shortwave radiation (SSR), and surface LW radiation are less than 0.1, 5.5, 13, and 16 W m$^{-2}$, respectively. Fortunately, the uncertainties in 2B-FLXHR fluxes decrease significantly for long time scale averages (L'Ecuyer et al., 2008). Therefore, the CRE derived from CloudSat can credibly describe the radiative effect

25   of clouds, especially in large space and time scales.

**2.2 CALIPSO data**

Cloud-Aerosol Lidar and Infrared Pathfinder Satellite Observations (CALIPSO) was also launched in April 2006 (Winker et al., 2007). CALIPSO mainly loads Cloud Aerosol Lidar with Orthogonal Polarization (CALIOP), which is an excellent active two-wavelength (532 and 1064 nm) polarization lidar (Winker et al., 2007). The layers detected by CALIOP

30   are correctly identified with a high degree of confidence (> 90%) by cloud–aerosol mask analysis (Liu et al., 2009). Moreover, aerosols detected by CALIOP can be classified into the following six sub-types with high accuracy: clean marine, dust, polluted continental, clean continental, polluted dust, and smoke (Omar et al., 2009). Therefore, CALIPSO provides 3D

scientific materials for and perspective on studying the changes, interactions, and transportation of aerosols and clouds at a global scale (Winker et al., 2010; Pan et al., 2015). In this study, CALIPSO Level 2 aerosol layer products are used to describe the recent changes in aerosol loading and its mostly sub-type component in the South Asian subcontinent. Moreover, the observations are compared with those of MODIS. CALIPSO is highly sensitive to thin cirrus clouds, but cannot easily

5   penetrate optically thick layers with optical depths exceeding 3–5 (Hu et al., 2009). In this case, CALIPSO determines no or low aerosol optical depth (AOD) at the near-surface due to the complete attenuation of lidar signals. To exclude the effects of unpenetrated profiles for lidar, we calculate the average AOD from CALIPSO only on the basis of the scene in which the lidar penetrates the entire atmosphere.

**2.3 CERES/MODIS data**

10      Aqua became a member of the A-Train constellation in May 2002, with the load of CERES, MODIS, and four other sensors (Remer et al., 2005; Wielicki et al., 2015). CERES/Aqua Edition 4A Single Scanner Footprint (SSF) and MODIS/Aqua Edition 6 Atmosphere Level 3 Joint Products (MYD08) data are used in this study. CERES SSF data sets combine CERES radiation measurements, cloud and aerosol microphysical property retrievals based on observations of MODIS, and ancillary meteorology fields to form a comprehensive, high-quality compilation of satellite-derived cloud,

15   aerosol, and radiation budget information for radiation and climate studies (Loeb and Manalosmith, 2005). CERES reports the observed SW and LW solar radiances with precisions approaching 1.0% (Loeb and Manalosmith, 2005). Moreover, this study uses land–ocean AOD at 0.55 μm from MYD08. On the basis of its comparison with the Aerosol Robotic Network, the MODIS AOD has been verified with absolute and relative errors of 0.05 and 15%, respectively, mainly from the retrieval algorithm (Levy et al., 2013; Wang et al., 2017). AOD data have been widely used and verified by many studies because of

20   their excellent capability of deriving AOD from dark to bright surfaces (Remer et al., 2005; Munchak et al., 2013). The most significant effects of cloud contamination on passive MODIS and CERES have to be emphasized. For example, artificially high AOD can be detected by MODIS due to the presence of thin clouds undetected by MODIS (Grandey et al., 2013). Moreover, the scattering of light by inhomogeneous clouds possibly generates uncertainty in the passive wide-field observations of MODIS and CERES, which are mainly caused by broken cloud systems (Grandey et al., 2013; Christensen

25   et al., 2016). Therefore, the uncertainties of MODIS and CERES observations are more complicated in widespread-cloud conditions than in insufficient-cloud conditions.

**2.4 Radiative transfer model**

      BUGSrad is the official radiative transfer model used by the product of CloudSat 2B-FLXHR (Fu and Liou, 1992; L'Ecuyer et al., 2008). This model is based on the two-stream doubling-adding solution to the radiative transfer equation

30   with the assumption of a plane-parallel atmosphere (Stephens et al., 2001). BUGSrad computes molecular absorption and scattering properties on the basis of correlated-k formulation (Fu and Liou, 1992). The calculation of BUGSrad is parallelly applied over six SW bands, and a constant hemisphere formulation is applied to 12 LW bands. These bands are appropriately

weighted and combined into the two broadband flux estimates that are ultimately reported, one covering the SW at 0–4 μm and the other covering the LW above 4 μm. According to the comparison experiment with observations of the Atmospheric Remotely-Sensed Clouds Locations of the Atmospheric Radiation Measurement program, the mean biases of SW and LW in clear-sky are 1.2 and 2.2 W m$^{-2}$ at TOA, respectively (Stephens et al., 2001; Stephens et al., 2003).

5 **2.5 Principle and methods**

In this study, multiple sensors from A-Train are used to investigate the possible reasons for the dimming and brightening variations during the monsoon season (June–September) in the South Asian subcontinent from 2006 to 2015. We define the study domain within the South Asian subcontinent to restrict the effect of the surrounding land and sea pixels on the results of this study, as shown in Figure 1. CloudSat, CALIPSO, and Aqua in A-Train are maintained in a tight orbital
10 coordination within 176 s (Stephens et al., 2017). This setting allows near-simultaneous observations of a wide variety of atmospheric and surface parameters, facilitating the comparison of satellite between each other, allowing for even more comprehensive studies of climate. The MYD08 and Level 2 aerosol layer products from MODIS and CALIPSO are used to describe the recent changes in aerosol loading in the South Asian subcontinent. The SSF product from CERES/Aqua is considered for evaluating the radiation energy variations. The CloudSat 2B-CLDCLASS, 2B-CWC-RO, and 2B-FLXHR
15 products are used to quantify the 3D changes in cloud macrophysical and microphysical characteristics and corresponding variable radiative effects of clouds. Moreover, Environmental conditions are obtained from the European Center for Medium-range Weather Forecast-AUXiliary analysis (ECMWF-AUX) product (Uppala et al., 2005). The CRE inferred from CERES is always disturbed by water vapor, aerosol, and the limited capability of MODIS to detect thin cirrus; furthermore, CERES considers the impact of multi-layer clouds insufficiently (Sohn et al., 2010). Therefore, this study uses the CloudSat
20 2B-FLXHR to evaluate the effects of cloud variations on radiation.

This study focuses on the effect of aerosol, cloud, and water vapor variations during the monsoon season. Due to the increased convective activity, the changes in aerosol, cloud, and water vapor and their impact on solar radiation possibly occur more obviously during the monsoon season than during non-monsoon seasons (Turner and Annamalai, 2012). Additionally, the information about aerosol, cloud, and water vapor is calculated on the basis of instantaneous observations,
25 which we then use to ensure consistent conditions for the observations of multiple satellites. The satellites or sensors of the A-Train cross the equator in the daytime at 13:30 local time, at which point the incoming solar radiation is maximum within the day (Liou, 2002). Therefore, the instantaneous observations of radiative fluxes used in this study are greater than the daily average. The results are more representative of the atmospheric condition variations at nominally 13:30 local time than the daily average. For the method of spatial and temporal averages, we firstly obtain the spatial distribution of temporal
30 average within a 1° × 1° grid during the monsoon season. We then determine the spatial average on the basis of the above temporal average in the entire South Asian subcontinent. For the spatial patterns of temporal changes, we identify the results of the linear temporal changes in atmospheric conditions during the monsoon season from 2006 to 2015.

Strict selection procedures are implemented to control the quality of the data products and thereby ensure credible conclusions. The CPR cloud mask and radar reflectivity from 2B-GEOPROF are set to be more than 20 and −28 dBZ, respectively, and the quality flag from the 2B-CLDCLASS is identified as confidence to CloudSat (Mace and Zhang, 2014). We define the cloud-sky condition for all products as the scenes in which the cloud mask from the CloudSat footprint

5    exceeds 20, and the others are defined as clear-sky condition. Under this screened method of cloud, the probability of a false detection to CloudSat is less than 5% in comparison with that by CALIPSO (Mace and Zhang, 2014). The all-sky condition is the total of cloud-sky and clear-sky condition. Data quality for CALIPSO is maintained by screening the cloud layer with a high degree of confidence (> 90%) (Hu et al., 2009). We only use the radiative data from CERES with a sensor viewing zenith angle less than 60° at the surface to restrict the uncertainty from the satellite non-nadir point (Christopher and Zhang,

10   2002). Due to the SW solar radiation (< 5 μm) comprising more than 99.5% of total solar energy, it is assumed as the total incoming solar radiation at the TOA and surface in this study. Although the BUGSrad defines the SW from 0–4 μm, the solar energy at 4–5 μm is excessively low (approximately 0.39% of total solar energy) (Liou, 2002). Therefore, the effect of this difference is ignorable between the definitions of SW between CERES and BUGSrad. Surface solar radiation is analyzed in this study in a relative sense with variable aerosols, clouds, and water vapor; thus, this study focuses mostly on

15   the relative changes in solar radiation but less on the absolute values of solar radiation.

CRE has been widely used to quantify the degree of cloud–radiation interactions (Henderson et al., 2013). CRE is the net (down minus up) flux difference between the all-sky condition and clear-sky condition on the atmosphere, surface, or TOA, as shown in Eq. (1):

$$CRE = (F^\downarrow - F^\uparrow)_{All-Sky} - (F^\downarrow - F^\uparrow)_{Clear-Sky} , \quad (1)$$

20   where $F^\downarrow$ and $F^\uparrow$ are the downwelling and upwelling radiative fluxes at the TOA and surface, respectively. One aim of this study is to quantify the impact of cloud variations on TOA and surface fluxes. Therefore, radiative data from the CloudSat 2B-FLXHR products at the surface and TOA are used. Moreover, this study uses precipitable water (PW) variation to describe the column changes of water vapor in the atmosphere. PW is the total water vapor in the atmospheric column, which is calculated on the basis of the data on environmental parameters from the ECMWF-AUX product, as shown in Eq.

25   (2):

$$PW = \frac{1}{g_0} \int_{P_{TOA}}^{P_{Surface}} q(P)dP , \quad (2)$$

where $g_0 = 9.80665$ m s$^{-2}$ is the standard acceleration due to gravity at the mean sea level (m.s.l.), $P$ is the atmospheric pressure, and $q(P)$ is the specific humidity of air as a function of atmospheric pressure (Bock et al., 2010). The PW data from ECMWF-AUX have been verified with a bias of approximately −1 mm depending on water vapor amount by comparing

30   observations from 21 ground-based Global Positioning System receiving stations and 14 radiosonde stations (Bock et al.,

2010). Therefore, although the PW inferred from ECMWF-AUX is generally slightly lower than that under actual condition, it can adequately indicate the change in water vapor amount in the South Asian subcontinent.

We accurately evaluate the effect of water vapor variations on solar radiation using the official radiative transfer model of CloudSat 2B-FLXHR (BUGSrad). By using BUGSrad, it is convenient to compare the results of the radiative transfer model with those of the CloudSat product. We use the enviromental parameters provided by CloudSat products as the input of the BUGSrad, including temperature, pressure, specific humidity and cloud mask, and so on. Then, we only change the average water vapor amount during the monsoon season, and other environmental parameters are set as the inter-annual average values during the monsoon season from 2006 to 2015 to exclude the effect of other factors. For the radiative transfer model, we directly use the averaged environmental parameters during the monsoon season to evaluate the radiative effect of water vapor by considering the quantity of the calculation. Due to the difference of averaged methods which were used by CloudSat and BUGSrad to evaluate the radiative effect of water vapor, there may are some difference between the radiative fluxes calculated based on CloudSat and BUGSrad.

**3 Results and Discussion**

**3.1 Changes in aerosols and solar radiation**

Figure 2 shows the recent changes in aerosol loading during different seasons, namely, pre-monsoon, monsoon, and dry seasons, respectively. These three seasons occur in the periods of March–May, June–September, and October–February, respectively. The aerosol loading increased, and the change in the average AOD detected by CALIPSO (approximately 0.13) was more intense than that detected by MODIS (approximately 0.05) during the monsoon season in the South Asian subcontinent from 2006 to 2015 (Figure 2a). These phenomena are mainly due to the effect of aerosols under clouds, which can be detected by CALIPSO but missed by MODIS because of MODIS only inferring the aerosol loading under clear-sky condition (Winker et al., 2010). Moreover, CALIPSO can identify aerosol and cloud layer with ~90% accuracy. The sum of dust, smoke, and polluted dust aerosol loadings is close to the total aerosol loading (orange line in Figure 2a). This phenomenon indicates that aerosols mostly consist of dust, smoke, and polluted dust during the monsoon season in the South Asian subcontinent, which is consistent with the previous study (Das et al., 2015). These aerosols all present distinct absorbing effects on solar radiation, thereby attenuating the incoming solar radiation more intensively than other types of aerosols (Logan et al., 2013).

Figure 2b shows the consistent increase in AOD during the pre-monsoon, monsoon, and dry seasons, respectively. The increment of AOD during the pre-monsoon and dry seasons (by 0.08 and 0.11, respectively) were greater than those during the monsoon season according to the MODIS observations from 2006 to 2015. The significant level of AOD variation detected by MODIS was higher during non-monsoon season ($p<0.01$) than during the monsoon season, which mainly due to the disturbance of widespread clouds during the monsoon season (Remer et al., 2005). The change in the average AOD

detected by CALIPSO showed a higher significance level ($p$=0.07) than that detected by MODIS due to the minimal disturbance of thin clouds and surface albedos to the aerosols detected by CALIPSO during the monsoon season (Redemann et al., 2012). Additionally, the robust and quick increase in aerosols in the South Asia has been verified by many previous studies (Suresh et al., 2013; Reddy et al., 2016; Srivastava, 2016).

As shown in Figure 3a, the average SSR (red line) in all-sky condition from CERES increased by 16.2 W m$^{-2}$ increments during the monsoon season in the South Asian subcontinent from 2006 to 2015. Moreover, the change in OSR (by 13.0 W m$^{-2}$) showed a highly negative correlation with that in the SSR during the study period (Figure 3b). These phenomena indicate that the attenuating effect of the atmosphere on solar radiation is gradually weakened, thereby increasing the incident solar radiation at the surface (brightening). There is the fact that the SSR is highly negatively correlated with aerosol loading (Folini and Wild, 2011). Therefore, the increase in AOD did not contribute to the recent brightening during the monsoon season in the South Asian subcontinent from 2006 to 2015.

Further, Figure 3a shows no distinct change in the SSR during the pre-monsoon season, even the reverse trend of SSR during the dry season (dimming) comparing to that during the monsoon season. The seasonal differences in OSR variations are similar to those of SSR (Figure 3b). Moreover, the trend of SSR during the dry season presented a higher significance level ($p$<0.01) than that during the monsoon season ($p$=0.40). Many reasons can contribute to the significance levels during different seasons. The average cloud fractions during the pre-monsoon and dry seasons are both approximately 20%, and much lower than that during the monsoon season (53.4%) according to CloudSat observations. Moreover, under clear-sky condition, aerosols control the change in the solar radiation that reaches the surface (Nair et al., 2016; Soni et al., 2016). Therefore, aerosols have greater weighted impact on solar radiation during the dry season than during the monsoon season. In addition, the resultant effects of aerosol, cloud, and water vapor are considerably complicated and varied due to the intensive atmospheric activities during the monsoon season. The pre-monsoon season is the transition season from the dry season to the monsoon season, and associated with the establishment of the southwestern wind regime over the South Asian subcontinent (Das et al., 2015). The atmospheric condition and radiation variations are not distinct comparing to the trend of SSR between the dry and monsoon seasons.

**3.2 Effect of clouds**

Figure 4 shows the inter-annual variations in the average cloud vertical frequency distribution in all-sky condition and the cloud microphysical parameters in cloud-sky condition during the monsoon season in the South Asian subcontinent from 2006 to 2015. The cloud vertical frequency sequentially decreased during the monsoon season in the South Asian subcontinent from 2006 to 2015, with a maximum decrement of 7.1%, which contributed to approximately 30% of the maximum cloud vertical frequency at approximately 12 km m.s.l. (Figure 4a). The cloud water content and number concentration of liquid and ice in cloud-sky condition showed a consistently decreasing trend, with maximum decrements of 18.0 mg m$^{-3}$ of LWC and 1.4 mg m$^{-3}$ of IWC, as well as approximately 6.0 cm$^{-3}$ of LNC and 2.0 L$^{-3}$ of INC (Figures 4d and

4g). These changes are consistent with those in the cloud vertical frequency during the study period. However, the LER and IER variations were not distinct and not highly consistent with the changes in the cloud vertical frequency (Figures 4c and 4f). Therefore, the clouds decreased with the reduced cloud water content and number concentration, but no clear change was observed in the particle effective radius during the monsoon season in the South Asian subcontinent from 2006 to 2015.

5    As shown in Figure 5a, the total cloud fraction detected by CloudSat declined by 8.8% ($p$=0.07) during the monsoon season in the South Asian subcontinent from 2006 to 2015. Furthermore, the change in CWP in cloud-sky condition was consistent with that of the total cloud fraction with a decrement of 54.0 g m$^{-2}$ ($p$<0.01). The uppermost cloud top height (CTH) declined by 661 m ($p$=0.08), and the cloud geometrical depth (CGD) declined by 280 m ($p$=0.21) with partial fluctuations in 2009 and 2010. By contrast, an insignificant trend occurred in the lowermost cloud base height (CBH) 10  variation. We then further analyze the spatial variations in cloud fraction, cloud height (CH), CWP, and CGD during the monsoon season in the South Asian subcontinent from 2006 to 2015. Figures 5c–5f illustrate the consistent decreases in cloud fraction, CH, CWP, and CGD, especially for the cloud fraction and CWP with significant and consistent changes in space. CH and CGD increased in certain parts of the western coastal area in the South Asian subcontinent. In general, we can conclude that the clouds decreased and thinned with decreases in water content and particle number concentration during the 15  monsoon season in the South Asian subcontinent from 2006 to 2015, thereby weakening the regulation of cloud on radiation.

The factors contributing to the changes in clouds are complicated and varied in the South Asian subcontinent. Ackerman et al. (2000) verified that aerosols with intensive absorption may lead to evaporation in the cloud layers (the aerosol semi-direct effect), burning off the clouds in the South Asia, including widespread dust, polluted dust, and smoke. Bollasina et al. (2011) showed the anthropogenic aerosol emissions weakened the South Asian summer monsoon, causing a 20  decrease in the observed occurrence and amount of precipitation. In the global scale, observed and simulated cloud change patterns are consistent with the poleward retreat of mid-latitude storm tracks, which is attributed to the increasing greenhouse gas concentrations and recovery from volcanic radiative cooling (Norris et al., 2016). Actual, with the reduction of clouds, the recent weakened rainfall and dried monsoon during the summer monsoon season have been verified by many previous studies (Bollasina et al., 2011; Turner and Annamalai, 2012; Annamalai et al., 2013). However, the decreasing clouds also 25  cause the increase in SSR, possibly enhancing the generation of rainfall (Wild, 2009). Therefore, the weakened rainfall was not steady, and can be attributed to the resultant force of many factors, including the changes in monsoon intensity, intrinsic cloud properties, and anthropogenic aerosols, to name a few.

Consequently, we investigate the radiative force of cloud vertical variations to determine their possible contribution to the recent rapid brightening (Figure 6). We use the vertical heating rates in all-sky condition to indicate the change in the 30  vertically radiative effects of clouds (Fu and Liou, 1992). Given the changes in cloud macrophysical and microphysical characteristics, the SW heat rating decreased by 0.3–0.4 K d$^{-1}$ at an altitude of 5–15 km from 2006 to 2015, whereas a few increase in SW heat rating occurred at the low-cloud and near-surface areas (below 4 km). With the reduction of clouds, the total cloud reflection and absorption (mainly located at 5–15 km) weakened consistently, thus enhancing atmospheric

transmission, causing more SW radiation than before to reach the near-surface area (Henderson et al., 2013). Due to the extremely little cloud detected by CloudSat above 15 km, no significant change in heat rating occurred above 15 km, as shown in the cloud vertical frequency distribution in Figure 4a. However, the change in LW vertical heating rate was not distinct compared with that of SW. Correspondingly, the LW radiative heating increased by less than 0.2 K d$^{-1}$ above 10 km,

5    whereas that of heating decreased by 0.1–0.2 K d$^{-1}$ within the cloud, especially below 4 km, from 2006 to 2015. The total cloud LW absorption emitted from the surface decreased with the reduction of clouds, thus leading to a weakened LW heating rate within the cloud layer (mainly below 10 km), and more LW emission on the top cloud layer above 10 km (L'Ecuyer et al., 2008). In general, the net vertical heating rate weakened consistently, indicating that the changes in the clouds physical characteristics weakened their vertical radiative effect of clouds.

10       As shown in Figure 7a, the SW CRE at the surface and TOA weakened by approximately 42.1 and 38.5 W m$^{-2}$ with significance levels $p$ of 0.05 and 0.04, respectively. Solar radiation consequently increased and reached at the surface during the monsoon season in the South Asian subcontinent from 2006 to 2015. The decrease in SW CRE at the surface and TOA was highly consistent with that of the total cloud faction (red line in Figure 7a). Although the CRE derived from CloudSat largely ignores the contribution of high thin clouds, this contribution is much smaller than that of low clouds (L'Ecuyer et al.,

15    2008; Henderson et al., 2013). Moreover, SW CRE (Figure 7b) at the surface is negatively correlated with SSR (Figure 7c). This observation is attributed to the spatial variations in the radiative effect at the surface caused by cloud variables. In general, clouds were reduced and thinned by approximately 8.8% and 280 m with the decrease in cloud water content by 54.0 g m$^{-2}$ and particle number concentration in cloud-sky condition. Consequently, the SW radiative effect of the clouds decreased by approximately 42.1 W m$^{-2}$, and the absorption of direct solar radiation at the surface increased, thus leading to

20    subsequent surface brightening.

**3.3 Effect of water vapor**

      In this section, we evaluate the effect of water vapor on the recent brightening in the South Asian subcontinent. We focus mainly on the effect of water vapor on solar radiation in clear-sky condition to eliminate the disturbance of clouds in the evaluation of water vapor in all-sky condition. The vertical relative humidity (RH) in all-sky condition gradually

25    decreased with a maximum decrement of approximately 10% during the monsoon season in the South Asian subcontinent from 2006 to 2015 (Figure 8a). Moreover, the change in RH in all-sky condition was highly consistent with that in clear-sky condition (Figures 8a and 8c). Figure 8b shows the spatial variation in PW in all-sky condition during the monsoon season in the South Asian subcontinent from 2006 to 2015; this spatial variation is highly consistent with that in clear-sky condition (Figure 8d). PW consistently decreased in space with a maximum decrement of 6.6 mm in the South Asian subcontinent,

30    especially in the western and central regions. Although a small increase in water vapor (less than 2 mm) occurred in part of the South Asian region, its increment was much lower than the decrease in water vapor in most regions of the South Asian subcontinent. The decrease in atmospheric water vapor may be attributed to the weakened monsoon intensity in the recent

decade during the monsoon season in the South Asia (Turner and Annamalai, 2012).

Figure 9a shows that radiation is highly and negatively correlated with PW from ECMWF-AUX, thereby indicating that water vapor is a substantial controlling factor in solar radiation variability. This control lies in the direct and indirect effects. The former refers to the absorption of solar radiation, in which the PW decreased by nearly 2.8 mm ($p$=0.13) over the brightening period. According to the evaluation of BUGSrad, solar brightening was greater than 2.5 W m$^{-2}$ ($p$=0.14) because of the weakened absorption caused by the decrease in water vapor; the value was much lower than that of cloud variations (42.1 W m$^{-2}$). The SSR under clear-sky condition from CloudSat increased by approximately 4.8 W m$^{-2}$ ($p$=0.05). Except for the contribution of water vapor variations, this radiative change was possibly caused by the change in atmospheric conditions and incoming solar radiation provided by ECMWF and CloudSat (Yang et al., 2012). The spatial variation in SSR under clear-sky condition was highly negative and consistent with that of PW, especially in the western and central regions of the South Asian subcontinent (Figure 9b), which partly consist of deserts. Moreover, in all-sky condition, there are the indirect effects of water vapor, which is named as water vapor-cloud interaction. Generally, aside from the effect of temperature, a high concentration of water vapor favors the generation of more or stronger convective clouds and results in decreased radiation at the surface, and vice versa (Yang et al., 2012). Additionally, water vapor regulates cloud amount by affecting the aerosol-cloud interaction, which generally shows a positive relationship under moist atmospheric conditions (Chen et al., 2014). Therefore, the impact of water vapor in all-sky condition is more complicated than that in clear-sky condition owing to the direct and indirect effects of water vapor on solar radiation.

**3.4 Climate and environmental implications**

Variations in surface solar radiation can potentially significantly impact the climate system based on the long temporal correlation between solar radiation and climate factors, especially in temperature and rainfall. Observed dimming was suggested to be responsible for the absence of a significant temperature rise between the 1950s and the 1980s in various parts of the world, such as in the Arctic, China, America, and India (Ramanathan et al., 2005; Wild, 2009). However, the suppression of global warming over global land surfaces only lasted into the 1980s with the transition of dimming to brightening (Wild et al., 2005). This condition indicates that brightening might have significantly contributed to the recent rapid warming after the 1980s according to ground-based observations (Philipona et al., 2009). This temperature evolution satisfactorily fits the observational surface solar radiation variations, and points to the crucial role that dimming and brightening may play in determining global warming. Moreover, surface solar radiation variations induce changes in the surface net radiation, thereby altering the energy available for evaporation, which equals precipitation in the global annual mean. A decrease in evaporation and the same globally averaged reduction in precipitation with the globally averaged dimming occurred in the 1980s (Liepert, 2002). Wild et al. (2009) showed that the increase in available surface energy is quantitatively consistent with the observed substantial increase in land precipitation (3.5 mm y$^{-1}$ between 1986 and 2000) and the associated intensification of the land-based hydrological cycle.

The recent rapid warming in South Asia has been verified by many studies (Turner and Annamalai, 2012; Annamalai et al., 2013). Moreover, ECMWF-AUX shows that the 2 m temperature increased by 0.14 K y$^{-1}$, even for surface temperature with the increment of 0.4 K y$^{-1}$ during the monsoon season in the South Asian subcontinent from 2006 to 2015 (not shown). Syed et al. (2014) predicted that the mean surface air temperature in the monsoon season would increase from 2.5 °C to 5 °C

5    with increasing greenhouse gas concentrations by the end of the century. Models may underestimate the rate of global warming due to the insufficient consideration of dimming and brightening (Wild, 2012). With brightening considered, warming may accelerate in South Asia at unprecedented rates. However, the recent monsoon rainfall roughly weakened with no homogeneity by combining the model and observations of satellites and sites (Bollasina et al., 2011; Turner and Annamalai, 2012; Annamalai et al., 2013). Reduced clouds may indicate a decrease in average rainfall during the monsoon

10   season, whereas brightening generally enhances the generation of rainfall. Turner and Annamalai (2012) showed consistent negative trends of rainfall over northwest India and coastal Burma, while the positive trends in southeast India; they also found a distinct disagreement among different rainfall data in northeast India. In general, the recent brightening may play a prominent role in modulating the warming and rainfall variations in the South Asian subcontinent.

**4 Conclusions**

15   Surface solar radiation is the ultimate energy source for life on the planet (IPCC, 2013). Any change in surface solar radiation profoundly affects the global ecosystem, further determines the living conditions of humans (Haywood et al., 2011). South Asia is experiencing a leveling-off trend with regard to solar radiation, even a transition from dimming to brightening. This process is significantly complicated because of the active atmospheric action during the monsoon season. In this study, we use observations from multiple satellites/sensors on the A-Train satellite to evaluate the effect of aerosol, cloud, and

20   water vapor variations on the recent changes in solar radiation at the surface during the monsoon season (June–September) in the South Asian subcontinent from 2006 to 2015, mainly including CloudSat, CALIPSO, MODIS and CERES.

We found the SSR and OSR increased by 16.2 W m$^{-2}$ and decreased by 13.0 W m$^{-2}$ with a partly fluctuation during the monsoon season, respectively. On the contrary, the increase in AOD was inconsistent with the SSR variation, and did not contribute to the recent brightening in the South Asian subcontinent. Decreases in water vapor amount and clouds

25   significantly contributed to solar brightening and subsequent surface warming in the South Asian subcontinent. In general, the clouds were generally reduced and thinned by approximately 8.8% and 280 m, respectively, when the cloud water path (by 54.0 g m$^{-2}$) and particle number concentration in cloud-sky condition decreased. Given the change in clouds, the atmospheric vertical SW heat rating decreased by 0.3–0.4 K d$^{-1}$ at the altitude of 5–15 km, whereas SW heat rating slightly increased at the low-cloud and near-surface areas (below 4 km) from 2006 to 2015. Correspondingly, SW CRE weakened by

30   approximately 42.1 W m$^{-2}$ at the surface. Moreover, the PW in clear-sky condition decreased by nearly 2.8 mm over the brightening period. Consequently, solar brightening increased by approximately 2.5 W m$^{-2}$ because of the weakened absorption. The decreases in water vapor amount and clouds weakened the effect of water vapor and clouds on solar

radiation, thereby resulting in the increased absorption of direct solar radiation at the surface and subsequent surface brightening.

Notably, CloudSat is insufficient in observing thin cirrus with small IWC, thereby resulting in uncertainties for quantifying changes in clouds. In a globally averaged experiment on radiative fluxes, L'Ecuyer et al. (2008) proposed that the impact of the thin cirrus that is not detected by CloudSat on SW radiation at the surface is $-1.2$ W m$^{-2}$, which is much smaller than the impact of low clouds. The data in this study were from the instantaneous observations of multiple sensors. Thus, the magnitudes of radiative impact may were overestimated for the water vapor and cloud variations. The results are more representative of the atmospheric condition variations at nominally constant solar zenith angle than the daily average. In sum, the conclusions of this study are beneficial in the evaluation of the recent variations in aerosol, cloud, and water vapor and their resultant influences on solar radiation. Additionally, the aerosol, cloud, and water vapor variations are correlated with regional sources and meteorology. Consequently, solar dimming and brightening may be of local or regional nature, and they are unavoidably inhomogeneous and unstable in space (Soni et al., 2016). The variations in aerosols, water vapor, and clouds highly interact with changes in monsoon circulation. Thus, it would be interesting to investigate the change mechanisms of aerosols, clouds, and water vapor in South Asia. The chicken-and-egg relationship between monsoon and aerosols, water vapor, and clouds can also be explored.

**Acknowledgments**

This study was supported by the National Science Foundation of China (41627804), National Key Research and Development Program of China (2016YFC0200900), Program for Innovative Research Team in University of Ministry of Education of China (IRT1278), and China Postdoctoral Science Foundation (2016T90731). We would like to thank the CloudSat, CALIPSO, CERES, and MODIS science teams for providing excellent and accessible data products that made the study possible. The CloudSat data in this study were acquired from CloudSat Data Processing Center (DPC, http://www.cloudsat.cira.colostate.edu/). The CERES SSF and CALIPSO data were obtained from NASA Langley Research Center Atmospheric Sciences Data Center (ASDC, https://eosweb.larc.nasa.gov/). The MODIS AOD data were obtained from the NASA Earth Observing System Data and Information System, Distributed Active Archive Center (DAAC, https://ladsweb.modaps.eosdis.nasa.gov/).

**References**

Ackerman, A. S., Toon, O. B., Stevens, D. E., Heymsfield, A. J., Ramanathan, V., and Welton, E. J.: Reduction of tropical cloudiness by soot, Science, 288, 1042-1047, 2000.

Annamalai, H., Hafner, J., Sooraj, K. P., and Pillai, P.: Global Warming Shifts the Monsoon Circulation, Drying South Asia, Journal of Climate, 26, 2701-2718, 2013.

Austin, R. T., Heymsfield, A. J., and Stephens, G. L.: Retrieval of ice cloud microphysical parameters using the CloudSat millimeter-wave radar and temperature, Journal of Geophysical Research Atmospheres, 114, 1065-1066, 2009.

Bock, O., Keil, C., Richard, E., Flamant, C., and Bouin, M. N.: Validation of precipitable water from ECMWF model analyses with GPS and radiosonde data during the MAP SOP, Quarterly Journal of the Royal Meteorological Society, 131, 3013-3036, 2010.

Bollasina, M. A., Ming, Y., and Ramaswamy, V.: Anthropogenic Aerosols and the Weakening of the South Asian Summer Monsoon, Science, 334, 502-505, 2011.

5 Chen, Y. C., Christensen, M. W., Stephens, G. L., and Seinfeld, J. H.: Satellite-based estimate of global aerosol-cloud radiative forcing by marine warm clouds, Nature Geoscience, 7, 643-646, 2014.

Christensen, M. W., Stephens, G. L., and Lebsock, M. D.: Exposing biases in retrieved low-cloud properties from cloudsat: A guide for evaluating observations and climate data †, Journal of Geophysical Research Atmospheres, 118, 12120-12131, 2013.

Christensen, M. W., Chen, Y. C., and Stephens, G. L.: Aerosol indirect effect dictated by liquid clouds, Journal of Geophysical Research: 10 Atmospheres, 121, 2016.

Christopher, S. A., and Zhang, J.: Shortwave Aerosol Radiative Forcing from MODIS and CERES observations over the oceans, Geophysical Research Letters, 29, 6-1–6-4, 2002.

Das, S., Dey, S., Dash, S. K., Giuliani, G., and Solmon, F.: Dust aerosol feedback on the Indian summer monsoon: Sensitivity to absorption property, Journal of Geophysical Research Atmospheres, 120, 9642-9652, 2015.

15 Folini, D., and Wild, M.: Aerosol emissions and dimming/brightening in Europe: Sensitivity studies with ECHAM5-HAM, Journal of Geophysical Research Atmospheres, 116, 21104, 2011.

Fu, Q., and Liou, K. N.: On the correlated k-distribution method for radiative transfer in nonhomogeneous atmospheres, Journal of Atmospheric Sciences, 49, 2139 - 2156, 1992.

Grandey, B. S., Stier, P., and Wagner, T. M.: Investigating relationships between aerosol optical depth and cloud fraction using satellite, 20 aerosol reanalysis and general circulation model data, Atmospheric Chemistry & Physics, 17, 30805-30823, 2013.

Haywood, J. M., Nicolas, B., Andy, J., Olivier, B., Martin, W., and Shine, K. P.: The roles of aerosol, water vapor and cloud in future global dimming/brightening, Journal of Geophysical Research Atmospheres, 116, -, 2011.

Henderson, D. S., L'Ecuyer, T., Stephens, G., Partain, P., and Sekiguchi, M.: A Multisensor Perspective on the Radiative Impacts of Clouds and Aerosols, Journal of Applied Meteorology & Climatology, 52, 853-871, 2013.

25 Heymsfield, A. J., Wang, Z., and Matrosov, S.: Improved Radar Ice Water Content Retrieval Algorithms Using Coincident Microphysical and Radar Measurements, Journal of Applied Meteorology, 44, 1391-1412, 2010.

Hu, Y., Winker, D., Vaughan, M., Lin, B., Omar, A., Trepte, C., Flittner, D., Yang, P., Nasiri, S. L., and Baum, B.: CALIPSO/CALIOP Cloud Phase Discrimination Algorithm, Journal of Atmospheric & Oceanic Technology, 26, 2293-2309, 2009.

IPCC: Climate Change 2013: The Physical Science Basis. Contribution of Working Group I to the Fifth Assessment Report of the 30 Intergovernmental Panel on Climate Change, Cambridge University Press, New York, USA, 2013.

Kambezidis, H. D., Kaskaoutis, D. G., Kharol, S. K., Moorthy, K. K., Satheesh, S. K., Kalapureddy, M. C. R., Badarinath, K. V. S., Sharma, A. R., and Wild, M.: Multi-decadal variation of the net downward shortwave radiation over south Asia: The solar dimming effect, Atmospheric Environment, 50, 360-372, 2012.

Kim, D., and Ramanathan, V.: Solar radiation budget and radiative forcing due to aerosols and clouds, Journal of Geophysical Research 35 Atmospheres, 113, 194-204, 2008.

Kvalevåg, M. M., and Myhre, G.: Human Impact on Direct and Diffuse Solar Radiation during the Industrial Era, Journal of Climate, 20, 4874-4883, 2007.

L'Ecuyer, T. S., Wood, N. B., Haladay, T., Stephens, G. L., and Stackhouse, P. W.: Impact of clouds on atmospheric heating based on the R04 CloudSat fluxes and heating rates data set, Journal of Geophysical Research Atmospheres, 113, 2013-2018, 2008.

40 Levy, R. C., Mattoo, S., Munchak, L. A., and Remer, L. A.: The Collection 6 MODIS aerosol products over land and ocean, Atmospheric Measurement Techniques, 6, 2989-3034, 2013.

Liepert, B. G.: Observed reductions of surface solar radiation at sites in the United States and worldwide from 1961 to 1990, Geophysical Research Letters, 29, 61-61–61-64, 2002.

Liou, K.-N.: An Introduction to Atmospheric Radiation, Academic press, San Diego, USA, 2002.

45 Liu, Z., Vaughan, M., Winker, D., Kittaka, C., Getzewich, B., Kuehn, R., Omar, A., Powell, K., Trepte, C., and Hostetler, C.: The CALIPSO Lidar Cloud and Aerosol Discrimination: Version 2 Algorithm and Initial Assessment of Performance, Journal of Atmospheric & Oceanic Technology, 26, 1198-1213, 2009.

Loeb, N. G., and Manalosmith, N.: Top-of-Atmosphere Direct Radiative Effect of Aerosols over Global Oceans from Merged CERES and MODIS Observations, Journal of Climate, 18, 3506-3526, 2005.

50 Logan, T., Xi, B., Dong, X., Li, Z., and Cribb, M.: Classification and investigation of Asian aerosol absorptive properties, Atmospheric Chemistry and Physics, 13, 2253-2265, 2013.

Long, C. N., Dutton, E. G., Augustine, J. A., Wiscombe, W., Wild, M., Mcfarlane, S. A., and Flynn, C. J.: Significant decadal brightening of downwelling shortwave in the continental United States, Journal of Geophysical Research Atmospheres, 114, 1291-1298, 2009.

Mace, G. G., Zhang, Q., Vaughan, M., Marchand, R., Stephens, G., Trepte, C., and Winker, D.: A description of hydrometeor layer 55 occurrence statistics derived from the first year of merged Cloudsat and CALIPSO data, Journal of Geophysical Research: Atmospheres, 114, 414-416, 10.1029/2007JD009755, 2009.

Mace, G. G., and Zhang, Q.: The CloudSat radar-lidar geometrical profile product (RL-GeoProf): Updates, improvements, and selected results, Journal of Geophysical Research: Atmospheres, 119, 9441-9462, 2014.

Munchak, L. A., Levy, R. C., Mattoo, S., and Remer, L. A.: MODIS 3 km aerosol product: applications over land in an urban/suburban region, Atmospheric Measurement Techniques, 6, 1683-1716, 2013.

5 Nair, V. S., Babu, S. S., Manoj, M. R., Moorthy, K. K., and Chin, M.: Direct radiative effects of aerosols over South Asia from observations and modeling, Climate Dynamics, 1-18, 2016.

Norris, J. R., Allen, R. J., Evan, A. T., Zelinka, M. D., O'Dell, C. W., and Klein, S. A.: Evidence for climate change in the satellite cloud record, Nature, 536, 72, 2016.

Omar, A. H., Winker, D. M., Kittaka, C., Vaughan, M. A., Liu, Z., Hu, Y., Trepte, C. R., Rogers, R. R., Ferrare, R. A., and Lee, K.-P.: The
10 CALIPSO Automated Aerosol Classification and Lidar Ratio Selection Algorithm, Journal of Atmospheric and Oceanic Technology, 26, 1994, 2009.

Padma, K. B., Londhe, A. L., Daniel, S., and Jadhav, D. B.: Observational evidence of solar dimming: Offsetting surface warming over India, Geophysical Research Letters, 34, 377-390, 2007.

Pan, Z., Gong, W., Mao, F., Li, J., Wang, W., Li, C., and Min, Q.: Macrophysical and optical properties of clouds over East Asia measured
15 by CALIPSO, Journal of Geophysical Research: Atmospheres, 120, 11653-11668, 10.1002/2015JD023735, 2015.

Pan, Z., Mao, F., Gong, W., Min, Q., and Wang, W.: The warming of Tibetan Plateau enhanced by 3D variation of low-level clouds during daytime, Remote Sensing of Environment, 198, 363-368, 2017.

Philipona, R., Behrens, K., and Ruckstuhl, C.: How declining aerosols and rising greenhouse gases forced rapid warming in Europe since the 1980s, Geophysical Research Letters, 36, 206-218, 2009.

20 Rajeevan, M., Rohini, P., Kumar, K. N., Srinivasan, J., and Unnikrishnan, C. K.: A Study of Vertical Cloud Structure of the Indian Summer Monsoon using CloudSat data, Climate Dynamics, 40, 637-650, 2012.

Ramanathan, V., Chung, C., Kim, D., Bettge, T., Buja, L., Kiehl, J., Washington, W., Fu, Q., Sikka, D., and Wild, M.: Atmospheric brown clouds: Impacts on South Asian climate and hydrological cycle, Proceedings of the National Academy of Sciences, 102, 5326-5333, 2005.

Reddy, K. R. O., Balakrishnaiah, G., Gopal, K. R., Reddy, N. S. K., Rao, T. C., Reddy, T. L., Hussain, S. N., Reddy, M. V., Reddy, R. R.,
25 and Boreddy, S. K. R.: Long term (2007–2013) observations of columnar aerosol optical properties and retrieved size distributions over Anantapur, India using multi wavelength solar radiometer, Atmospheric Environment, 142, 238-250, 2016.

Redemann, J., Vaughan, M. A., Zhang, Q., Shinozuka, Y., Russell, P. B., Livingston, J. M., Kacenelenbogen, M., and Remer, L. A.: The comparison of MODIS-Aqua (C5) and CALIOP (V2 & V3) aerosol optical depth, Atmospheric Chemistry and Physics, 12, 3025-3043, 2012.

30 Remer, L. A., Kaufman, Y. J., Tanré, D., Mattoo, S., Chu, D. A., Martins, J. V., Li, R. R., Ichoku, C., Levy, R. C., and Kleidman, R. G.: The MODIS Aerosol Algorithm, Products, and Validation, Journal of Atmospheric Sciences, 62, 947-973, 2005.

Sohn, B. J., Nakajima, T., Satoh, M., and Jang, H. S.: Impact of different definitions of clear-sky flux on the determination of longwave cloud radiative forcing: NICAM simulation results, Atmospheric Chemistry and Physics, 10, 11641-11646, 2010.

Soni, V. K., Pandithurai, G., and Pai, D. S.: Is there a transition of solar radiation from dimming to brightening over India?, Atmospheric
35 Research, 169, 209-224, 2016.

Srivastava, R.: Trends in aerosol optical properties over South Asia, International Journal of Climatology, 37, n/a-n/a, 2016.

Stephens, G., Winker, D., Pelon, J., Trepte, C., Vane, D., Yuhas, C., L'Ecuyer, T., and Lebsock, M.: CloudSat and CALIPSO within the A-Train: Ten years of actively observing the Earth system, Bulletin of the American Meteorological Society, 0, null, 10.1175/bams-d-16-0324.1, 2017.

40 Stephens, G. L., Gabriel, P. M., and Partain, P. T.: Parameterization of Atmospheric Radiative Transfer. Part I: Validity of Simple Models, Journal of the Atmospheric Sciences, 58, 3391-3409, 2001.

Stephens, G. L., Wood, N. B., and Gabriel, P. M.: An Assessment of the Parameterization of Subgrid-Scale Cloud Effects on Radiative Transfer. Part I: Vertical Overlap, Journal of the Atmospheric Sciences, 61, 47-52, 2003.

Suresh, B. S., Manoj, M. R., Krishna, M. K., Gogoi, M. M., Nair, V. S., Kumar, K. S., Satheesh, S. K., Niranjan, K., Ramagopal, K., and
45 Bhuyan, P. K.: Trends in aerosol optical depth over Indian region: Potential causes and impact indicators, Journal of Geophysical Research Atmospheres, 118, 11,794–711,806, 2013.

Syed, F. S., Iqbal, W., Syed, A. A. B., and Rasul, G.: Uncertainties in the regional climate models simulations of South-Asian summer monsoon and climate change, Climate Dynamics, 42, 2079-2097, 2014.

Turner, A. G., and Annamalai, H.: Climate change and the South Asian summer monsoon, Nature Climate Change, 2, 587-595, 2012.

50 Uppala, S. M., Kållberg, P. W., Simmons, A. J., Andrae, U., Bechtold, V. D. C., Fiorino, M., Gibson, J. K., Haseler, J., Hernandez, A., and Kelly, G. A.: The ERA-40 re-analysis, Quarterly Journal of the Royal Meteorological Society, 131, 2961-3012, 2005.

Wang, W., Mao, F., Pan, Z., Du, L., and Gong, W.: Validation of VIIRS AOD through a Comparison with a Sun Photometer and MODIS AODs over Wuhan, Remote Sensing, 9, 403, 2017.

Wang, Y., and Wild, M.: A new look at solar dimming and brightening in China, Geophysical Research Letters, 43, 2016.

55 Wielicki, B. A., Barkstrom, B. R., Harrison, E. F., Iii, R. B. L., Smith, G. L., and Cooper, J. E.: Clouds and the Earth's Radiant Energy System (CERES): An Earth Observing System Experiment, Bulletin of the American Meteorological Society, 77, 853-868, 2015.

Wild, M., Gilgen, H., Roesch, A., Ohmura, A., Long, C. N., Dutton, E. G., Forgan, B., Kallis, A., Russak, V., and Tsvetkov, A.: From dimming to brightening: decadal changes in solar radiation at Earth's surface, Science, 308, 847-850, 2005.

Wild, M.: Global dimming and brightening: A review, Journal of Geophysical Research Atmospheres, 114, D00D16, 2009.

Wild, M., Trüssel, B., Ohmura, A., Long, C. N., König‑Langlo, G., Dutton, E. G., and Anatoly, T.: Global dimming and brightening: An
5  update beyond 2000, Journal of Geophysical Research Atmospheres, 114, 895-896, 2009.

Wild, M.: Enlightening Global Dimming and Brightening, Bulletin of the American Meteorological Society, 93, 27-37, 2012.

Willson, R. C., and Mordvinov, A. V.: Secular total solar irradiance trend during solar cycles 21–23, Geophysical Research Letters, 30, 3-1, 2003.

Winker, D. M., Hunt, W. H., and McGill, M. J.: Initial performance assessment of CALIOP, Geophysical Research Letters, 34, 228-262,
10  2007.

Winker, D. M., Pelon, J., Coakley, J. A., Ackerman, S. A., Charlson, R. J., Colarco, P. R., Flamant, P., Fu, Q., Hoff, R. M., and Kittaka, C.: The CALIPSO Mission: A Global 3D View of Aerosols and Clouds, Bulletin of the American Meteorological Society, 91, 1211-1229, 2010.

Woods, C. P., Waliser, D. E., Li, J. L., Austin, R. T., Stephens, G. L., and Vane, D. G.: Evaluating CloudSat ice water content retrievals
15  using a cloud-resolving model: Sensitivities to frozen particle properties, Journal of Geophysical Research, 113, 2739-2740, 2008.

Yang, K., Ding, B., Qin, J., Tang, W., Ning, L., and Lin, C.: Can aerosol loading explain the solar dimming over the Tibetan Plateau, Geophysical Research Letters, 39, L20710, 2012.

**Table 1. Observations used in analysis along with their sources and spatial resolutions.**

| Sensor | Spatial Resolution | Products | Parameter |
|---|---|---|---|
| CloudSat | 1.3 × 1.7 km | 2B-CLDCLASS | Cloud fraction; cloud top/base height |
| | | 2B-CWC-RO | Cloud water content, particle effective radius, and particle number concentration |
| | | 2B-FLXHR | Vertical heating rate; SW/LW radiative fluxes |
| CALIPSO | 5 × 5 km | Level 2 Aerosol Layer | Aerosol optical depth; vertical feature mask |
| MODIS | 1° × 1° | Level 3 MYD08 | Aerosol optical depth |
| ECMWF | 2.5° × 2.5° | ECMWF-AUX | Pressure; temperature; relative humidity |
| CERES | 20 × 20 km | Level 2 Single Scanner Footprint | SW radiation at TOA and surface |

[Figure]

**Figure 1. Terrain map during the monsoon season in the South Asia and surrounding regions. The blue line indicates rivers; the study region is within the red line.**

[Figure]

**Figure 2.** Temporal variations in spatial average AOD from (a) CALIPSO and MODIS during the monsoon season, as well as (b) MODIS during pre-monsoon, monsoon, and dry seasons in the South Asian subcontinent from 2006 to 2015, respectively. The dashed lines indicate the linear fit line of the solid line with the same color; *p* is the significance level; *k* is the slope of the line trend of each time series.

[Figure]

**Figure 3. Temporal variations in spatial average (a) SSR and (b) OSR from CERES during pre-monsoon, monsoon, and dry seasons in the South Asian subcontinent from 2006 to 2015, respectively. The dashed lines indicate the linear fit line of the solid line with the same color; *p* is the significance level; *k* is the slope of the line trend of each time series.**

[Figure]

**Figure 4. Temporal variations in spatial average vertical cloud physical parameters from CloudSat during the monsoon season in the South Asian subcontinent from 2006 to 2015: (a) cloud vertical frequency distribution, (b) LWC and (e) IWC, (c) LER and (f) IER, and (d) LNC and (g) INC.**

[Figure]

**Figure 5. Temporal variations in spatial average (a) cloud fraction and CWP, as well as (b) uppermost CTH, lowermost CBH, and CGD; spatial distributions of the temporal changes in average (c) cloud fraction, (d) CH, (e) CWP, and (f) CGD from CloudSat during the monsoon season in the South Asian subcontinent from 2006 to 2015. The dashed lines indicate the linear fit line of the solid line with the same color; *p* is the significance level; *k* is the slope of the line trend of each time series.**

[Figure]

**Figure 6.** Temporal variations in spatial average vertical (a) SW, (b) LW, and (c) net heat rating in all-sky condition from CloudSat during the monsoon season in the South Asian subcontinent from 2006 to 2015.

[Figure]

**Figure 7. Temporal variations in spatial average (a) cloud fraction and CRE, as well as spatial distribution of the temporal changes in average (b) SW CRE at the surface and (c) SSR from CloudSat during the monsoon season in the South Asian subcontinent from 2006 to 2015. The dashed lines indicate the linear fit line of the solid line with the same color; *p* is the significance level; *k* is the slope of the line trend of each time series.**

[Figure]

**Figure 8. Temporal variations in spatial average vertical RH in (a) all-sky and (c) clear-sky condition; spatial distribution of temporal changes in average PW in (b) all-sky and (d) clear-sky condition from ECMWF-AUX during the monsoon season in the South Asian subcontinent from 2006 to 2015.**

[Figure]

**Figure 9.** Temporal variations in spatial average (a) PW (blue line), as well as SSR from BUGSrad (red line) and CloudSat (orange line); spatial distribution of temporal changes in average (b) SSR from BUGSrad in clear-sky condition during the monsoon season in the South Asian subcontinent from 2006 to 2015. The dashed lines indicate the linear fit line of the solid line with the same color; *p* is the significance level; *k* is the slope of the line trend of each time series.

---

## Author Comment (AC2) · 23 Sep 2017

Dr. Zengxin PAN

State Key Laboratory of Information Engineering in Surveying, Mapping and Remote Sensing, Wuhan University, Wuhan 430079, China. Tel: (+86) 15107154425,

Email: pzx@whu.edu.cn

**Manuscript ID:** acp-2017-429 to ACP

**Title:** Estimating the effects of aerosol, cloud, and water vapor on the recent brightening during the South Asian monsoon season

**Author:** Feiyue Mao, Zengxin Pan*, Wei Wang, Xin Lu, Wei Gong.

Dear Anonymous Reviewer 2:

We greatly appreciate for your insightful comments and suggestions. We have revised our paper thoroughly in light of your comments. The changes have been highlighted in color in the manuscript, and also they are summarized below:

1. We have added a more thorough literature review about reasons and underlying mechanisms of brightening and dimming in Section 1;

2. The details of products used in this study have been added as the Table 1. Moreover, the analysis of uncertainty about the products have added at the introduction of every products in Section 2;

3. We defined the study domain within the South Asian subcontinent to restrict the effect of the Tibet Plateau and sea pixels on the results of this study (Figure 1);

4. The Conclusion have been expanded, and the content about related climate implications of brightening have been divided as the separate Section 3.4;

5. We have added the spatial distribution of the temporal changes in average PW in all-sky condition, as shown in Figure 8.

Please refer to the following point-to-point responses for more details.

Thank you for your time.

Sincerely,

Dr. Zengxin PAN

**General Comments:**

This manuscript examines the effect of aerosol, cloud, and water vapor variations on recent change of surface solar radiation and the associated dimming and brightening during monsoon season in India. Multi-year observations from A-Train satellite constellation have been used. The topic is quite interesting and important for the studied region. However, I have quit a few concerns that the authors need to address before it can be published.

**Major Comments:**

1. My biggest concern is that the authors failed to accurately present and connect the observational findings and tended to draw conclusions without providing further evidences. I listed some of these in the specific comments, e.g., # 22, 26, 27, 30, and so on.

   **RE:** Thank you for your suggestion. We have added the necessary description for all following corresponding questions.

2. The second one is about the *Abstract*. It should basically give the readers a clear and concise overview of the manuscript in terms of the aim, main points, and conclusions. However, the *Abstract* of this study is just repeating what's been stated in the *Conclusion*. For example, the first few sentences in Lines 12-17 are exactly the same as those in the beginning of *Conclusion*.

   **RE:** Thank you for your suggestion. We have revised the Abstract as the following:

   South Asia is experiencing a leveling-off trend of solar radiation and even a transition from dimming to brightening. As the only significant energy source for the global ecosystem, any change in incident solar radiation affects our habitats profoundly. This process is significantly complicated because of the active atmospheric action during the monsoon season. Here, we use observations from multiple sensors in the A-Train satellite constellation to evaluate the effects of aerosol, cloud, and water vapor variations on the recent changes in surface solar radiation during the monsoon season (June–September) in the South Asian subcontinent from 2006 to 2015. Results show that during this period, the surface shortwave radiation (SSR) and outgoing shortwave (SW) radiation increased by 16.2 W m$^{-2}$ and decreased by 13.0 W m$^{-2}$, respectively. However, the increase in spatial average aerosol optical depth is inconsistent with the variation in SSR in

the South Asian subcontinent. Instead, the decreases in the amount of water vapor and clouds significantly contributed to brightening, thus further affecting the surface warming in the South Asian subcontinent. Clouds generally reduced and thinned by approximately 9.4% and 182 m, respectively, with the decrease in cloud water path (by 53.4 g m$^{-2}$) and particle number concentration under cloud-sky condition. Given the change in the clouds, the atmospheric vertical SW heat rating decreased by 0.3–0.4 K d$^{-1}$ at an altitude of 5–15 km, whereas the SW heat rating slightly increased at the low-cloud and near-surface areas (below 4 km) from 2006 to 2015. The SW cloud radiative effect decreased by approximately 45.5 W m$^{-2}$ at the surface. Moreover, the precipitable water under clear-sky condition decreased by approximately 2.8 mm over the brightening period. Correspondingly, solar brightening increased by roughly 2.5 W m$^{-2}$ owing to the weakened absorption. Overall, the decreases in water vapor and clouds resulted in the increased absorption of direct solar radiation at the surface and subsequent surface brightening. Hence, brightening may play a prominent role in modulating the warming and rainfall variations in the South Asian subcontinent.

3. My third concern is that the authors need to provide a more thorough literature review about the global and regional change of surface solar radiation and the underlying mechanisms so that readers can have a better context to understand all the discussions followed.

   **RE:** Thank you for your suggestion. We have rewrote the review of surface solar radiation variation in the global and regional scale in Section 1 of the manuscript. However, due to the too long space of revised content, we are not shown in the Response. Please check in the revised manuscript, which is located at the end of this Response.

4. Another one is related to the uncertainty and limitation of data sources. I suggest the authors to rewrite the section of *Data and Methods*. It would be easier to follow if the authors could specify what variables from each dataset have been used after a general description is provided. And please also provide what are the main uncertainties of such variables, either from instrument/model itself or from the retrieval methods. Instead of just providing a reference, the authors need to expand it more including, but not limited to, how those uncertainties affect the current study.

**RE:** Thank you for your suggestion. We have added the corresponding table of products and variables used in this study in Table 1 as the following:

**Table 1. Observations used in analysis along with their sources and spatial resolutions.**

| Sensor | Spatial Resolution | Products | Parameter |
|---|---|---|---|
| CloudSat | 1.3 × 1.7 km | 2B-CLDCLASS | Cloud Fraction; cloud top/base height; |
| | | 2B-CWC-RO | Cloud water content, particle number concentration, and particle effective radius |
| | | 2B-FLXHR | Vertical heating rate; SW/LW radiative fluxes |
| CALIPSO | 5 × 5 km | Level 2 Aerosol Layer | Aerosol optical depth; vertical feature mask |
| MODIS | 1° × 1° | Level 3 MYD08 | Aerosol optical depth |
| ECMWF | 2.5° × 2.5° | ECMWF-AUX | Pressure; temperature; relative humidity |
| CERES | 20 × 20 km | Level 2 Single Scanner Footprint | SW radiation in TOA and Surface |

The analysis of intrinsic uncertainty about the products used in this study have added at the introduction of every products in Section 2. However, due to the too long space of revised content, we are not shown in the Response. Please check in the revised manuscript, which is located at the end of this Response.

5.  Last but not least, a terrain map showing the study domain should be provided before any spatial average results are shown. Also figure and panel numbers are missing a lot when the relevant results are discussed, which had made it difficult for me to follow.

**RE:** Thank you for your suggestion. We have added the terrain map and showed the study region of this study in Figure 1. Also, we have added the figure and panel numbers of the relevant results in the entire manuscript.

[Figure]

Figure 1. Terrain map during the monsoon season in the South Asia and surrounding regions. The blue line indicates rivers; the study region is within the red line.

**Minor comments:**

1. Abstract, 13: Add a sentence illustrating why this transition needs to be concerned. In other words, why is this study important?

**RE:** Thank you for your suggestion. We have revised the corresponding description in Abstract as the following:

As the only significant energy source for the global ecosystem, any change in incident solar radiation affects our habitats profoundly.

2. Abstract, 16: Add '*surface*' before '*solar radiation*'.

**RE:** Thank you for your suggestion. The 'surface' have been added.

3. Abstract, 18: Specify this is the spatial- and temporal-average '*increase*'.

**RE:** Thank you for your suggestion. We have revised the corresponding description in Abstract as the following:

However, the increase in spatial average aerosol optical depth is inconsistent with the variation in SSR in the South Asian subcontinent.

4. P2, 37: Add reference to support this statement.

**RE:** Thank you for your suggestion. We have added the corresponding reference to support this statement:

Haywood, J. M., Nicolas, B., Andy, J., Olivier, B., Martin, W., and Shine, K. P.: The roles of aerosol, water vapor and cloud in future global dimming/brightening, Journal of Geophysical Research Atmospheres, 116, 2011.

5. P2, 38: Be more specific: was this reduction global or regional and in what season did it occur?

**RE:** Thank you for your suggestion. We have revised the corresponding description at Paragraph 2 of Section 1 as the following:

Widespread reduction in the annual average surface solar radiation from the 1960s to the 1980s has been reported by many researchers at the global and regional scales, including those from America, Europe, and China (Liepert, 2002; Wild et al., 2005).

6. P2, 44: What region was this trend observed over?

**RE:** Thank you for your suggestion. We have revised the corresponding description at Paragraph 2 of Section 1 as the following:

Wild (2009) presented that the globally averaged trends in the 1980s typically reversed from dimming to brightening; the study reported trends of 2.2–6.6 W m$^{-2}$ per decade from the 1980s to the 2000s.

7. P2, 53: Add reference.

**RE:** Thank you for your suggestion. We have added the corresponding reference to support this statement as the following:

Soni, V. K., Pandithurai, G., and Pai, D. S.: Is there a transition of solar radiation from dimming to brightening over India?, Atmospheric Research, 169, 209-224, 2016.

8. P3, 78-79: Specify why only the changes of aerosols, clouds and water vapor are considered. This is the basis of the current study.

**RE:** Thank you for your suggestion. We have revised the corresponding description at Paragraph 4 of Section 1 as the following:

Here, we focus mainly on the dimming and brightening caused by changes in

aerosols, water vapor, and clouds, which are the dominant factors that alter atmospheric transparency, and thereby regulate the solar radiation incident on the Earth's surface (Wild, 2009).

9. P4, 90: Add reference.

**RE:** Thank you for your suggestion. We have added corresponding reference to support this statement as the following:

Mace, G. G., Zhang, Q., Vaughan, M., Marchand, R., Stephens, G., Trepte, C., and Winker, D.: A description of hydrometeor layer occurrence statistics derived from the first year of merged Cloudsat and CALIPSO data, Journal of Geophysical Research: Atmospheres, 114, 414-416, 10.1029/2007JD009755, 2009.

10. P4, 114: Write out what CERES stands for.

**RE:** Thank you for your suggestion. We have added the corresponding description at Paragraph 2 of Section 2.1 as the following:

'...Clouds and Earth's Radiant Energy System (CERES)…'.

11. P5, 118: Please be more specific and describe how it's relevant to this study.

**RE:** Thank you for your suggestion. This study mainly focus on the inter-annual change of radiation and its reasons in the South Asia. The large space- and time-scales average from satellites is favorable to ensure the accurate of radiative fluxes.

12. P5, 131-133: This is a good place to state how CALIPSO data is used in the current study and how the authors combine it with other datasets.

**RE:** Thank you for your suggestion. We have added the corresponding description at Paragraph 1 of Section 2.2 as the following:

In this study, CALIPSO Level 2 aerosol layer products are used to describe the recent changes in aerosol loading and its mostly sub-type component in the South Asian subcontinent. Moreover, the observations are compared with those of MODIS. CALIPSO is highly sensitive to thin cirrus clouds, but cannot easily penetrate optically thick layers with optical depths exceeding 3–5 (Hu et al., 2009). In this case, CALIPSO determines no or low aerosol optical depth (AOD) at the near-surface due to the complete attenuation of lidar signals. To exclude the effects of unpenetrated profiles for lidar, we calculate the average AOD from

CALIPSO only on the basis of the scene in which the lidar penetrates the entire atmosphere.

13. P5, 138: Add reference.

**RE:** Thank you for your suggestion. We have added the two corresponding reference to support this statement as the following:

Remer, L. A., Kaufman, Y. J., Tanré, D., Mattoo, S., Chu, D. A., Martins, J. V., Li, R. R., Ichoku, C., Levy, R. C., and Kleidman, R. G.: The MODIS Aerosol Algorithm, Products, and Validation, Journal of Atmospheric Sciences, 62, 947-973, 2005.

Wielicki, B. A., Barkstrom, B. R., Harrison, E. F., Iii, R. B. L., Smith, G. L., and Cooper, J. E.: Clouds and the Earth's Radiant Energy System (CERES): An Earth Observing System Experiment, Bulletin of the American Meteorological Society, 77, 853-868, 2015.

14. P6, Sec. 2.4: The authors described a radiative transfer model here. But what is more important is how this model is relevant to this study.

**RE:** Thank you for your suggestion. BUGSrad is the official radiative transfer model used by the product of CloudSat 2B-FLXHR. We have added the corresponding explanation about BUGSrad at Paragraph 5 of Section 2.5 as the following:

We accurately evaluate the effect of water vapor variations on solar radiation using the official radiative transfer model of CloudSat 2B-FLXHR (BUGSrad). By using BUGSrad, it is convenient to compare the results of the radiative transfer model with those of the CloudSat product. We use the enviromental parameters provided by CloudSat products as the input of the BUGSrad, including temperature, pressure, specific humidity and cloud mask, and so on.

15. P7, 173: Add reference for the ECMWF-AUX product.

**RE:** Thank you for your suggestion. We have added the two corresponding reference to support this statement as the following:

Uppala, S. M., Kållberg, P. W., Simmons, A. J., Andrae, U., Bechtold, V. D. C., Fiorino, M., Gibson, J. K., Haseler, J., Hernandez, A., and Kelly, G. A.: The ERA-40 re-analysis, Quarterly Journal of the Royal Meteorological Society, 131,

2961-3012, 2005.

16. P7, 195: Add reference.

**RE:** Thank you for your suggestion. We have added the two corresponding reference to support this statement as the following:

Henderson, D. S., L'Ecuyer, T., Stephens, G., Partain, P., and Sekiguchi, M.: A Multisensor Perspective on the Radiative Impacts of Clouds and Aerosols, Journal of Applied Meteorology & Climatology, 52, 853-871, 2013.

17. P7, 196: Define what the '*all-sky*' condition is.

**RE:** Thank you for your suggestion. We have revised the corresponding description about the definition of all-sky and clear-sky condition at Paragraph 2 of Section 2.5 as the following:

We define the cloud-sky condition for all products as the scenes in which the cloud mask from the CloudSat footprint exceeds 20, and the others are defined as clear-sky condition. Under this screened method of cloud, the probability of a false detection to CloudSat is less than 5% in comparison with that by CALIPSO (Mace and Zhang, 2014). The all-sky condition is the total of cloud-sky and clear-sky condition.

18. P8, 207: What is the reference for this equation? Are there any uncertainties and limitations to apply it? If the answer is yes, how does it affect this study?

**RE:** Thank you for your suggestion. We have added the corresponding reference about the equation of precipitable water as the following:

Bock, O., Keil, C., Richard, E., Flamant, C., and Bouin, M. N.: Validation of precipitable water from ECMWF model analyses with GPS and radiosonde data during the MAP SOP, Quarterly Journal of the Royal Meteorological Society, 131, 3013-3036, 2010.

Moreover, we have added the uncertainties of precipitable water from ECMWF-AUX at Paragraph 4 of Section 2.5 as the following:

The PW data from ECMWF-AUX have been verified with a bias of approximately −1 mm depending on water vapor amount by comparing observations from 21 ground-based Global Positioning System receiving stations and 14 radiosonde stations (Bock et al., 2010). Therefore, although the PW inferred from ECMWF-AUX is generally slightly lower than that under actual

condition, it can adequately indicate the change in water vapor amount in the South Asian subcontinent.

19. P8, 214-216: Be more specific.

   **RE:** Thank you for your suggestion. We have revised the corresponding sentence at Paragraph 5 of Section 2.5 as the following:

   For the radiative transfer model, we directly use the averaged environmental parameters during the monsoon season to evaluate the radiative effect of water vapor by considering the quantity of the calculation. Due to the difference of averaged methods which were used by CloudSat and BUGSrad to evaluate the radiative effect of water vapor, there may are some difference between the radiative fluxes calculated based on CloudSat and BUGSrad.

20. P8, 219: Figure 1 shows the time series of the spatial-average AOD. A terrain map with the studied domain should be provided first. Also explain in detail how the spatial and temporal average were done. Provide statistical analysis and show how significant the results are. Please do this for the other relevant results too. Also, the orange lines in Figure 1 are not mentioned in the text at all.

   **RE:** Thank you for your suggestion. We defined the study domain within the South Asian subcontinent to restrict the effect of the Tibet Plateau and sea pixels on the results of this study, as shown in Figure 1. We have added the terrain map and showed the study region of this study in Figure 1 as the following:

[Figure]

**Figure 1. Terrain map during the monsoon season in the South Asia and surrounding regions. The blue line indicates rivers; the study region is within the red line.**

Moreover, we have added the explanation about how to calculate the time series of the spatial-average parameter of aerosol, cloud and water vapor and their radiative effect at Paragraph 2 of Section 2.5 as the following:

For the method of spatial and temporal averages, we firstly obtain the spatial distribution of temporal average within a 1° × 1° grid during the monsoon season. We then determine the spatial average on the basis of the above temporal average in the entire South Asian subcontinent. For the spatial patterns of temporal changes, we identify the results of the linear temporal changes in atmospheric conditions during the monsoon season from 2006 to 2015.

Moreover, we have added the description about significant level of AOD and its significant contribution factor at Paragraph 2 of Section 3.1 as the following:

Figure 2b shows the consistent increase in AOD during the pre-monsoon, monsoon, and dry seasons, respectively. The increment of AOD during the pre-monsoon and dry seasons (by 0.08 and 0.11, respectively) were greater than those during the monsoon season according to the MODIS observations from

2006 to 2015. The significant level of AOD variation detected by MODIS was higher during non-monsoon season ($p<0.01$) than during the monsoon season, which mainly due to the disturbance of widespread clouds during the monsoon season (Remer et al., 2005). The change in the average AOD detected by CALIPSO showed a higher significance level ($p=0.07$) than that detected by MODIS due to the minimal disturbance of thin clouds and surface albedos to the aerosols detected by CALIPSO during the monsoon season (Redemann et al., 2012). Additionally, the robust and quick increase in aerosols in the South Asia has been verified by many previous studies (Suresh et al., 2013; Reddy et al., 2016; Srivastava, 2016).

We also have added the comments about the significant level of other results.

21. P9, 226: Explain why aerosols under clouds are missed by MODIS.

**RE:** Thank you for your suggestion. In the official algorithm, the MODIS only infer the aerosol loading under clear-sky. The aerosol distribution can be typically be considered homogeneous. Therefore, the aerosol loading detected by MODIS in clear-sky are assumed as the average representative for aerosol loading in all-sky. We have revised the corresponding sentence at Paragraph 1 of Section 3.1 as the following:

These phenomena are mainly due to the effect of aerosols under clouds, which can be detected by CALIPSO but missed by MODIS because of MODIS only inferring the aerosol loading under clear-sky condition (Winker et al., 2010).

22. P9, 235: Add figure panel number from which the observation is made. The significance of the trend needs to be discussed before this statement can be made.

**RE:** Thank you for your suggestion. We have added the comment about shortwave surface radiation at Paragraph 4 of Section 3.1 as the following:

Further, Figure 3a shows no distinct change in the SSR during the pre-monsoon season, even the reverse trend of SSR during the dry season (dimming) comparing to that during the monsoon season. The seasonal differences in OSR variations are similar to those of SSR (Figure 3b). Moreover, the trend of SSR during the dry season presented a higher significance level ($p<0.01$) than that during the monsoon season ($p=0.40$). Many reasons can contribute to the significance levels during different seasons. The average cloud fractions during the pre-monsoon and dry seasons are both approximately 20%, and much lower

than that during the monsoon season (53.4%) according to CloudSat observations. Moreover, under clear-sky condition, aerosols control the change in the solar radiation that reaches the surface (Nair et al., 2016; Soni et al., 2016). Therefore, aerosols have greater weighted impact on solar radiation during the dry season than during the monsoon season. In addition, the resultant effects of aerosol, cloud, and water vapor are considerably complicated and varied due to the intensive atmospheric activities during the monsoon season. The pre-monsoon season is the transition season from the dry season to the monsoon season, and associated with the establishment of the southwestern wind regime over the South Asian subcontinent (Das et al., 2015). The atmospheric condition and radiation variations are not distinct comparing to the trend of SSR between the dry and monsoon seasons.

23. P9, 240-243: I don't follow these two sentences. Please rewrite them.

**RE:** Thank you for your suggestion. We have revised the corresponding sentences at Paragraph 2 of Section 3.1 as the following:

There is the fact that the SSR is highly negatively correlated with aerosol loading (Folini and Wild, 2011). Therefore, the increase in AOD did not contribute to the recent brightening during the monsoon season in the South Asian subcontinent from 2006 to 2015.

24. P9, 247-249: The '*distribution*' cannot '*decrease*'! What level is this maximum decrease is associated and what does it indicate?

**RE:** Thank you for your suggestion. We have revised the corresponding sentences at Paragraph 1 of Section 3.2 as the following:

The cloud vertical frequency sequentially decreased during the monsoon season in the South Asian subcontinent from 2006 to 2015, with a maximum decrement of 7.1%, which contributed to approximately 30% of the maximum cloud vertical frequency at approximately 12 km m.s.l. (Figure 4a).

25. P10, 256-258: Is this consistent with previous studies?

**RE:** Thank you for your suggestion. Yes, this consistent with previous studies. We have added the corresponding reference about the change of cloud as described in the Question 27.

26. P10, 265-266: Provide evidence for this statement.

**RE:** Thank you for your suggestion. We have removed the corresponding sentence because this statement is unnecessary to the main point of the revised manuscript.

27. P10, 267-274: This paragraph needs to be expanded. What has caused such spatial variation for different cloud properties? Do the spatial patterns change year by year? How significant are the trends shown?

**RE:** Thank you for your suggestion. We have added the corresponding sentences about how to calculate the spatial patterns of temporal change at Paragraph 2 of Section 2.5 as the following:

For the spatial patterns of temporal changes, we identify the results of the linear temporal changes in atmospheric conditions during the monsoon season from 2006 to 2015.

Moreover, we have added the corresponding sentences about the cloud variations and its possible reasons at Paragraph 2 and 3 of Section 3.2 as the following:

As shown in Figure 5a, the total cloud fraction detected by CloudSat declined by 8.8% ($p$=0.07) during the monsoon season in the South Asian subcontinent from 2006 to 2015. Furthermore, the change in CWP in cloud-sky condition was consistent with that of the total cloud fraction with a decrement of 54.0 g m$^{-2}$ ($p$<0.01). The uppermost cloud top height (CTH) declined by 661 m ($p$=0.08), and the cloud geometrical depth (CGD) declined by 280 m ($p$=0.21) with partial fluctuations in 2009 and 2010. By contrast, an insignificant trend occurred in the lowermost cloud base height (CBH) variation.

The factors contributing to the changes in clouds are complicated and varied in the South Asian subcontinent. Ackerman et al. (2000) verified that aerosols with intensive absorption may lead to evaporation in the cloud layers (the aerosol semi-direct effect), burning off the clouds in the South Asia, including widespread dust, polluted dust, and smoke. Bollasina et al. (2011) showed the anthropogenic aerosol emissions weakened the South Asian summer monsoon, causing a decrease in the observed occurrence and amount of precipitation. In the global scale, observed and simulated cloud change patterns are consistent with the poleward retreat of mid-latitude storm tracks, which is attributed to the increasing greenhouse gas concentrations and recovery from volcanic radiative cooling (Norris et al., 2016). Actual, with the reduction of clouds, the recent weakened rainfall and dried monsoon during the summer monsoon season have been

verified by many previous studies (Bollasina et al., 2011; Turner and Annamalai, 2012; Annamalai et al., 2013). However, the decreasing clouds also cause the increase in SSR, possibly enhancing the generation of rainfall (Wild, 2009). Therefore, the weakened rainfall was not steady, and can be attributed to the resultant force of many factors, including the changes in monsoon intensity, intrinsic cloud properties, and anthropogenic aerosols, to name a few.

28. P11, 277: Again, how the '*all-sky*' condition is defined?

   **RE:** Thank you for your suggestion. We have revised the corresponding description about the definition of all-sky and clear-sky condition at Paragraph 2 of Section 2.5 as the following:

   We define the cloud-sky condition for all products as the scenes in which the cloud mask from the CloudSat footprint exceeds 20, and the others are defined as clear-sky condition. Under this screened method of cloud, the probability of a false detection to CloudSat is less than 5% in comparison with that by CALIPSO (Mace and Zhang, 2014). The all-sky condition is the total of cloud-sky and clear-sky condition.

29. P11, 278-283: How about the trends above 15 km?

   **RE:** Thank you for your suggestion. The clouds detected by CloudSat are mostly located below 15 km. We have added the corresponding description about the change of heat rating above 15 km at Paragraph 4 of Section 3.2 as the following:

   Due to the extremely little cloud detected by CloudSat above 15 km, no significant change in heat rating occurred above 15 km, as shown in the cloud vertical frequency distribution in Figure 4a.

30. P10-P11, 281-283 and 286-288: Please provide evidence for these statements.

   **RE:** Thank you for your suggestion. Cloud is dominant role on vertically atmospheric heat rating (Liou, 2002). With the reduction of cloud, the total cloud reflection weaken consistently, enhancing atmospheric transmission, and generating more SW radiation reach to the near surface. Correspondingly, total cloud LW absorption emitted from surface decrease with the reduction of cloud, causing the weakened LW heating rate within cloud layer (mainly below 10km). We have revised the corresponding description at Paragraph 4 of Section 3.2 as the following:

   With the reduction of clouds, the total cloud reflection and absorption (mainly

located at 5–15 km) weakened consistently, thus enhancing atmospheric transmission, causing more SW radiation than before to reach the near-surface area (Henderson et al., 2013). Due to the extremely little cloud detected by CloudSat above 15 km, no significant change in heat rating occurred above 15 km, as shown in the cloud vertical frequency distribution in Figure 4a. However, the change in LW vertical heating rate was not distinct compared with that of SW. Correspondingly, the LW radiative heating increased by less than 0.2 K d$^{-1}$ above 10 km, whereas that of heating decreased by 0.1–0.2 K d$^{-1}$ within the cloud, especially below 4 km, from 2006 to 2015. The total cloud LW absorption emitted from the surface decreased with the reduction of clouds, thus leading to a weakened LW heating rate within the cloud layer (mainly below 10 km), and more LW emission on the top cloud layer above 10 km (L'Ecuyer et al., 2008).

31. P11, 284: I cannot see the similarity here.

    **RE:** Thank you for your suggestion. We have removed the corresponding statement.

32. P11, 299-300: Expand this with more details.

    **RE:** Thank you for your suggestion. We have revised the corresponding description at Paragraph 5 of Section 3.2 as the following:

    In general, clouds were reduced and thinned by approximately 8.8% and 280 m with the decrease in cloud water content by 54.0 g m$^{-2}$ and particle number concentration in cloud-sky condition. Consequently, the SW radiative effect of the clouds decreased by approximately 42.1 W m$^{-2}$, and the absorption of direct solar radiation at the surface increased, thus leading to subsequent surface brightening.

33. P12, 311-312: This would be better seen if the difference between the two had been plotted.

    **RE:** Thank you for your suggestion. The spatial variation of PW in all-sky condition is highly consistent with that in clear-sky condition during monsoon season in the South Asian subcontinent from 2006 to 2015. We have added the spatial distribution of the temporal changes of average PW in all-sky at Figure 8 as the following:

[Figure]

**Figure 8. Temporal variations in spatial average vertical RH in (a) all-sky and (c) clear-sky condition; spatial distribution of temporal changes in average PW in (b) all-sky and (d) clear-sky condition from ECMWF-AUX during the monsoon season in the South Asian subcontinent from 2006 to 2015.**

34.  P12, 314-317: What has caused the spatial variation of PW?

**RE:** Thank you for your suggestion. The atmospheric water vapor variation is highly correlated with the change of monsoon intensity in monsoon region, especially in monsoon season. We have added the possible reason about the change of water vapor at Paragraph 1 of Section 3.3 as the following:

The decrease in atmospheric water vapor may be attributed to the weakened monsoon intensity in the recent decade during the monsoon season in the South Asia (Turner and Annamalai, 2012).

35.  P12, 330-332: Describe and compare in more details the direct and indirect

effects of water vapor.

**RE:** Thank you for your suggestion. We have revised the corresponding sentence at Paragraph 2 of Section 3.3 as the following:

The spatial variation in SSR under clear-sky condition was highly negative and consistent with that of PW, especially in the western and central regions of the South Asian subcontinent (Figure 9b), which partly consist of deserts. Moreover, in all-sky condition, there are the indirect effects of water vapor, which is named as water vapor-cloud interaction. Generally, aside from the effect of temperature, a high concentration of water vapor favors the generation of more or stronger convective clouds and results in decreased radiation at the surface, and vice versa (Yang et al., 2012). Additionally, water vapor regulates cloud amount by affecting the aerosol-cloud interaction, which generally shows a positive relationship under moist atmospheric conditions (Chen et al., 2014). Therefore, the impact of water vapor in all-sky condition is more complicated than that in clear-sky condition owing to the direct and indirect effects of water vapor on solar radiation.

36. P12, 333: What do you mean by '*stronger clouds*'?

**RE:** Thank you for your suggestion. The 'stronger clouds' have been revised to 'stronger convective clouds'.

37. Conclusion: This section needs to be expanded. Can the methodology used in this study be applied for those in other regions? How will the findings benefit the representation of surface solar heating in climate models? Are there any potential implications for climate variations?

**RE:** Thank you for your suggestion. For responding the above considerations, this Conclusion have been expanded and divide as the separate Section 3.4, as shown in following:

Variations in surface solar radiation can potentially significantly impact the climate system based on the long temporal correlation between solar radiation and climate factors, especially in temperature and rainfall. Observed dimming was suggested to be responsible for the absence of a significant temperature rise between the 1950s and the 1980s in various parts of the world, such as in the Arctic, China, America, and India (Ramanathan et al., 2005; Wild, 2009). However, the suppression of global warming over global land surfaces only lasted into the 1980s with the transition of dimming to brightening (Wild et al., 2005).

This condition indicates that brightening might have significantly contributed to the recent rapid warming after the 1980s according to ground-based observations (Philipona et al., 2009). This temperature evolution satisfactorily fits the observational surface solar radiation variations, and points to the crucial role that dimming and brightening may play in determining global warming. Moreover, surface solar radiation variations induce changes in the surface net radiation, thereby altering the energy available for evaporation, which equals precipitation in the global annual mean. A decrease in evaporation and the same globally averaged reduction in precipitation with the globally averaged dimming occurred in the 1980s (Liepert, 2002). Wild et al. (2009) showed that the increase in available surface energy is quantitatively consistent with the observed substantial increase in land precipitation (3.5 mm $y^{-1}$ between 1986 and 2000) and the associated intensification of the land-based hydrological cycle.

The recent rapid warming in South Asia has been verified by many studies (Turner and Annamalai, 2012; Annamalai et al., 2013). Moreover, ECMWF-AUX shows that the 2 m temperature increased by 0.14 K $y^{-1}$, even for surface temperature with the increment of 0.4 K $y^{-1}$ during the monsoon season in the South Asian subcontinent from 2006 to 2015 (FIG 1 of the Response, not shown in the manuscript). Syed et al. (2014) predicted that the mean surface air temperature in the monsoon season would increase from 2.5 °C to 5 °C with increasing greenhouse gas concentrations by the end of the century. Models may underestimate the rate of global warming due to the insufficient consideration of dimming and brightening (Wild, 2012). With brightening considered, warming may accelerate in South Asia at unprecedented rates. However, the recent monsoon rainfall roughly weakened with no homogeneity by combining the model and observations of satellites and sites (Bollasina et al., 2011; Turner and Annamalai, 2012; Annamalai et al., 2013). Reduced clouds may indicate a decrease in average rainfall during the monsoon season, whereas brightening generally enhances the generation of rainfall. Turner and Annamalai (2012) showed consistent negative trends of rainfall over northwest India and coastal Burma, while the positive trends in southeast India; they also found a distinct disagreement among different rainfall data in northeast India. In general, the recent brightening may play a prominent role in modulating the warming and rainfall variations in the South Asian subcontinent.

[Figure]

**FIG 1 of the Response. Temporal variation of spatial average skin and 2-m temperature (red and blue lines) from ECMWF-AUX during monsoon season during monsoon season in the South Asian subcontinent from 2006 to 2015. The dashed lines indicate the linear fit line of the solid line with same color; p is the significance level; k is the slope of the line trend of each time series.**

**References**

Ackerman, A. S., Toon, O. B., Stevens, D. E., Heymsfield, A. J., Ramanathan, V., and Welton, E. J.: Reduction of tropical cloudiness by soot, Science, 288, 1042-1047, 2000.

Annamalai, H., Hafner, J., Sooraj, K. P., and Pillai, P.: Global Warming Shifts the Monsoon Circulation, Drying South Asia, Journal of Climate, 26, 2701-2718, 2013.

Bock, O., Keil, C., Richard, E., Flamant, C., and Bouin, M. N.: Validation of precipitable water from ECMWF model analyses with GPS and radiosonde data during the MAP SOP, Quarterly Journal of the Royal Meteorological Society, 131, 3013-3036, 2010.

Bollasina, M. A., Ming, Y., and Ramaswamy, V.: Anthropogenic Aerosols and the Weakening of the South Asian Summer Monsoon, Science, 334, 502-505, 2011.

Chen, Y. C., Christensen, M. W., Stephens, G. L., and Seinfeld, J. H.: Satellite-based estimate of global aerosol-cloud radiative forcing by marine warm clouds, Nature Geoscience, 7, 643-646, 2014.

Das, S., Dey, S., Dash, S. K., Giuliani, G., and Solmon, F.: Dust aerosol feedback on the Indian summer monsoon: Sensitivity to absorption property, Journal of Geophysical Research Atmospheres, 120, 9642-9652, 2015.

Folini, D., and Wild, M.: Aerosol emissions and dimming/brightening in Europe: Sensitivity studies with ECHAM5-HAM, Journal of Geophysical Research Atmospheres, 116, 21104, 2011.

Henderson, D. S., L'Ecuyer, T., Stephens, G., Partain, P., and Sekiguchi, M.: A Multisensor Perspective on the Radiative Impacts of Clouds and Aerosols, Journal of Applied Meteorology & Climatology, 52, 853-871, 2013.

Hu, Y., Winker, D., Vaughan, M., Lin, B., Omar, A., Trepte, C., Flittner, D., Yang, P., Nasiri, S. L., and Baum, B.: CALIPSO/CALIOP Cloud Phase Discrimination Algorithm, Journal of Atmospheric & Oceanic Technology, 26, 2293-2309, 2009.

L'Ecuyer, T. S., Wood, N. B., Haladay, T., Stephens, G. L., and Stackhouse, P. W.: Impact of clouds on atmospheric heating based on the R04 CloudSat fluxes and heating rates data set, Journal of Geophysical Research Atmospheres, 113, 2013-2018, 2008.

Liepert, B. G.: Observed reductions of surface solar radiation at sites in the United States and worldwide from 1961 to 1990, Geophysical Research Letters, 29, 61-61–61-64, 2002.

Liou, K.-N.: An Introduction to Atmospheric Radiation, Academic press, San Diego, USA, 2002.

Mace, G. G., and Zhang, Q.: The CloudSat radar-lidar geometrical profile product (RL-GeoProf): Updates, improvements, and selected results, Journal of Geophysical Research: Atmospheres, 119, 9441-9462, 2014.

Nair, V. S., Babu, S. S., Manoj, M. R., Moorthy, K. K., and Chin, M.: Direct radiative effects of aerosols over South Asia from observations and modeling, Climate Dynamics, 1-18, 2016.

Norris, J. R., Allen, R. J., Evan, A. T., Zelinka, M. D., O'Dell, C. W., and Klein, S. A.: Evidence for climate change in the satellite cloud record, Nature, 536, 72, 2016.

Philipona, R., Behrens, K., and Ruckstuhl, C.: How declining aerosols and rising greenhouse gases forced rapid warming in Europe since the 1980s, Geophysical Research Letters, 36, 206-218, 2009.

Ramanathan, V., Chung, C., Kim, D., Bettge, T., Buja, L., Kiehl, J., Washington, W., Fu, Q., Sikka, D., and Wild, M.: Atmospheric brown clouds: Impacts on South Asian climate and hydrological cycle, Proceedings of the National Academy of Sciences, 102, 5326-5333, 2005.

Reddy, K. R. O., Balakrishnaiah, G., Gopal, K. R., Reddy, N. S. K., Rao, T. C., Reddy, T. L., Hussain, S. N., Reddy, M. V., Reddy, R. R., and Boreddy, S. K. R.: Long term (2007–2013) observations of columnar aerosol optical properties and retrieved size distributions over Anantapur, India using multi

wavelength solar radiometer, Atmospheric Environment, 142, 238-250, 2016.

Redemann, J., Vaughan, M. A., Zhang, Q., Shinozuka, Y., Russell, P. B., Livingston, J. M., Kacenelenbogen, M., and Remer, L. A.: The comparison of MODIS-Aqua (C5) and CALIOP (V2 & V3) aerosol optical depth, Atmospheric Chemistry and Physics, 12, 3025-3043, 2012.

Remer, L. A., Kaufman, Y. J., Tanré, D., Mattoo, S., Chu, D. A., Martins, J. V., Li, R. R., Ichoku, C., Levy, R. C., and Kleidman, R. G.: The MODIS Aerosol Algorithm, Products, and Validation, Journal of Atmospheric Sciences, 62, 947-973, 2005.

Soni, V. K., Pandithurai, G., and Pai, D. S.: Is there a transition of solar radiation from dimming to brightening over India?, Atmospheric Research, 169, 209-224, 2016.

Srivastava, R.: Trends in aerosol optical properties over South Asia, International Journal of Climatology, 37, n/a-n/a, 2016.

Suresh, B. S., Manoj, M. R., Krishna, M. K., Gogoi, M. M., Nair, V. S., Kumar, K. S., Satheesh, S. K., Niranjan, K., Ramagopal, K., and Bhuyan, P. K.: Trends in aerosol optical depth over Indian region: Potential causes and impact indicators, Journal of Geophysical Research Atmospheres, 118, 11,794–711,806, 2013.

Syed, F. S., Iqbal, W., Syed, A. A. B., and Rasul, G.: Uncertainties in the regional climate models simulations of South-Asian summer monsoon and climate change, Climate Dynamics, 42, 2079-2097, 2014.

Turner, A. G., and Annamalai, H.: Climate change and the South Asian summer monsoon, Nature Climate Change, 2, 587-595, 2012.

Wild, M., Gilgen, H., Roesch, A., Ohmura, A., Long, C. N., Dutton, E. G., Forgan, B., Kallis, A., Russak, V., and Tsvetkov, A.: From dimming to brightening: decadal changes in solar radiation at Earth's surface, Science, 308, 847-850, 2005.

Wild, M.: Global dimming and brightening: A review, Journal of Geophysical Research Atmospheres, 114, D00D16, 2009.

Wild, M., Trüssel, B., Ohmura, A., Long, C. N., König‐Langlo, G., Dutton, E. G., and Anatoly, T.: Global dimming and brightening: An update beyond 2000, Journal of Geophysical Research Atmospheres, 114, 895-896, 2009.

Wild, M.: Enlightening Global Dimming and Brightening, Bulletin of the American Meteorological Society, 93, 27-37, 2012.

Winker, D. M., Pelon, J., Coakley, J. A., Ackerman, S. A., Charlson, R. J., Colarco, P. R., Flamant, P., Fu, Q., Hoff, R. M., and Kittaka, C.: The CALIPSO Mission: A Global 3D View of Aerosols and Clouds, Bulletin of the American Meteorological Society, 91, 1211-1229, 2010.

Yang, K., Ding, B., Qin, J., Tang, W., Ning, L., and Lin, C.: Can aerosol loading explain the solar dimming over the Tibetan Plateau, Geophysical Research Letters, 39, L20710, 2012.

**Estimating the effects of aerosol, cloud, and water vapor on the recent brightening during the South Asian monsoon season**

Feiyue Mao [1, 2, 3], Zengxin Pan [2, *], Wei Wang [2], Xin Lu [2], Wei Gong [2, 3]

[1] School of Remote Sensing and Information Engineering, Wuhan University, Wuhan 430079, China
[2] State Key Laboratory of Information Engineering in Surveying, Mapping and Remote Sensing, Wuhan University, Wuhan 430079, China
[3] Collaborative Innovation Center for Geospatial Technology, Wuhan 430079, China

*Correspondence to*: Zengxin Pan (pzx@whu.edu.cn)

**Abstract.** South Asia is experiencing a leveling-off trend of solar radiation and even a transition from dimming to brightening. As the only significant energy source for the global ecosystem, any change in incident solar radiation affects our habitats profoundly. This process is significantly complicated because of the active atmospheric action during the monsoon season. Here, we use observations from multiple sensors in the A-Train satellite constellation to evaluate the effects of aerosol, cloud, and water vapor variations on the recent changes in surface solar radiation during the monsoon season (June–September) in the South Asian subcontinent from 2006 to 2015. Results show that during this period, the surface shortwave radiation (SSR) and outgoing shortwave (SW) radiation increased by 16.2 W m$^{-2}$ and decreased by 13.0 W m$^{-2}$, respectively. However, the increase in spatial average aerosol optical depth is inconsistent with the variation in SSR in the South Asian subcontinent. Instead, the decreases in the amount of water vapor and clouds significantly contributed to brightening, thus further affecting the surface warming in the South Asian subcontinent. Clouds generally reduced and thinned by approximately 9.4% and 182 m, respectively, with the decrease in cloud water path (by 53.4 g m$^{-2}$) and particle number concentration under cloud-sky condition. Given the change in the clouds, the atmospheric vertical SW heat rating decreased by 0.3–0.4 K d$^{-1}$ at an altitude of 5–15 km, whereas the SW heat rating slightly increased at the low-cloud and near-surface areas (below 4 km) from 2006 to 2015. The SW cloud radiative effect decreased by approximately 45.5 W m$^{-2}$ at the surface. Moreover, the precipitable water under clear-sky condition decreased by approximately 2.8 mm over the brightening period. Correspondingly, solar brightening increased by roughly 2.5 W m$^{-2}$ owing to the weakened absorption. Overall, the decreases in water vapor and clouds resulted in the increased absorption of direct solar radiation at the surface and subsequent surface brightening. Hence, brightening may play a prominent role in modulating the warming and rainfall variations in the South Asian subcontinent.

**1 Transition from Dimming to Brightening in South Asia**

Solar radiation incidence on the Earth's surface plays a fundamental determinant role in climate and life on our planet (IPCC, 2013). Surface solar radiation is a major component of the surface energy balance and governs many diverse surface processes, such as evaporation and associated hydrological components, snow and glacier melting, plant photosynthesis and related terrestrial carbon uptake, as well as the diurnal and seasonal courses of surface temperatures. Negative trends in the downwelling surface solar radiation are collectively called "dimming," whereas positive trends are called "brightening" (Wild et al., 2005). Any change in the amount of solar radiation profoundly affects the temperature field, atmospheric and oceanic general circulation, and hydrological cycle (Haywood et al., 2011).

Widespread reduction in the annual average surface solar radiation from the 1960s to the 1980s has been reported by many researchers at the global and regional scales, including those from America, Europe, and China (Liepert, 2002; Wild et al., 2005). Subsequently, the term "brightening" was coined to emphasize that global solar radiation no longer declines at many sites after the late 1980s (Wild et al., 2005). Long et al. (2009) found that solar dimming reversed at an increasing trend of 6 W m$^{-2}$ per decade in the continental United States in 1995–2007. Wild (2009) presented that the globally averaged trends in the 1980s typically reversed from dimming to brightening; the study reported trends of 2.2–6.6 W m$^{-2}$ per decade from the 1980s to the 2000s. However, recent studies indicate that the developments in dimming and brightening after 2000 show mixed tendencies. Wild et al. (2009) reported a continuation of brightening at sites in Europe, United States, and parts of Asia, a leveling-off at sites in Japan and Antarctica, and indications of renewed dimming in China. Conversely, the most recent related study shows that brightening has been continuing in China after 2000 (Wang and Wild, 2016).

South Asia is endowed with abundant solar energy because of its geographic position in the tropical belt (Soni et al., 2016). Dimming or brightening in South Asia is more evident and complicated than that in other regions, and its effects on regional and global climate and ecosystem are amplified by the monsoon circulation in the country (Padma et al., 2007; Wild, 2012). The South Asian monsoon occurs between June and September, which correspond to the monsoon onset and end times, respectively (Turner and Annamalai, 2012). Contrary to the variable trend in surface solar radiation in other regions globally, Padma et al. (2007) found that continued dimming of −8.6 W m$^{-2}$ per decade occurred in India from 1981 to 2004 according to 12 stations over the Indian region. Wild et al. (2009) proposed to focus on the slight tendency toward the stabilization of surface solar radiation since the late 1990s. Furthermore, Soni et al. (2016) observed a trend reversal and partial recovery from dimming to brightening in India around 2001.

Various mechanisms can potentially contribute to dimming and brightening. Changes in surface solar radiation can be caused by either external changes in the amount of solar radiation incident on the planet at the top of atmosphere (TOA) or internal changes (within the climate system) in the transparency of the atmosphere, which modifies the solar beam on its way to the Earth's surface. However, Willson and Mordvinov (2003) verified that the decadal dimming or brightening cannot be explained by changes in the luminosity of the sun because these changes are at least an order of magnitude smaller than

those in climate system. Therefore, the observed dimming or brightening has to originate from alterations in the transparency of the atmosphere, which depends on the presence of clouds, aerosols, and radiatively active gases, particularly water vapor, which is a strong absorber of solar radiation (Kim and Ramanathan, 2008).

Kvalevåg and Myhre (2007) concluded that the major contributor to dimming is aerosols ($-2.4$ W m$^{-2}$) and that the secondary effect is the increase in gas concentrations ($-0.64$ W m$^{-2}$), including those of tropospheric ozone and water vapor, since pre-industrial times; they also identified $NO_2$, $CH_4$, and $CO_2$ as minor contributors. Many studies have proposed that the changes in atmospheric aerosol loading, cloud cover, and cloud properties are the main factors that determine solar dimming and brightening (Wild, 2009; Soni et al., 2016). Liepert (2002) showed that the decrease in global radiation in 1961–1990 is attributable to the increases in cloud optical thickness and the direct effects of aerosols, which reduced solar radiation by 18 and 8 W m$^{-2}$, respectively. Kambezidis et al. (2012) argued that the decline in surface solar radiation in South Asia is attributable to the increase in the amount of anthropogenic aerosols during the last 30 years of the 20th century. This deduction is supported by the agreement between the observed decadal changes in anthropogenic aerosol emissions and trends in global solar radiation (Wild, 2009; Folini and Wild, 2011). Moreover, solar dimming and brightening may be of local or regional nature and is unavoidably influenced by regional sources and meteorology (Soni et al., 2016). Here, we focus mainly on the dimming and brightening caused by changes in aerosols, water vapor, and clouds, which are the dominant factors that alter atmospheric transparency, and thereby regulate the solar radiation incident on the Earth's surface (Wild, 2009).

In this study, we use observations from multiple sensors in the A-Train satellite constellation to evaluate the effects of aerosol, cloud, and water vapor variations on the recent changes in solar radiation at the surface during the monsoon season (June–September) in the South Asian subcontinent from 2006 to 2015. This study mainly aims to identify the possible reasons for the changes in dimming and brightening and assess the connection between the variations in aerosols, clouds, and water vapor and the recent brightening in the South Asian subcontinent.

**2 Data and Methods**

**2.1 CloudSat data**

Cloud Profile Radar (CPR) is an active millimeter-wave radar and is the only instrument on CloudSat, which was launched into the A-Train constellation in April 2006 (Mace et al., 2009). CPR has a 1.3 km cross-track and a 1.7 km along-track footprint resolution, and its effective vertical resolution at nadir is 240 m (Pan et al., 2017). CloudSat CPR is well suited for sensing a wide variety of cloud systems, from cirrus and stratus to deep convective systems, and shows slight sensitivity to the time of day or season (Rajeevan et al., 2012). However, the estimated operational sensitivity of CloudSat CPR ($-32$ dBZ to $-30$ dBZ) is insufficient for observing thin cirrus clouds with low ice water content (IWC) (Mace et al., 2009). CloudSat can infer the cloud microphysical characteristics on the basis of the backscatter return signals measured by

CloudSat CPR, including cloud particle number concentration, size, shape, and phase (Heymsfield et al., 2010). Further, by combining with a broadband radiative flux model known as BUGSRad, CloudSat can be used to quantify the three-dimensional (3D) information of cloud macrophysical and microphysical characteristics, as well as the corresponding cloud radiative effect (CRE) (L'Ecuyer et al., 2008).

5     The CloudSat Radar-Only Cloud Water Content (2B-CWC-RO) product contains retrieved estimates of cloud liquid water content (LWC) and IWC, cloud liquid number concentration (LNC) and ice particle number concentration (INC), cloud liquid effective radius (LER) and ice particle effective radius (IER), and related quantities for each radar profile measured by CPR on CloudSat (Woods et al., 2008). Considering the insensitivity of CloudSat to thin cirrus, Austin et al. (2009) showed that 2B-CWC-RO produces typical biases in IWC of −40% to +25% versus the observationally derived

10    synthetic data. However, L'Ecuyer et al. (2008) proposed that the impact of the thin cirrus that is not detected by CloudSat on shortwave (SW) radiation at the surface is −1.2 W m$^{-2}$ in a globally averaged experiment on radiative fluxes; this value is much smaller than the impact value of low clouds. Moreover, Christensen et al. (2013) found that CloudSat-derived cloud liquid water path (LWP) is generally too high and systematically exceed those from Moderate Resolution Imaging Spectroradiometer (MODIS) by approximately 50%. Most previous studies have verified the CloudSat 2B-CWC-RO product,

15    thereby allowing us to discuss the 3D microphysical characteristics of clouds in detail (Austin et al., 2009; Rajeevan et al., 2012). Therefore, the products containing 3D cloud characteristics from CloudSat are significant in evaluating cloud characteristics and their variations.

     Furthermore, retrieved profiles of cloud microphysical properties form the basis of the algorithm of another data product (2B-FLXHR), which consists of high vertical resolution profiles of radiative fluxes and atmospheric heating rates. L'Ecuyer

20    et al. (2008) detected biases between the radiative data of 2B-FLXHR from CloudSat and those from Clouds and Earth's Radiant Energy System (CERES) with monthly 5° means in the global scale. The biases of outgoing shortwave radiation (OSR), outgoing longwave (LW) radiation, surface shortwave radiation (SSR), and surface LW radiation are less than 0.1, 5.5, 13, and 16 W m$^{-2}$, respectively. Fortunately, the uncertainties in 2B-FLXHR fluxes decrease significantly for long time scale averages (L'Ecuyer et al., 2008). Therefore, the CRE derived from CloudSat can credibly describe the radiative effect

25    of clouds, especially in large space and time scales.

**2.2 CALIPSO data**

     Cloud-Aerosol Lidar and Infrared Pathfinder Satellite Observations (CALIPSO) was also launched in April 2006 (Winker et al., 2007). CALIPSO mainly loads Cloud Aerosol Lidar with Orthogonal Polarization (CALIOP), which is an excellent active two-wavelength (532 and 1064 nm) polarization lidar (Winker et al., 2007). The layers detected by CALIOP

30    are correctly identified with a high degree of confidence (> 90%) by cloud–aerosol mask analysis (Liu et al., 2009). Moreover, aerosols detected by CALIOP can be classified into the following six sub-types with high accuracy: clean marine, dust, polluted continental, clean continental, polluted dust, and smoke (Omar et al., 2009). Therefore, CALIPSO provides 3D

scientific materials for and perspective on studying the changes, interactions, and transportation of aerosols and clouds at a global scale (Winker et al., 2010; Pan et al., 2015). In this study, CALIPSO Level 2 aerosol layer products are used to describe the recent changes in aerosol loading and its mostly sub-type component in the South Asian subcontinent. Moreover, the observations are compared with those of MODIS. CALIPSO is highly sensitive to thin cirrus clouds, but cannot easily

5 penetrate optically thick layers with optical depths exceeding 3–5 (Hu et al., 2009). In this case, CALIPSO determines no or low aerosol optical depth (AOD) at the near-surface due to the complete attenuation of lidar signals. To exclude the effects of unpenetrated profiles for lidar, we calculate the average AOD from CALIPSO only on the basis of the scene in which the lidar penetrates the entire atmosphere.

**2.3 CERES/MODIS data**

10 Aqua became a member of the A-Train constellation in May 2002, with the load of CERES, MODIS, and four other sensors (Remer et al., 2005; Wielicki et al., 2015). CERES/Aqua Edition 4A Single Scanner Footprint (SSF) and MODIS/Aqua Edition 6 Atmosphere Level 3 Joint Products (MYD08) data are used in this study. CERES SSF data sets combine CERES radiation measurements, cloud and aerosol microphysical property retrievals based on observations of MODIS, and ancillary meteorology fields to form a comprehensive, high-quality compilation of satellite-derived cloud,

15 aerosol, and radiation budget information for radiation and climate studies (Loeb and Manalosmith, 2005). CERES reports the observed SW and LW solar radiances with precisions approaching 1.0% (Loeb and Manalosmith, 2005). Moreover, this study uses land–ocean AOD at 0.55 μm from MYD08. On the basis of its comparison with the Aerosol Robotic Network, the MODIS AOD has been verified with absolute and relative errors of 0.05 and 15%, respectively, mainly from the retrieval algorithm (Levy et al., 2013; Wang et al., 2017). AOD data have been widely used and verified by many studies because of

20 their excellent capability of deriving AOD from dark to bright surfaces (Remer et al., 2005; Munchak et al., 2013). The most significant effects of cloud contamination on passive MODIS and CERES have to be emphasized. For example, artificially high AOD can be detected by MODIS due to the presence of thin clouds undetected by MODIS (Grandey et al., 2013). Moreover, the scattering of light by inhomogeneous clouds possibly generates uncertainty in the passive wide-field observations of MODIS and CERES, which are mainly caused by broken cloud systems (Grandey et al., 2013; Christensen

25 et al., 2016). Therefore, the uncertainties of MODIS and CERES observations are more complicated in widespread-cloud conditions than in insufficient-cloud conditions.

**2.4 Radiative transfer model**

BUGSrad is the official radiative transfer model used by the product of CloudSat 2B-FLXHR (Fu and Liou, 1992; L'Ecuyer et al., 2008). This model is based on the two-stream doubling-adding solution to the radiative transfer equation

30 with the assumption of a plane-parallel atmosphere (Stephens et al., 2001). BUGSrad computes molecular absorption and scattering properties on the basis of correlated-k formulation (Fu and Liou, 1992). The calculation of BUGSrad is parallelly applied over six SW bands, and a constant hemisphere formulation is applied to 12 LW bands. These bands are appropriately

weighted and combined into the two broadband flux estimates that are ultimately reported, one covering the SW at 0–4 μm and the other covering the LW above 4 μm. According to the comparison experiment with observations of the Atmospheric Remotely-Sensed Clouds Locations of the Atmospheric Radiation Measurement program, the mean biases of SW and LW in clear-sky are 1.2 and 2.2 W m$^{-2}$ at TOA, respectively (Stephens et al., 2001; Stephens et al., 2003).

**2.5 Principle and methods**

In this study, multiple sensors from A-Train are used to investigate the possible reasons for the dimming and brightening variations during the monsoon season (June–September) in the South Asian subcontinent from 2006 to 2015. We define the study domain within the South Asian subcontinent to restrict the effect of the surrounding land and sea pixels on the results of this study, as shown in Figure 1. CloudSat, CALIPSO, and Aqua in A-Train are maintained in a tight orbital coordination within 176 s (Stephens et al., 2017). This setting allows near-simultaneous observations of a wide variety of atmospheric and surface parameters, facilitating the comparison of satellite between each other, allowing for even more comprehensive studies of climate. The MYD08 and Level 2 aerosol layer products from MODIS and CALIPSO are used to describe the recent changes in aerosol loading in the South Asian subcontinent. The SSF product from CERES/Aqua is considered for evaluating the radiation energy variations. The CloudSat 2B-CLDCLASS, 2B-CWC-RO, and 2B-FLXHR products are used to quantify the 3D changes in cloud macrophysical and microphysical characteristics and corresponding variable radiative effects of clouds. Moreover, Environmental conditions are obtained from the European Center for Medium-range Weather Forecast-AUXiliary analysis (ECMWF-AUX) product (Uppala et al., 2005). The CRE inferred from CERES is always disturbed by water vapor, aerosol, and the limited capability of MODIS to detect thin cirrus; furthermore, CERES considers the impact of multi-layer clouds insufficiently (Sohn et al., 2010). Therefore, this study uses the CloudSat 2B-FLXHR to evaluate the effects of cloud variations on radiation.

This study focuses on the effect of aerosol, cloud, and water vapor variations during the monsoon season. Due to the increased convective activity, the changes in aerosol, cloud, and water vapor and their impact on solar radiation possibly occur more obviously during the monsoon season than during non-monsoon seasons (Turner and Annamalai, 2012). Additionally, the information about aerosol, cloud, and water vapor is calculated on the basis of instantaneous observations, which we then use to ensure consistent conditions for the observations of multiple satellites. The satellites or sensors of the A-Train cross the equator in the daytime at 13:30 local time, at which point the incoming solar radiation is maximum within the day (Liou, 2002). Therefore, the instantaneous observations of radiative fluxes used in this study are greater than the daily average. The results are more representative of the atmospheric condition variations at nominally 13:30 local time than the daily average. For the method of spatial and temporal averages, we firstly obtain the spatial distribution of temporal average within a 1° × 1° grid during the monsoon season. We then determine the spatial average on the basis of the above temporal average in the entire South Asian subcontinent. For the spatial patterns of temporal changes, we identify the results of the linear temporal changes in atmospheric conditions during the monsoon season from 2006 to 2015.

Strict selection procedures are implemented to control the quality of the data products and thereby ensure credible conclusions. The CPR cloud mask and radar reflectivity from 2B-GEOPROF are set to be more than 20 and −28 dBZ, respectively, and the quality flag from the 2B-CLDCLASS is identified as confidence to CloudSat (Mace and Zhang, 2014). We define the cloud-sky condition for all products as the scenes in which the cloud mask from the CloudSat footprint

5    exceeds 20, and the others are defined as clear-sky condition. Under this screened method of cloud, the probability of a false detection to CloudSat is less than 5% in comparison with that by CALIPSO (Mace and Zhang, 2014). The all-sky condition is the total of cloud-sky and clear-sky condition. Data quality for CALIPSO is maintained by screening the cloud layer with a high degree of confidence (> 90%) (Hu et al., 2009). We only use the radiative data from CERES with a sensor viewing zenith angle less than 60° at the surface to restrict the uncertainty from the satellite non-nadir point (Christopher and Zhang,

10   2002). Due to the SW solar radiation (< 5 μm) comprising more than 99.5% of total solar energy, it is assumed as the total incoming solar radiation at the TOA and surface in this study. Although the BUGSrad defines the SW from 0–4 μm, the solar energy at 4–5 μm is excessively low (approximately 0.39% of total solar energy) (Liou, 2002). Therefore, the effect of this difference is ignorable between the definitions of SW between CERES and BUGSrad. Surface solar radiation is analyzed in this study in a relative sense with variable aerosols, clouds, and water vapor; thus, this study focuses mostly on

15   the relative changes in solar radiation but less on the absolute values of solar radiation.

CRE has been widely used to quantify the degree of cloud–radiation interactions (Henderson et al., 2013). CRE is the net (down minus up) flux difference between the all-sky condition and clear-sky condition on the atmosphere, surface, or TOA, as shown in Eq. (1):

$$CRE = (F^{\downarrow} - F^{\uparrow})_{All-Sky} - (F^{\downarrow} - F^{\uparrow})_{Clear-Sky} , \tag{1}$$

20   where $F^{\downarrow}$ and $F^{\uparrow}$ are the downwelling and upwelling radiative fluxes at the TOA and surface, respectively. One aim of this study is to quantify the impact of cloud variations on TOA and surface fluxes. Therefore, radiative data from the CloudSat 2B-FLXHR products at the surface and TOA are used. Moreover, this study uses precipitable water (PW) variation to describe the column changes of water vapor in the atmosphere. PW is the total water vapor in the atmospheric column, which is calculated on the basis of the data on environmental parameters from the ECMWF-AUX product, as shown in Eq.

25   (2):

$$PW = \frac{1}{g_0} \int_{P_{TOA}}^{P_{Surface}} q(P)dP , \tag{2}$$

where $g_0 = 9.80665$ m s$^{-2}$ is the standard acceleration due to gravity at the mean sea level (m.s.l.), $P$ is the atmospheric pressure, and $q(P)$ is the specific humidity of air as a function of atmospheric pressure (Bock et al., 2010). The PW data from ECMWF-AUX have been verified with a bias of approximately −1 mm depending on water vapor amount by comparing

30   observations from 21 ground-based Global Positioning System receiving stations and 14 radiosonde stations (Bock et al.,

2010). Therefore, although the PW inferred from ECMWF-AUX is generally slightly lower than that under actual condition, it can adequately indicate the change in water vapor amount in the South Asian subcontinent.

We accurately evaluate the effect of water vapor variations on solar radiation using the official radiative transfer model of CloudSat 2B-FLXHR (BUGSrad). By using BUGSrad, it is convenient to compare the results of the radiative transfer

5    model with those of the CloudSat product. We use the enviromental parameters provided by CloudSat products as the input of the BUGSrad, including temperature, pressure, specific humidity and cloud mask, and so on. Then, we only change the average water vapor amount during the monsoon season, and other environmental parameters are set as the inter-annual average values during the monsoon season from 2006 to 2015 to exclude the effect of other factors. For the radiative transfer model, we directly use the averaged environmental parameters during the monsoon season to evaluate the radiative effect of

10    water vapor by considering the quantity of the calculation. Due to the difference of averaged methods which were used by CloudSat and BUGSrad to evaluate the radiative effect of water vapor, there may are some difference between the radiative fluxes calculated based on CloudSat and BUGSrad.

**3 Results and Discussion**

**3.1 Changes in aerosols and solar radiation**

15    Figure 2 shows the recent changes in aerosol loading during different seasons, namely, pre-monsoon, monsoon, and dry seasons, respectively. These three seasons occur in the periods of March–May, June–September, and October–February, respectively. The aerosol loading increased, and the change in the average AOD detected by CALIPSO (approximately 0.13) was more intense than that detected by MODIS (approximately 0.05) during the monsoon season in the South Asian subcontinent from 2006 to 2015 (Figure 2a). These phenomena are mainly due to the effect of aerosols under clouds, which

20    can be detected by CALIPSO but missed by MODIS because of MODIS only inferring the aerosol loading under clear-sky condition (Winker et al., 2010). Moreover, CALIPSO can identify aerosol and cloud layer with ~90% accuracy. The sum of dust, smoke, and polluted dust aerosol loadings is close to the total aerosol loading (orange line in Figure 2a). This phenomenon indicates that aerosols mostly consist of dust, smoke, and polluted dust during the monsoon season in the South Asian subcontinent, which is consistent with the previous study (Das et al., 2015). These aerosols all present distinct

25    absorbing effects on solar radiation, thereby attenuating the incoming solar radiation more intensively than other types of aerosols (Logan et al., 2013).

Figure 2b shows the consistent increase in AOD during the pre-monsoon, monsoon, and dry seasons, respectively. The increment of AOD during the pre-monsoon and dry seasons (by 0.08 and 0.11, respectively) were greater than those during the monsoon season according to the MODIS observations from 2006 to 2015. The significant level of AOD variation

30    detected by MODIS was higher during non-monsoon season ($p < 0.01$) than during the monsoon season, which mainly due to the disturbance of widespread clouds during the monsoon season (Remer et al., 2005). The change in the average AOD

detected by CALIPSO showed a higher significance level (*p*=0.07) than that detected by MODIS due to the minimal disturbance of thin clouds and surface albedos to the aerosols detected by CALIPSO during the monsoon season (Redemann et al., 2012). Additionally, the robust and quick increase in aerosols in the South Asia has been verified by many previous studies (Suresh et al., 2013; Reddy et al., 2016; Srivastava, 2016).

5      As shown in Figure 3a, the average SSR (red line) in all-sky condition from CERES increased by 16.2 W m$^{-2}$ increments during the monsoon season in the South Asian subcontinent from 2006 to 2015. Moreover, the change in OSR (by 13.0 W m$^{-2}$) showed a highly negative correlation with that in the SSR during the study period (Figure 3b). These phenomena indicate that the attenuating effect of the atmosphere on solar radiation is gradually weakened, thereby increasing the incident solar radiation at the surface (brightening). There is the fact that the SSR is highly negatively correlated with

10 aerosol loading (Folini and Wild, 2011). Therefore, the increase in AOD did not contribute to the recent brightening during the monsoon season in the South Asian subcontinent from 2006 to 2015.

     Further, Figure 3a shows no distinct change in the SSR during the pre-monsoon season, even the reverse trend of SSR during the dry season (dimming) comparing to that during the monsoon season. The seasonal differences in OSR variations are similar to those of SSR (Figure 3b). Moreover, the trend of SSR during the dry season presented a higher significance

15 level (*p*<0.01) than that during the monsoon season (*p*=0.40). Many reasons can contribute to the significance levels during different seasons. The average cloud fractions during the pre-monsoon and dry seasons are both approximately 20%, and much lower than that during the monsoon season (53.4%) according to CloudSat observations. Moreover, under clear-sky condition, aerosols control the change in the solar radiation that reaches the surface (Nair et al., 2016; Soni et al., 2016). Therefore, aerosols have greater weighted impact on solar radiation during the dry season than during the monsoon season.

20 In addition, the resultant effects of aerosol, cloud, and water vapor are considerably complicated and varied due to the intensive atmospheric activities during the monsoon season. The pre-monsoon season is the transition season from the dry season to the monsoon season, and associated with the establishment of the southwestern wind regime over the South Asian subcontinent (Das et al., 2015). The atmospheric condition and radiation variations are not distinct comparing to the trend of SSR between the dry and monsoon seasons.

25 **3.2 Effect of clouds**

     Figure 4 shows the inter-annual variations in the average cloud vertical frequency distribution in all-sky condition and the cloud microphysical parameters in cloud-sky condition during the monsoon season in the South Asian subcontinent from 2006 to 2015. The cloud vertical frequency sequentially decreased during the monsoon season in the South Asian subcontinent from 2006 to 2015, with a maximum decrement of 7.1%, which contributed to approximately 30% of the

30 maximum cloud vertical frequency at approximately 12 km m.s.l. (Figure 4a). The cloud water content and number concentration of liquid and ice in cloud-sky condition showed a consistently decreasing trend, with maximum decrements of 18.0 mg m$^{-3}$ of LWC and 1.4 mg m$^{-3}$ of IWC, as well as approximately 6.0 cm$^{-3}$ of LNC and 2.0 L$^{-3}$ of INC (Figures 4d and

4g). These changes are consistent with those in the cloud vertical frequency during the study period. However, the LER and IER variations were not distinct and not highly consistent with the changes in the cloud vertical frequency (Figures 4c and 4f). Therefore, the clouds decreased with the reduced cloud water content and number concentration, but no clear change was observed in the particle effective radius during the monsoon season in the South Asian subcontinent from 2006 to 2015.

As shown in Figure 5a, the total cloud fraction detected by CloudSat declined by 8.8% ($p$=0.07) during the monsoon season in the South Asian subcontinent from 2006 to 2015. Furthermore, the change in CWP in cloud-sky condition was consistent with that of the total cloud fraction with a decrement of 54.0 g m$^{-2}$ ($p$<0.01). The uppermost cloud top height (CTH) declined by 661 m ($p$=0.08), and the cloud geometrical depth (CGD) declined by 280 m ($p$=0.21) with partial fluctuations in 2009 and 2010. By contrast, an insignificant trend occurred in the lowermost cloud base height (CBH) variation. We then further analyze the spatial variations in cloud fraction, cloud height (CH), CWP, and CGD during the monsoon season in the South Asian subcontinent from 2006 to 2015. Figures 5c–5f illustrate the consistent decreases in cloud fraction, CH, CWP, and CGD, especially for the cloud fraction and CWP with significant and consistent changes in space. CH and CGD increased in certain parts of the western coastal area in the South Asian subcontinent. In general, we can conclude that the clouds decreased and thinned with decreases in water content and particle number concentration during the monsoon season in the South Asian subcontinent from 2006 to 2015, thereby weakening the regulation of cloud on radiation.

The factors contributing to the changes in clouds are complicated and varied in the South Asian subcontinent. Ackerman et al. (2000) verified that aerosols with intensive absorption may lead to evaporation in the cloud layers (the aerosol semi-direct effect), burning off the clouds in the South Asia, including widespread dust, polluted dust, and smoke. Bollasina et al. (2011) showed the anthropogenic aerosol emissions weakened the South Asian summer monsoon, causing a decrease in the observed occurrence and amount of precipitation. In the global scale, observed and simulated cloud change patterns are consistent with the poleward retreat of mid-latitude storm tracks, which is attributed to the increasing greenhouse gas concentrations and recovery from volcanic radiative cooling (Norris et al., 2016). Actual, with the reduction of clouds, the recent weakened rainfall and dried monsoon during the summer monsoon season have been verified by many previous studies (Bollasina et al., 2011; Turner and Annamalai, 2012; Annamalai et al., 2013). However, the decreasing clouds also cause the increase in SSR, possibly enhancing the generation of rainfall (Wild, 2009). Therefore, the weakened rainfall was not steady, and can be attributed to the resultant force of many factors, including the changes in monsoon intensity, intrinsic cloud properties, and anthropogenic aerosols, to name a few.

Consequently, we investigate the radiative force of cloud vertical variations to determine their possible contribution to the recent rapid brightening (Figure 6). We use the vertical heating rates in all-sky condition to indicate the change in the vertically radiative effects of clouds (Fu and Liou, 1992). Given the changes in cloud macrophysical and microphysical characteristics, the SW heat rating decreased by 0.3–0.4 K d$^{-1}$ at an altitude of 5–15 km from 2006 to 2015, whereas a few increase in SW heat rating occurred at the low-cloud and near-surface areas (below 4 km). With the reduction of clouds, the total cloud reflection and absorption (mainly located at 5–15 km) weakened consistently, thus enhancing atmospheric

transmission, causing more SW radiation than before to reach the near-surface area (Henderson et al., 2013). Due to the extremely little cloud detected by CloudSat above 15 km, no significant change in heat rating occurred above 15 km, as shown in the cloud vertical frequency distribution in Figure 4a. However, the change in LW vertical heating rate was not distinct compared with that of SW. Correspondingly, the LW radiative heating increased by less than 0.2 K d$^{-1}$ above 10 km, whereas that of heating decreased by 0.1–0.2 K d$^{-1}$ within the cloud, especially below 4 km, from 2006 to 2015. The total cloud LW absorption emitted from the surface decreased with the reduction of clouds, thus leading to a weakened LW heating rate within the cloud layer (mainly below 10 km), and more LW emission on the top cloud layer above 10 km (L'Ecuyer et al., 2008). In general, the net vertical heating rate weakened consistently, indicating that the changes in the clouds physical characteristics weakened their vertical radiative effect of clouds.

As shown in Figure 7a, the SW CRE at the surface and TOA weakened by approximately 42.1 and 38.5 W m$^{-2}$ with significance levels $p$ of 0.05 and 0.04, respectively. Solar radiation consequently increased and reached at the surface during the monsoon season in the South Asian subcontinent from 2006 to 2015. The decrease in SW CRE at the surface and TOA was highly consistent with that of the total cloud faction (red line in Figure 7a). Although the CRE derived from CloudSat largely ignores the contribution of high thin clouds, this contribution is much smaller than that of low clouds (L'Ecuyer et al., 2008; Henderson et al., 2013). Moreover, SW CRE (Figure 7b) at the surface is negatively correlated with SSR (Figure 7c). This observation is attributed to the spatial variations in the radiative effect at the surface caused by cloud variables. In general, clouds were reduced and thinned by approximately 8.8% and 280 m with the decrease in cloud water content by 54.0 g m$^{-2}$ and particle number concentration in cloud-sky condition. Consequently, the SW radiative effect of the clouds decreased by approximately 42.1 W m$^{-2}$, and the absorption of direct solar radiation at the surface increased, thus leading to subsequent surface brightening.

**3.3 Effect of water vapor**

In this section, we evaluate the effect of water vapor on the recent brightening in the South Asian subcontinent. We focus mainly on the effect of water vapor on solar radiation in clear-sky condition to eliminate the disturbance of clouds in the evaluation of water vapor in all-sky condition. The vertical relative humidity (RH) in all-sky condition gradually decreased with a maximum decrement of approximately 10% during the monsoon season in the South Asian subcontinent from 2006 to 2015 (Figure 8a). Moreover, the change in RH in all-sky condition was highly consistent with that in clear-sky condition (Figures 8a and 8c). Figure 8b shows the spatial variation in PW in all-sky condition during the monsoon season in the South Asian subcontinent from 2006 to 2015; this spatial variation is highly consistent with that in clear-sky condition (Figure 8d). PW consistently decreased in space with a maximum decrement of 6.6 mm in the South Asian subcontinent, especially in the western and central regions. Although a small increase in water vapor (less than 2 mm) occurred in part of the South Asian region, its increment was much lower than the decrease in water vapor in most regions of the South Asian subcontinent. The decrease in atmospheric water vapor may be attributed to the weakened monsoon intensity in the recent

decade during the monsoon season in the South Asia (Turner and Annamalai, 2012).

Figure 9a shows that radiation is highly and negatively correlated with PW from ECMWF-AUX, thereby indicating that water vapor is a substantial controlling factor in solar radiation variability. This control lies in the direct and indirect effects. The former refers to the absorption of solar radiation, in which the PW decreased by nearly 2.8 mm ($p$=0.13) over the brightening period. According to the evaluation of BUGSrad, solar brightening was greater than 2.5 W m$^{-2}$ ($p$=0.14) because of the weakened absorption caused by the decrease in water vapor; the value was much lower than that of cloud variations (42.1 W m$^{-2}$). The SSR under clear-sky condition from CloudSat increased by approximately 4.8 W m$^{-2}$ ($p$=0.05). Except for the contribution of water vapor variations, this radiative change was possibly caused by the change in atmospheric conditions and incoming solar radiation provided by ECMWF and CloudSat (Yang et al., 2012). The spatial variation in SSR under clear-sky condition was highly negative and consistent with that of PW, especially in the western and central regions of the South Asian subcontinent (Figure 9b), which partly consist of deserts. Moreover, in all-sky condition, there are the indirect effects of water vapor, which is named as water vapor-cloud interaction. Generally, aside from the effect of temperature, a high concentration of water vapor favors the generation of more or stronger convective clouds and results in decreased radiation at the surface, and vice versa (Yang et al., 2012). Additionally, water vapor regulates cloud amount by affecting the aerosol-cloud interaction, which generally shows a positive relationship under moist atmospheric conditions (Chen et al., 2014). Therefore, the impact of water vapor in all-sky condition is more complicated than that in clear-sky condition owing to the direct and indirect effects of water vapor on solar radiation.

**3.4 Climate and environmental implications**

Variations in surface solar radiation can potentially significantly impact the climate system based on the long temporal correlation between solar radiation and climate factors, especially in temperature and rainfall. Observed dimming was suggested to be responsible for the absence of a significant temperature rise between the 1950s and the 1980s in various parts of the world, such as in the Arctic, China, America, and India (Ramanathan et al., 2005; Wild, 2009). However, the suppression of global warming over global land surfaces only lasted into the 1980s with the transition of dimming to brightening (Wild et al., 2005). This condition indicates that brightening might have significantly contributed to the recent rapid warming after the 1980s according to ground-based observations (Philipona et al., 2009). This temperature evolution satisfactorily fits the observational surface solar radiation variations, and points to the crucial role that dimming and brightening may play in determining global warming. Moreover, surface solar radiation variations induce changes in the surface net radiation, thereby altering the energy available for evaporation, which equals precipitation in the global annual mean. A decrease in evaporation and the same globally averaged reduction in precipitation with the globally averaged dimming occurred in the 1980s (Liepert, 2002). Wild et al. (2009) showed that the increase in available surface energy is quantitatively consistent with the observed substantial increase in land precipitation (3.5 mm y$^{-1}$ between 1986 and 2000) and the associated intensification of the land-based hydrological cycle.

The recent rapid warming in South Asia has been verified by many studies (Turner and Annamalai, 2012; Annamalai et al., 2013). Moreover, ECMWF-AUX shows that the 2 m temperature increased by 0.14 K y$^{-1}$, even for surface temperature with the increment of 0.4 K y$^{-1}$ during the monsoon season in the South Asian subcontinent from 2006 to 2015 (not shown). Syed et al. (2014) predicted that the mean surface air temperature in the monsoon season would increase from 2.5 °C to 5 °C

5     with increasing greenhouse gas concentrations by the end of the century. Models may underestimate the rate of global warming due to the insufficient consideration of dimming and brightening (Wild, 2012). With brightening considered, warming may accelerate in South Asia at unprecedented rates. However, the recent monsoon rainfall roughly weakened with no homogeneity by combining the model and observations of satellites and sites (Bollasina et al., 2011; Turner and Annamalai, 2012; Annamalai et al., 2013). Reduced clouds may indicate a decrease in average rainfall during the monsoon

10    season, whereas brightening generally enhances the generation of rainfall. Turner and Annamalai (2012) showed consistent negative trends of rainfall over northwest India and coastal Burma, while the positive trends in southeast India; they also found a distinct disagreement among different rainfall data in northeast India. In general, the recent brightening may play a prominent role in modulating the warming and rainfall variations in the South Asian subcontinent.

**4 Conclusions**

15    Surface solar radiation is the ultimate energy source for life on the planet (IPCC, 2013). Any change in surface solar radiation profoundly affects the global ecosystem, further determines the living conditions of humans (Haywood et al., 2011). South Asia is experiencing a leveling-off trend with regard to solar radiation, even a transition from dimming to brightening. This process is significantly complicated because of the active atmospheric action during the monsoon season. In this study, we use observations from multiple satellites/sensors on the A-Train satellite to evaluate the effect of aerosol, cloud, and

20    water vapor variations on the recent changes in solar radiation at the surface during the monsoon season (June–September) in the South Asian subcontinent from 2006 to 2015, mainly including CloudSat, CALIPSO, MODIS and CERES.

We found the SSR and OSR increased by 16.2 W m$^{-2}$ and decreased by 13.0 W m$^{-2}$ with a partly fluctuation during the monsoon season, respectively. On the contrary, the increase in AOD was inconsistent with the SSR variation, and did not contribute to the recent brightening in the South Asian subcontinent. Decreases in water vapor amount and clouds

25    significantly contributed to solar brightening and subsequent surface warming in the South Asian subcontinent. In general, the clouds were generally reduced and thinned by approximately 8.8% and 280 m, respectively, when the cloud water path (by 54.0 g m$^{-2}$) and particle number concentration in cloud-sky condition decreased. Given the change in clouds, the atmospheric vertical SW heat rating decreased by 0.3–0.4 K d$^{-1}$ at the altitude of 5–15 km, whereas SW heat rating slightly increased at the low-cloud and near-surface areas (below 4 km) from 2006 to 2015. Correspondingly, SW CRE weakened by

30    approximately 42.1 W m$^{-2}$ at the surface. Moreover, the PW in clear-sky condition decreased by nearly 2.8 mm over the brightening period. Consequently, solar brightening increased by approximately 2.5 W m$^{-2}$ because of the weakened absorption. The decreases in water vapor amount and clouds weakened the effect of water vapor and clouds on solar

radiation, thereby resulting in the increased absorption of direct solar radiation at the surface and subsequent surface brightening.

Notably, CloudSat is insufficient in observing thin cirrus with small IWC, thereby resulting in uncertainties for quantifying changes in clouds. In a globally averaged experiment on radiative fluxes, L'Ecuyer et al. (2008) proposed that the impact of the thin cirrus that is not detected by CloudSat on SW radiation at the surface is $-1.2$ W m$^{-2}$, which is much smaller than the impact of low clouds. The data in this study were from the instantaneous observations of multiple sensors. Thus, the magnitudes of radiative impact may were overestimated for the water vapor and cloud variations. The results are more representative of the atmospheric condition variations at nominally constant solar zenith angle than the daily average. In sum, the conclusions of this study are beneficial in the evaluation of the recent variations in aerosol, cloud, and water vapor and their resultant influences on solar radiation. Additionally, the aerosol, cloud, and water vapor variations are correlated with regional sources and meteorology. Consequently, solar dimming and brightening may be of local or regional nature, and they are unavoidably inhomogeneous and unstable in space (Soni et al., 2016). The variations in aerosols, water vapor, and clouds highly interact with changes in monsoon circulation. Thus, it would be interesting to investigate the change mechanisms of aerosols, clouds, and water vapor in South Asia. The chicken-and-egg relationship between monsoon and aerosols, water vapor, and clouds can also be explored.

**Acknowledgments**

This study was supported by the National Science Foundation of China (41627804), National Key Research and Development Program of China (2016YFC0200900), Program for Innovative Research Team in University of Ministry of Education of China (IRT1278), and China Postdoctoral Science Foundation (2016T90731). We would like to thank the CloudSat, CALIPSO, CERES, and MODIS science teams for providing excellent and accessible data products that made the study possible. The CloudSat data in this study were acquired from CloudSat Data Processing Center (DPC, http://www.cloudsat.cira.colostate.edu/). The CERES SSF and CALIPSO data were obtained from NASA Langley Research Center Atmospheric Sciences Data Center (ASDC, https://eosweb.larc.nasa.gov/). The MODIS AOD data were obtained from the NASA Earth Observing System Data and Information System, Distributed Active Archive Center (DAAC, https://ladsweb.modaps.eosdis.nasa.gov/).

**References**

Ackerman, A. S., Toon, O. B., Stevens, D. E., Heymsfield, A. J., Ramanathan, V., and Welton, E. J.: Reduction of tropical cloudiness by soot, Science, 288, 1042-1047, 2000.

Annamalai, H., Hafner, J., Sooraj, K. P., and Pillai, P.: Global Warming Shifts the Monsoon Circulation, Drying South Asia, Journal of Climate, 26, 2701-2718, 2013.

Austin, R. T., Heymsfield, A. J., and Stephens, G. L.: Retrieval of ice cloud microphysical parameters using the CloudSat millimeter-wave radar and temperature, Journal of Geophysical Research Atmospheres, 114, 1065-1066, 2009.

Bock, O., Keil, C., Richard, E., Flamant, C., and Bouin, M. N.: Validation of precipitable water from ECMWF model analyses with GPS and radiosonde data during the MAP SOP, Quarterly Journal of the Royal Meteorological Society, 131, 3013-3036, 2010.

Bollasina, M. A., Ming, Y., and Ramaswamy, V.: Anthropogenic Aerosols and the Weakening of the South Asian Summer Monsoon, Science, 334, 502-505, 2011.

5  Chen, Y. C., Christensen, M. W., Stephens, G. L., and Seinfeld, J. H.: Satellite-based estimate of global aerosol-cloud radiative forcing by marine warm clouds, Nature Geoscience, 7, 643-646, 2014.

Christensen, M. W., Stephens, G. L., and Lebsock, M. D.: Exposing biases in retrieved low-cloud properties from cloudsat: A guide for evaluating observations and climate data †, Journal of Geophysical Research Atmospheres, 118, 12120-12131, 2013.

Christensen, M. W., Chen, Y. C., and Stephens, G. L.: Aerosol indirect effect dictated by liquid clouds, Journal of Geophysical Research: 10  Atmospheres, 121, 2016.

Christopher, S. A., and Zhang, J.: Shortwave Aerosol Radiative Forcing from MODIS and CERES observations over the oceans, Geophysical Research Letters, 29, 6-1–6-4, 2002.

Das, S., Dey, S., Dash, S. K., Giuliani, G., and Solmon, F.: Dust aerosol feedback on the Indian summer monsoon: Sensitivity to absorption property, Journal of Geophysical Research Atmospheres, 120, 9642-9652, 2015.

15  Folini, D., and Wild, M.: Aerosol emissions and dimming/brightening in Europe: Sensitivity studies with ECHAM5-HAM, Journal of Geophysical Research Atmospheres, 116, 21104, 2011.

Fu, Q., and Liou, K. N.: On the correlated k-distribution method for radiative transfer in nonhomogeneous atmospheres, Journal of Atmospheric Sciences, 49, 2139 - 2156, 1992.

Grandey, B. S., Stier, P., and Wagner, T. M.: Investigating relationships between aerosol optical depth and cloud fraction using satellite, 20  aerosol reanalysis and general circulation model data, Atmospheric Chemistry & Physics, 17, 30805-30823, 2013.

Haywood, J. M., Nicolas, B., Andy, J., Olivier, B., Martin, W., and Shine, K. P.: The roles of aerosol, water vapor and cloud in future global dimming/brightening, Journal of Geophysical Research Atmospheres, 116, -, 2011.

Henderson, D. S., L'Ecuyer, T., Stephens, G., Partain, P., and Sekiguchi, M.: A Multisensor Perspective on the Radiative Impacts of Clouds and Aerosols, Journal of Applied Meteorology & Climatology, 52, 853-871, 2013.

25  Heymsfield, A. J., Wang, Z., and Matrosov, S.: Improved Radar Ice Water Content Retrieval Algorithms Using Coincident Microphysical and Radar Measurements, Journal of Applied Meteorology, 44, 1391-1412, 2010.

Hu, Y., Winker, D., Vaughan, M., Lin, B., Omar, A., Trepte, C., Flittner, D., Yang, P., Nasiri, S. L., and Baum, B.: CALIPSO/CALIOP Cloud Phase Discrimination Algorithm, Journal of Atmospheric & Oceanic Technology, 26, 2293-2309, 2009.

IPCC: Climate Change 2013: The Physical Science Basis. Contribution of Working Group I to the Fifth Assessment Report of the 30  Intergovernmental Panel on Climate Change, Cambridge University Press, New York, USA, 2013.

Kambezidis, H. D., Kaskaoutis, D. G., Kharol, S. K., Moorthy, K. K., Satheesh, S. K., Kalapureddy, M. C. R., Badarinath, K. V. S., Sharma, A. R., and Wild, M.: Multi-decadal variation of the net downward shortwave radiation over south Asia: The solar dimming effect, Atmospheric Environment, 50, 360-372, 2012.

Kim, D., and Ramanathan, V.: Solar radiation budget and radiative forcing due to aerosols and clouds, Journal of Geophysical Research 35  Atmospheres, 113, 194-204, 2008.

Kvalevåg, M. M., and Myhre, G.: Human Impact on Direct and Diffuse Solar Radiation during the Industrial Era, Journal of Climate, 20, 4874-4883, 2007.

L'Ecuyer, T. S., Wood, N. B., Haladay, T., Stephens, G. L., and Stackhouse, P. W.: Impact of clouds on atmospheric heating based on the R04 CloudSat fluxes and heating rates data set, Journal of Geophysical Research Atmospheres, 113, 2013-2018, 2008.

40  Levy, R. C., Mattoo, S., Munchak, L. A., and Remer, L. A.: The Collection 6 MODIS aerosol products over land and ocean, Atmospheric Measurement Techniques, 6, 2989-3034, 2013.

Liepert, B. G.: Observed reductions of surface solar radiation at sites in the United States and worldwide from 1961 to 1990, Geophysical Research Letters, 29, 61-61–61-64, 2002.

Liou, K.-N.: An Introduction to Atmospheric Radiation, Academic press, San Diego, USA, 2002.

45  Liu, Z., Vaughan, M., Winker, D., Kittaka, C., Getzewich, B., Kuehn, R., Omar, A., Powell, K., Trepte, C., and Hostetler, C.: The CALIPSO Lidar Cloud and Aerosol Discrimination: Version 2 Algorithm and Initial Assessment of Performance, Journal of Atmospheric & Oceanic Technology, 26, 1198-1213, 2009.

Loeb, N. G., and Manalosmith, N.: Top-of-Atmosphere Direct Radiative Effect of Aerosols over Global Oceans from Merged CERES and MODIS Observations, Journal of Climate, 18, 3506-3526, 2005.

50  Logan, T., Xi, B., Dong, X., Li, Z., and Cribb, M.: Classification and investigation of Asian aerosol absorptive properties, Atmospheric Chemistry and Physics, 13, 2253-2265, 2013.

Long, C. N., Dutton, E. G., Augustine, J. A., Wiscombe, W., Wild, M., Mcfarlane, S. A., and Flynn, C. J.: Significant decadal brightening of downwelling shortwave in the continental United States, Journal of Geophysical Research Atmospheres, 114, 1291-1298, 2009.

Mace, G. G., Zhang, Q., Vaughan, M., Marchand, R., Stephens, G., Trepte, C., and Winker, D.: A description of hydrometeor layer 55  occurrence statistics derived from the first year of merged Cloudsat and CALIPSO data, Journal of Geophysical Research: Atmospheres, 114, 414-416, 10.1029/2007JD009755, 2009.

Mace, G. G., and Zhang, Q.: The CloudSat radar-lidar geometrical profile product (RL-GeoProf): Updates, improvements, and selected results, Journal of Geophysical Research: Atmospheres, 119, 9441-9462, 2014.

Munchak, L. A., Levy, R. C., Mattoo, S., and Remer, L. A.: MODIS 3 km aerosol product: applications over land in an urban/suburban region, Atmospheric Measurement Techniques, 6, 1683-1716, 2013.

5 Nair, V. S., Babu, S. S., Manoj, M. R., Moorthy, K. K., and Chin, M.: Direct radiative effects of aerosols over South Asia from observations and modeling, Climate Dynamics, 1-18, 2016.

Norris, J. R., Allen, R. J., Evan, A. T., Zelinka, M. D., O'Dell, C. W., and Klein, S. A.: Evidence for climate change in the satellite cloud record, Nature, 536, 72, 2016.

Omar, A. H., Winker, D. M., Kittaka, C., Vaughan, M. A., Liu, Z., Hu, Y., Trepte, C. R., Rogers, R. R., Ferrare, R. A., and Lee, K.-P.: The
10 CALIPSO Automated Aerosol Classification and Lidar Ratio Selection Algorithm, Journal of Atmospheric and Oceanic Technology, 26, 1994, 2009.

Padma, K. B., Londhe, A. L., Daniel, S., and Jadhav, D. B.: Observational evidence of solar dimming: Offsetting surface warming over India, Geophysical Research Letters, 34, 377-390, 2007.

Pan, Z., Gong, W., Mao, F., Li, J., Wang, W., Li, C., and Min, Q.: Macrophysical and optical properties of clouds over East Asia measured
15 by CALIPSO, Journal of Geophysical Research: Atmospheres, 120, 11653-11668, 10.1002/2015JD023735, 2015.

Pan, Z., Mao, F., Gong, W., Min, Q., and Wang, W.: The warming of Tibetan Plateau enhanced by 3D variation of low-level clouds during daytime, Remote Sensing of Environment, 198, 363-368, 2017.

Philipona, R., Behrens, K., and Ruckstuhl, C.: How declining aerosols and rising greenhouse gases forced rapid warming in Europe since the 1980s, Geophysical Research Letters, 36, 206-218, 2009.

20 Rajeevan, M., Rohini, P., Kumar, K. N., Srinivasan, J., and Unnikrishnan, C. K.: A Study of Vertical Cloud Structure of the Indian Summer Monsoon using CloudSat data, Climate Dynamics, 40, 637-650, 2012.

Ramanathan, V., Chung, C., Kim, D., Bettge, T., Buja, L., Kiehl, J., Washington, W., Fu, Q., Sikka, D., and Wild, M.: Atmospheric brown clouds: Impacts on South Asian climate and hydrological cycle, Proceedings of the National Academy of Sciences, 102, 5326-5333, 2005.

Reddy, K. R. O., Balakrishnaiah, G., Gopal, K. R., Reddy, N. S. K., Rao, T. C., Reddy, T. L., Hussain, S. N., Reddy, M. V., Reddy, R. R.,
25 and Boreddy, S. K. R.: Long term (2007–2013) observations of columnar aerosol optical properties and retrieved size distributions over Anantapur, India using multi wavelength solar radiometer, Atmospheric Environment, 142, 238-250, 2016.

Redemann, J., Vaughan, M. A., Zhang, Q., Shinozuka, Y., Russell, P. B., Livingston, J. M., Kacenelenbogen, M., and Remer, L. A.: The comparison of MODIS-Aqua (C5) and CALIOP (V2 & V3) aerosol optical depth, Atmospheric Chemistry and Physics, 12, 3025-3043, 2012.

30 Remer, L. A., Kaufman, Y. J., Tanré, D., Mattoo, S., Chu, D. A., Martins, J. V., Li, R. R., Ichoku, C., Levy, R. C., and Kleidman, R. G.: The MODIS Aerosol Algorithm, Products, and Validation, Journal of Atmospheric Sciences, 62, 947-973, 2005.

Sohn, B. J., Nakajima, T., Satoh, M., and Jang, H. S.: Impact of different definitions of clear-sky flux on the determination of longwave cloud radiative forcing: NICAM simulation results, Atmospheric Chemistry and Physics, 10, 11641-11646, 2010.

Soni, V. K., Pandithurai, G., and Pai, D. S.: Is there a transition of solar radiation from dimming to brightening over India?, Atmospheric
35 Research, 169, 209-224, 2016.

Srivastava, R.: Trends in aerosol optical properties over South Asia, International Journal of Climatology, 37, n/a-n/a, 2016.

Stephens, G., Winker, D., Pelon, J., Trepte, C., Vane, D., Yuhas, C., L'Ecuyer, T., and Lebsock, M.: CloudSat and CALIPSO within the A-Train: Ten years of actively observing the Earth system, Bulletin of the American Meteorological Society, 0, null, 10.1175/bams-d-16-0324.1, 2017.

40 Stephens, G. L., Gabriel, P. M., and Partain, P. T.: Parameterization of Atmospheric Radiative Transfer. Part I: Validity of Simple Models, Journal of the Atmospheric Sciences, 58, 3391-3409, 2001.

Stephens, G. L., Wood, N. B., and Gabriel, P. M.: An Assessment of the Parameterization of Subgrid-Scale Cloud Effects on Radiative Transfer. Part I: Vertical Overlap, Journal of the Atmospheric Sciences, 61, 47-52, 2003.

Suresh, B. S., Manoj, M. R., Krishna, M. K., Gogoi, M. M., Nair, V. S., Kumar, K. S., Satheesh, S. K., Niranjan, K., Ramagopal, K., and
45 Bhuyan, P. K.: Trends in aerosol optical depth over Indian region: Potential causes and impact indicators, Journal of Geophysical Research Atmospheres, 118, 11,794–711,806, 2013.

Syed, F. S., Iqbal, W., Syed, A. A. B., and Rasul, G.: Uncertainties in the regional climate models simulations of South-Asian summer monsoon and climate change, Climate Dynamics, 42, 2079-2097, 2014.

Turner, A. G., and Annamalai, H.: Climate change and the South Asian summer monsoon, Nature Climate Change, 2, 587-595, 2012.

50 Uppala, S. M., Kållberg, P. W., Simmons, A. J., Andrae, U., Bechtold, V. D. C., Fiorino, M., Gibson, J. K., Haseler, J., Hernandez, A., and Kelly, G. A.: The ERA-40 re-analysis, Quarterly Journal of the Royal Meteorological Society, 131, 2961-3012, 2005.

Wang, W., Mao, F., Pan, Z., Du, L., and Gong, W.: Validation of VIIRS AOD through a Comparison with a Sun Photometer and MODIS AODs over Wuhan, Remote Sensing, 9, 403, 2017.

Wang, Y., and Wild, M.: A new look at solar dimming and brightening in China, Geophysical Research Letters, 43, 2016.

55 Wielicki, B. A., Barkstrom, B. R., Harrison, E. F., Iii, R. B. L., Smith, G. L., and Cooper, J. E.: Clouds and the Earth's Radiant Energy System (CERES): An Earth Observing System Experiment, Bulletin of the American Meteorological Society, 77, 853-868, 2015.

Wild, M., Gilgen, H., Roesch, A., Ohmura, A., Long, C. N., Dutton, E. G., Forgan, B., Kallis, A., Russak, V., and Tsvetkov, A.: From dimming to brightening: decadal changes in solar radiation at Earth's surface, Science, 308, 847-850, 2005.

Wild, M.: Global dimming and brightening: A review, Journal of Geophysical Research Atmospheres, 114, D00D16, 2009.

Wild, M., Trüssel, B., Ohmura, A., Long, C. N., König‑Langlo, G., Dutton, E. G., and Anatoly, T.: Global dimming and brightening: An update beyond 2000, Journal of Geophysical Research Atmospheres, 114, 895-896, 2009.

Wild, M.: Enlightening Global Dimming and Brightening, Bulletin of the American Meteorological Society, 93, 27-37, 2012.

Willson, R. C., and Mordvinov, A. V.: Secular total solar irradiance trend during solar cycles 21–23, Geophysical Research Letters, 30, 3-1, 2003.

Winker, D. M., Hunt, W. H., and McGill, M. J.: Initial performance assessment of CALIOP, Geophysical Research Letters, 34, 228-262, 2007.

Winker, D. M., Pelon, J., Coakley, J. A., Ackerman, S. A., Charlson, R. J., Colarco, P. R., Flamant, P., Fu, Q., Hoff, R. M., and Kittaka, C.: The CALIPSO Mission: A Global 3D View of Aerosols and Clouds, Bulletin of the American Meteorological Society, 91, 1211-1229, 2010.

Woods, C. P., Waliser, D. E., Li, J. L., Austin, R. T., Stephens, G. L., and Vane, D. G.: Evaluating CloudSat ice water content retrievals using a cloud-resolving model: Sensitivities to frozen particle properties, Journal of Geophysical Research, 113, 2739-2740, 2008.

Yang, K., Ding, B., Qin, J., Tang, W., Ning, L., and Lin, C.: Can aerosol loading explain the solar dimming over the Tibetan Plateau, Geophysical Research Letters, 39, L20710, 2012.

**Table 1. Observations used in analysis along with their sources and spatial resolutions.**

| Sensor | Spatial Resolution | Products | Parameter |
|---|---|---|---|
| CloudSat | 1.3 × 1.7 km | 2B-CLDCLASS | Cloud fraction; cloud top/base height |
| | | 2B-CWC-RO | Cloud water content, particle effective radius, and particle number concentration |
| | | 2B-FLXHR | Vertical heating rate; SW/LW radiative fluxes |
| CALIPSO | 5 × 5 km | Level 2 Aerosol Layer | Aerosol optical depth; vertical feature mask |
| MODIS | 1° × 1° | Level 3 MYD08 | Aerosol optical depth |
| ECMWF | 2.5° × 2.5° | ECMWF-AUX | Pressure; temperature; relative humidity |
| CERES | 20 × 20 km | Level 2 Single Scanner Footprint | SW radiation at TOA and surface |

[Figure]

**Figure 1. Terrain map during the monsoon season in the South Asia and surrounding regions. The blue line indicates rivers; the study region is within the red line.**

[Figure]

**Figure 2. Temporal variations in spatial average AOD from (a) CALIPSO and MODIS during the monsoon season, as well as (b) MODIS during pre-monsoon, monsoon, and dry seasons in the South Asian subcontinent from 2006 to 2015, respectively. The dashed lines indicate the linear fit line of the solid line with the same color; $p$ is the significance level; $k$ is the slope of the line trend of each time series.**

[Figure]

**Figure 3. Temporal variations in spatial average (a) SSR and (b) OSR from CERES during pre-monsoon, monsoon, and dry seasons in the South Asian subcontinent from 2006 to 2015, respectively. The dashed lines indicate the linear fit line of the solid line with the same color; *p* is the significance level; *k* is the slope of the line trend of each time series.**

[Figure]

**Figure 4.** Temporal variations in spatial average vertical cloud physical parameters from CloudSat during the monsoon season in the South Asian subcontinent from 2006 to 2015: (a) cloud vertical frequency distribution, (b) LWC and (e) IWC, (c) LER and (f) IER, and (d) LNC and (g) INC.

[Figure]

**Figure 5. Temporal variations in spatial average (a) cloud fraction and CWP, as well as (b) uppermost CTH, lowermost CBH, and CGD; spatial distributions of the temporal changes in average (c) cloud fraction, (d) CH, (e) CWP, and (f) CGD from CloudSat during the monsoon season in the South Asian subcontinent from 2006 to 2015. The dashed lines indicate the linear fit line of the solid line with the same color; _p_ is the significance level; _k_ is the slope of the line trend of each time series.**

[Figure]

**Figure 6. Temporal variations in spatial average vertical (a) SW, (b) LW, and (c) net heat rating in all-sky condition from CloudSat during the monsoon season in the South Asian subcontinent from 2006 to 2015.**

[Figure]

**Figure 7. Temporal variations in spatial average (a) cloud fraction and CRE, as well as spatial distribution of the temporal changes in average (b) SW CRE at the surface and (c) SSR from CloudSat during the monsoon season in the South Asian subcontinent from 2006 to 2015. The dashed lines indicate the linear fit line of the solid line with the same color; *p* is the significance level; *k* is the slope of the line trend of each time series.**

[Figure]

**Figure 8. Temporal variations in spatial average vertical RH in (a) all-sky and (c) clear-sky condition; spatial distribution of temporal changes in average PW in (b) all-sky and (d) clear-sky condition from ECMWF-AUX during the monsoon season in the South Asian subcontinent from 2006 to 2015.**

[Figure]

**Figure 9.** Temporal variations in spatial average (a) PW (blue line), as well as SSR from BUGSrad (red line) and CloudSat (orange line); spatial distribution of temporal changes in average (b) SSR from BUGSrad in clear-sky condition during the monsoon season in the South Asian subcontinent from 2006 to 2015. The dashed lines indicate the linear fit line of the solid line with the same color; $p$ is the significance level; $k$ is the slope of the line trend of each time series.